# Global donor and acceptor splicing site kinetics in human cells

Leonhard Wachutka[1†], Livia Caizzi[2†], Julien Gagneur[1*], Patrick Cramer[2*]

[1]Department of Informatics, Technical University of Munich, Garching, Germany; [2]Department of Molecular Biology, Max-Planck-Institute for Biophysical Chemistry, Göttingen, Germany

**Abstract** RNA splicing is an essential part of eukaryotic gene expression. Although the mechanism of splicing has been extensively studied in vitro, in vivo kinetics for the two-step splicing reaction remain poorly understood. Here, we combine transient transcriptome sequencing (TT-seq) and mathematical modeling to quantify RNA metabolic rates at donor and acceptor splice sites across the human genome. Splicing occurs in the range of minutes and is limited by the speed of RNA polymerase elongation. Splicing kinetics strongly depends on the position and nature of nucleotides flanking splice sites, and on structural interactions between unspliced RNA and small nuclear RNAs in spliceosomal intermediates. Finally, we introduce the 'yield' of splicing as the efficiency of converting unspliced to spliced RNA and show that it is highest for mRNAs and independent of splicing kinetics. These results lead to quantitative models describing how splicing rates and yield are encoded in the human genome.
DOI: https://doi.org/10.7554/eLife.45056.001

*For correspondence:
gagneur@in.tum.de (JG);
patrick.cramer@mpibpc.mpg.de
(PC)

[†]These authors contributed equally to this work

**Competing interests:** The authors declare that no competing interests exist.

## Introduction

Transcription of eukaryotic genes produces precursor RNA molecules that are processed by splicing. Splicing is a two-step reaction that results in the removal of introns from precursor RNA and the formation of mature RNA with joined exons. Splicing is catalyzed by the spliceosome, a dynamic ribonucleoprotein machine assembled from small nuclear ribonucleoprotein complexes (snRNPs) and several non-RNP factors (*Herzel et al., 2017*; *Wahl et al., 2009*). In metazoa, the majority of the introns are excised by the major spliceosome, whereas a minority is removed by the minor spliceosome (*Will and Lührmann, 2011*). Spliceosomes are recruited through conserved RNA elements, the 5′-splice site at the exon-intron border (donor site), the 3′-splice site at the intron-exon border (acceptor site), and the branch point, which is followed by a polypyrimidine track (*Coolidge et al., 1997*; *Turunen et al., 2013*) and located ~18–40 nucleotides (nt) upstream of the acceptor site (*Ruskin and Green, 1985*; *Ruskin et al., 1985*; *Taggart et al., 2012*). Additional RNA sequences such as splicing enhancers and silencers are found in both introns and exons and can influence the choice between splice sites (*Zhu et al., 2001*).

In recent years, the intricate mechanisms of splicing were investigated with a combination of structural and functional studies (*Mayerle and Guthrie, 2017*; *Shi, 2017*; *Wilkinson et al., 2018*; *Will and Lührmann, 2011*). The spliceosome assembles in a stepwise manner, adopting different intermediates with varying composition, conformation, and interactions between RNAs and proteins (*Will and Lührmann, 2011*). Assembly of the major spliceosome begins with recognition of the donor site by U1 snRNP (*Kondo et al., 2015*; *Lerner et al., 1980*; *Seraphin and Rosbash, 1989*; *Zhuang and Weiner, 1986*). The U2 auxiliary factor then binds to the poly-pyrimidine track and the acceptor site (*Valcárcel et al., 1996*; *Zamore and Green, 1989*; *Zamore et al., 1992*) generating the E complex. U2 snRNP then binds the branchpoint, resulting in the A complex (*Konarska and Sharp, 1986*; *Perriman and Ares, 2010*). In the A complex, U1 snRNA binds the donor site and U2

**eLife digest** Genes are portions of DNA that carry the instructions to build proteins. In particular, they are formed of segments called exons, which contain the protein-building information, and of non-coding segments known as introns. Exons and introns alternate within a gene.

To create a given protein, the cell first uses an enzyme, Polymerase II, to copy the entire related gene – including introns and exons – into a molecule of ribonucleic acid, or RNA. As the gene is copied, a machine called the spliceosome comes onto the RNA molecule to remove the introns and create the final RNA template used to produce proteins.

The spliceosome works by recognizing specific sequences that signal the border between introns and exons. Once the machine is bound to these 'splice sites' on each side of an intron, it brings the two neighboring exons close together and cuts out the intron. The two ends of the exons are then attached together. Previous studies have measured how fast introns are removed, but it remained unclear how long it takes to cut individual splice sites genome-wide.

To address this question, Wachutka, Caizzi et al. combined a mathematical approach with a biochemical method that purifies newly made RNA in human cells. The experiments showed that it only took a few minutes to cut most splice sites. Cutting splice sites that bordered very long introns was slower, presumably because the Polymerase II took longer to produce these introns. In addition, the genetic sequences of the splice sites affected the time it took to remove the introns: some made it harder for the spliceosome to recognize where to cut, but others made it easier.

Mistakes in removing introns from RNA can lead to producing abnormal proteins, and many diseases such as cystic fibrosis and Duchenne muscular dystrophy can be caused by such errors. In particular, small changes in the sequences at the splice sites or in the surrounding areas can create problems when it comes to eliminating introns. Decrypting the dynamics of intron cutting and removal may give scientists new insight into the molecular causes of cystic fibrosis and many other genetic disorders.

DOI: https://doi.org/10.7554/eLife.45056.002

snRNA binds the branchpoint, rendering the branchpoint adenosine base available for interaction with the acceptor site (*Berglund et al., 2001*; *Query et al., 1994*). Binding of the U4/U5/U6 tri-snRNP leads to the B complex (*Bertram et al., 2017a*), which is first activated (Bact) and then converted to the catalytically active B* complex. In the B* complex, the donor site is positioned close to the branchpoint in a RNA network formed between the precursor RNA and U2, U5 and U6 snRNAs (*Zhang et al., 2018*).

The activated spliceosome catalyzes intron removal in two steps, which are both transesterification reactions. In the first step, the 2′-hydroxyl group of the branchpoint adenosine serves as a nucleophile to attack the donor site and to generate a cleaved 5′-exon and the lariat intermediate. This leads to the C complex (*Zhan et al., 2018*) that is then rearranged to form the C* complex, which is catalytically active to carry out the second step. In the C* complex, the ends of exons to be joined are juxtaposed (*Bertram et al., 2017b*; *Zhang et al., 2017*), and this enables the 3′-hydroxyl group of the last nucleotide of the 5′-exon to attack the acceptor site, leading to exon ligation and excision of the intron lariat. The resulting P complex contains the ligated exons, which are subsequently released, completing the splicing process (*Bai et al., 2017*; *Wilkinson et al., 2017*).

Taken together, extensive in vitro studies have strongly advanced our understanding of the splicing process, but the kinetics and mechanisms of splicing in vivo remain far less understood. Although biochemical assays show that splicing can occur in the absence of transcription, in vivo splicing happens mainly co-transcriptionally, when newly transcribed RNA is still attached to RNA polymerase II (Pol II) and chromatin (for review see *Alexander and Beggs, 2010*; *Bentley, 2014*; *Saldi et al., 2016*). Furthermore, compromising Pol II transcription elongation increases alternative splicing (*de la Mata et al., 2003*; *Dujardin et al., 2014*; *Ip et al., 2011*; *Pagani et al., 2003*), providing evidence that an optimal elongation rate is essential for a co-transcriptional splicing (*Davis-Turak et al., 2018*; *Fong et al., 2014*). Native elongating transcript sequencing in human cells (NET-seq) indicates that splicing occurs soon after introns are synthesized (*Mayer et al., 2015*;

*Nojima et al., 2015*). Further, the combination of single molecule intron tracking (SMIT) and long read sequencing in yeast shows that splicing is 50% complete when Pol II is 45 nt downstream the acceptor spice site (*Oesterreich et al., 2016*).

Despite these advances, the in vivo kinetics of splicing remain poorly understood. In particular, different estimates for splicing rates have been reported. Splicing rates have been measured for selected endogenous human genes with the use of live cell imaging (*Coulon et al., 2014*; *Martin et al., 2013*; *Rino et al., 2014*; *Schmidt et al., 2011*) or with a combination of cellular RNA extraction and quantitative PCR (*Pandya-Jones and Black, 2009*; *Singh and Padgett, 2009*). Such studies led to very different splicing rate estimates, ranging from 15 to 30 s (*Huranová et al., 2010*; *Martin et al., 2013*; *Rino et al., 2014*) to 4.3-10 min (*Coulon et al., 2014*; *Schmidt et al., 2011*; *Singh and Padgett, 2009*) per splicing event. These discrepancies may stem from the difference in methods used, which can introduce perturbations, from the selection of genes studied, and from the great variance in intron lengths between human genes.

Other studies have estimated in vivo splicing rates globally with the use of RNA sequencing technologies. The short sequence reads collected from steady-state in vivo samples reflect RNA synthesis, splicing and degradation, which are entangled (*Wachutka and Gagneur, 2017*). In particular, the 'splicing efficiency' is typically defined by the ratio of spliced over unspliced RNAs at steady state (*Braberg et al., 2013*; *Wilhelm et al., 2008*). However, the same ratio has been successfully employed to study RNA stability, with the assumption that unspliced RNA levels reflect RNA synthesis and spliced RNA levels reflect the ratio of RNA synthesis over degradation (*Gaidatzis et al., 2015*; *Zeisel et al., 2011*). This ambiguity in definitions and interpretations questions the use of splicing efficiency and calls for alternative concepts and metrics. To overcome the limitations of steady-state transcriptome sequencing, one approach is to sequence new transcripts from chromatin-associated RNA fractions and compare them to cytoplasmic fractions, which led to splicing rate estimates from 43 s (*Davis-Turak et al., 2015*) to 15–120 min (*Bhatt et al., 2012*; *Pandya-Jones et al., 2013*) per splicing event.

An alternative method to investigate splicing kinetics in vivo is the use of metabolic RNA labeling with 4-thiouracil (4sU) (*Dölken et al., 2008*; *Rabani et al., 2011*; *Rabani et al., 2014*; *Windhager et al., 2012*) coupled to sequencing of the labeled RNA (4sU-seq). We previously combined 4sU-seq with kinetic modeling to obtain RNA synthesis, splicing, and degradation rates in the fission yeast *S. pombe* (*Eser et al., 2016*). Others have used 4sU-seq and similar approaches to obtain median splicing rate estimates in human cells of 6.7 min (*Mukherjee et al., 2017*) or 14 min (*Rabani et al., 2014*) per splicing event. However, 4sU-seq also introduces biases when applied to the human system because the obtained data are artificially biased toward pre-existing 5'-regions of the RNA due to the length of human genes (*Schwalb et al., 2016*). These RNA 5'-regions predate the labeling time and are generally already observed to be spliced by 4sU-seq, leading to potential errors in the rate estimates. As a result of these difficulties, in vivo splicing kinetics remain unclear, and individual rate estimates for the two steps of splicing are lacking. However, such information is highly desirable because it may be interpreted alongside the mechanistic information obtained in vitro to provide a better understanding of the splicing process.

To study kinetics of splicing in vivo, we performed TT-seq (*Schwalb et al., 2016*) after different 4sU-labeling time points in human K562 cells. In contrast to 4sU-seq, the TT-seq protocol includes RNA fragmentation before 4sU-labeled RNA is purified and sequenced. This is a crucial step that eliminates 5' regions of nascent RNAs that were already transcribed, and spliced, prior to incorporation of the label, and thus removes the 5'-bias. We developed a computational approach that estimates the metabolic rates of single phosphodiester bonds. This approach enabled uncoupled quantification of donor- and acceptor-specific kinetics and to relate these to the two transesterification reactions and to the contribution of single nucleotides in spliceosome intermediates defined by structural studies. Moreover, to calculate the amount of precursor RNA successfully spliced into mature RNA, we introduced the 'splicing yield' as the conversion efficiency of unspliced to spliced RNA. As a result, our analysis provides genome-wide metabolic rates for donor and acceptor splice sites and identifies RNA-RNA interactions in the spliceosome that could contribute to in vivo splicing kinetics. Furthermore, we provide genome-wide estimates of the splicing yield that is not biased by splicing kinetics. From this work emerges a comprehensive global view of splicing kinetics and yield in human cells.

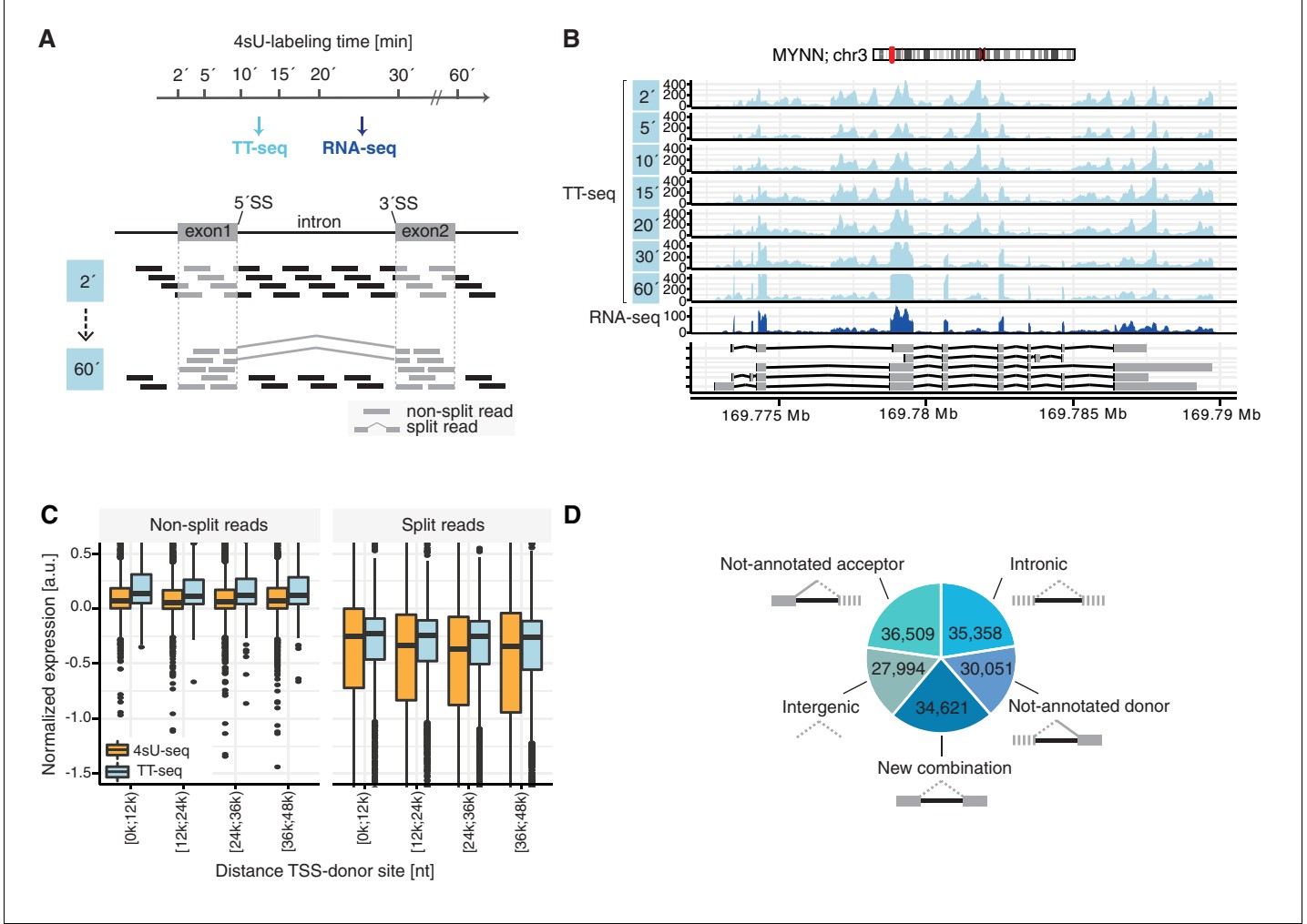

**Figure 1.** TT-seq monitors human RNA splicing. (A) Experimental design. TT-seq and RNA-seq were performed after 2, 5, 10, 15, 20, 30 and 60 min of 4sU-labeling in human K562 cells. The number of reads spanning exon-intron/intron-exon splice-sites (non-split reads) gradually decrease with longer labeling duration, while exonic reads and reads spanning exon-exon junctions (split reads) increase during the time course, converging to levels similar to RNA-seq. (B) Coverage track of MYNN gene for 2, 5, 10, 15, 20, 30 and 60 min of 4sU-labeling followed by TT-seq and 2 min of 4sU-labeling followed by total RNA-seq. (C) Distribution of non-split reads (left) and split-reads (right) in previously published 4sU-seq (orange) and TT-seq (blue) (*Schwalb et al., 2016*) split by quartiles of the transcription start site to donor splice-site distance. Black bars represent the median values for each group. Lower and upper boxes are the first and third quartile, respectively. (D) Fractions of novel splicing events detected in intragenic and intergenic genomic regions. Solid boxes represent known exons, dashed boxes represent novel exons. (D based on *Supplementary file 1*).

DOI: https://doi.org/10.7554/eLife.45056.003

The following figure supplement is available for figure 1:

**Figure supplement 1.** TT-seq is highly reproducible and captures newly synthesized RNAs.

DOI: https://doi.org/10.7554/eLife.45056.004

# Results

## RNA labeling monitors intron removal

To monitor global RNA metabolism in human cells, we performed TT-seq analysis in K562 cells after different times of RNA labeling with 4-thiouracil (4sU) (*Figure 1A*). We previously showed that such a labeling time series can estimate splicing rates in the yeast *S. pombe* (*Eser et al., 2016*). We exposed K562 cells to 500 µM of 4sU for a labeling time of 2, 5, 10, 15, 20, 30, or 60 min, isolated RNA, and conducted both TT-seq and total RNA-seq (*Figure 1A*). On average, we obtained 250 million and 55 million 150-nucleotide (nt) paired-end reads for each of the TT-seq and RNA-seq

samples, respectively. We next mapped TT-seq and RNA-seq data to the human genome (Materials and methods). The experiments were highly reproducible (*Figure 1—figure supplement 1A*). Visual inspection of the mapped reads revealed strong TT-seq signals in transcribed regions covering both introns and exons (*Figure 1B*), whereas RNA-seq data covered mainly exons (*Figure 1B*, *Figure 1—figure supplement 1B*). The relative number of intronic reads in TT-seq data decreased with 4sU-labeling time, whereas the signal for exons increased. These observations were consistent with capture of newly synthesized precursor RNA because spliced introns are more rapidly degraded than exons that are maintained in mature, stable RNA. Thus, our time series data contained information about the kinetics of precursor RNA splicing that we exploited further.

## New and alternative splice sites

We first analyzed our TT-seq data for the occurrence of reads that are informative of precursor RNA splicing, that is reads spanning exon-exon boundaries ('split reads') and reads spanning exon-intron ('donor') and intron-exon ('acceptor') boundaries ('non-split reads'). Using a threshold of at least 10 split reads for an exon-exon boundary, we found 341,855 putative introns (Materials and methods). The relative number of these non-split reads compared to split reads was highest after 2 min of 4sU-labeling and progressively decreased with longer labeling times, eventually converging to a similar coverage as in the RNA-seq samples (*Figure 1—figure supplement 1C*). Coverage with split reads decreased with the distance to the transcription start site (TSS) in 4sU-seq data but not in TT-seq data (*Figure 1C*, data from *Schwalb et al., 2016*), suggesting that the previously reported estimation of splicing rates with 4sU-seq (*Mukherjee et al., 2017*; *Rabani et al., 2014*), overestimated splicing rates for 5' introns compared to 3' introns. With the use of TT-seq we could avoid a 5'-bias in splicing rate estimations.

About one half of the putative introns (177,322) mapped to splice sites that had been annotated in the database of transcribed regions GENCODE (Materials and methods), whereas the other half did not (164,533). Of these putative introns that had not been previously annotated as splice sites in GENCODE, more than 99% represented introns ending with the GT|AG canonical dinucleotides. Moreover, 21% represented new combinations of already annotated donors and acceptors (*Figure 1D*). Furthermore, 18% map to a non-annotated donor site and to a previously annotated acceptor site, 22% to a non-annotated acceptor site and to a previously annotated donor site. Interestingly, another 38% mapped to both non-annotated donor sites and non-annotated acceptor sites. Of these, about one half was located within an annotated GENCODE gene (intronic), whereas the other half was located in regions of the genome not annotated in GENCODE (intergenic). Overall, this analysis indicates that the number of splice sites has previously been underestimated, in agreement with recent studies that integrated very large datasets of the public RNA-seq repository (*Nellore et al., 2016*) or studies that used full-length mRNA sequencing (*Anvar et al., 2018*).

## Kinetic modeling

Defining the kinetics of RNA synthesis, splicing, and degradation from short-read-based protocols is inherently ambiguous due to the many RNA species overlapping any genomic position, including precursor RNAs and multiple splice isoforms. In the future, quantitative and high-throughput full-length transcriptome sequencing may become available to improve the situation; however, co-transcriptional alternative splicing would still cause ambiguities. We have therefore shown it is accurate to analyze the metabolism of phosphodiester bonds rather than RNA species themselves (*Wachutka and Gagneur, 2017*). Following this idea, we modeled the steady-state rates of synthesis and degradation (or equivalently cleavage) of each of three different phosphodiester bonds individually: the exon-intron bond at the donor site, the intron-exon bond at the acceptor site, and the bond between the two joined exons after successful exon ligation to yield product RNA (*Figure 2A* left, Appendix). We refer to these definitions throughout when we use the terms 'splicing kinetics' or 'splice site kinetics'.

We then considered the metabolism of these three phosphodiester bond types at steady-state. Synthesis balances out degradation at steady-state for any molecular species, independently of the kinetics. The steady-state synthesis rate (amount produced per unit of time) and the steady-state degradation rate (ratio of steady state amount by the steady-state synthesis rate) are defined quantities without any assumption on the kinetics (Appendix). Synthesis of the donor and acceptor bonds

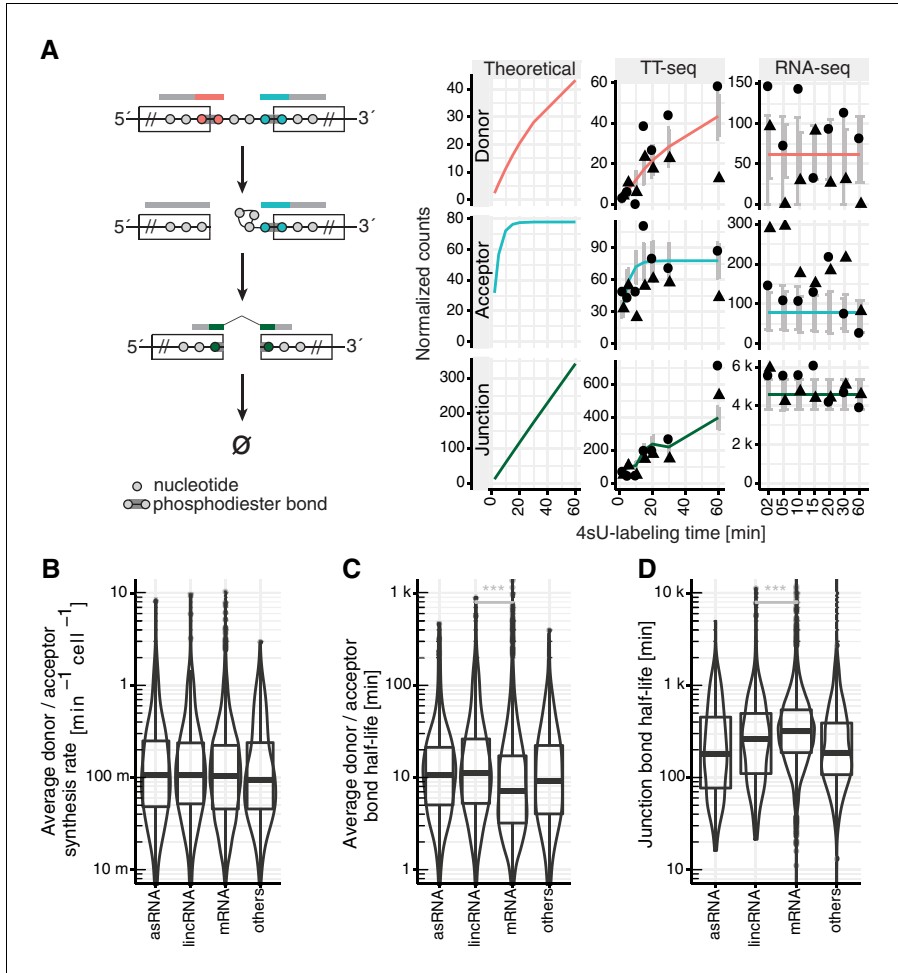

**Figure 2.** Estimation of RNA metabolic rates at individual phosphodiester bonds. (**A**) Definition of kinetic parameters at the level of individual phosphodiester bonds enables independent estimation of the rates of donor (red) and acceptor (blue) bond half-life and of junction (product, green) formation (left). Reads covering the introns were extracted and divided into three classes: reads starting at the upstream exon and extending into the intron (red), reads starting within the intron and extending to the downstream exon (blue) and reads mapped to the upstream and downstream exon but gaping the intron (green). Theoretical analytical curves as well as the curves corrected for cross-contamination for TT-seq and RNA-seq are shown. Sequencing depth-normalized counts are provided for both replicates (circle, triangle). Estimated standard errors are depicted in grey. A dip in expectation value of the junction count was observed at 30 min 4sU-labeling time due to variation of cross-contamination across samples (unlabeled RNA in TT-seq samples), (right). (**B–D**) Violin plots representing the distribution of synthesis rate (**B**), average donor and acceptor bond half-life (**C**) and junction bond half-life (**D**) for mRNAs (n = 8,770), lincRNAs (n = 204), antisense RNAs (asRNA, n = 162) and other ncRNAs (n = 290). Black bars represent the median values for each group. Lower and upper boxes are the first and third quartile, respectively. (**B–D** based on *Supplementary file 3*).

DOI: https://doi.org/10.7554/eLife.45056.005

The following figure supplement is available for figure 2:

**Figure supplement 1.** Details on kinetic modeling of RNA metabolic rates.

DOI: https://doi.org/10.7554/eLife.45056.006

reflects precursor RNA synthesis. Cleavage of the donor bond is caused by either splicing or by precursor RNA degradation. The half-life of those donor bonds that get spliced depends on intronic transcription at least up to the branchpoint and on the first transesterification step. Cleavage of the acceptor bond is caused by either splicing or by precursor RNA degradation. The half-life of those acceptor bonds that get spliced is determined by the first and the second transesterification step

but not by intronic transcription. Synthesis of the junction bond is the outcome of completed splicing. Cleavage of the junction bond indicates RNA degradation (Materials and methods, Appendix).

Our experimental design includes the injection of labeled and unlabeled spike-ins at constant concentrations in all samples, prior to the purification step. These spike-ins allowed for estimating the variations in sequencing depth as well as the overall newly synthesized RNA fraction of every sample (Materials and methods). The unlabeled spike-ins also allowed estimating the amount of cross-contamination, that is of unlabeled RNAs that are purified and which can represent a large fraction of all RNA-seq reads at short labeling durations (Materials and methods). These technical parameters estimated from the spike-ins read counts were then used as covariates to model expected read counts of all three types of bonds in each sample.

## Application and testing of the kinetic model

Using these considerations, we fitted the abundance of each of these three types of bonds with a first-order kinetic model for a total of 162,134 donor, 177,543 acceptor and 156,825 junction bonds that showed at least 100 supporting reads across the full dataset (*Figure 2A* right, Materials and methods, Appendix). Overall, TT-seq read counts agreed with the expected counts of our kinetic model (*Figure 2A* central column, *Figure 2—figure supplement 1A*). The synthesis rates for donors and acceptors, and the product half-life inferred from distinct splice junctions (Materials and methods) agreed well, demonstrating the robustness of our approach (Spearman rank correlation >0.33 for synthesis time downstream of the first exon, $p < 2 \times 10^{-16}$ and Spearman rank correlation >0.76 for half-life, $p < 2 \times 10^{-16}$, variation of 180% fold for synthesis rate, and 32% fold for half-life, Materials and methods, *Figure 2—figure supplement 1B*). Variations were larger for synthesis rates because these estimates are in a first approximation proportional to the coverage in the short-labeled TT-seq samples and are therefore more sensitive to sequencing biases. In contrast, half-lives, which are in a first approximation proportional to the ratio of coverages in short-labeled TT-seq samples and in RNA-seq, better control for sequencing biases.

We furthermore conducted extensive simulations to assess the performance and limitations of the fitting procedure to estimate the rates when the ground truth is known. We simulated counts based on the estimated distributions of synthesis rates, splicing half-times and half-life times based on the experimental data. Based on simulated data, our method leads to unbiased estimates of ground truth synthesis rates (*Appendix 1—figure 12*), splicing half-time and half-life time (*Appendix 1—figure 13*) with high precision compared to the dynamic range. Also, we used simulations to explore how estimation accuracy is affected when using data with much lower read coverage or for extremely slow or fast rates. These simulations showed that lowering the total read coverage cut-off below 100 reads would lead to relative errors typically surpassing 100% (median, *Appendix 1—figure 23*). These simulations also showed that estimations of half-lives shorter or much longer than our labeling durations (shorter than 1 min or longer than 3 days) would lead to median error surpassing 100% (*Appendix 1—figure 24*).

First-order kinetic models are simple models that grossly model the underlying biochemical processes. We also investigated two alternative models that potentially capture more complex kinetics. The first one is a delay differential equation model for donor bond kinetics that modeled the time to transcribe the intron up to the branchpoint with a delay, followed by first-order kinetics for the first transesterification step (Appendix). Simulations indicated that identifying the parameters of this delay differential equation model is difficult (*Appendix 1—figures 2*, *20–22*) because the data do not support distinguishing the contribution of transcription from the one of the first transesterification step. However, fitting a first order kinetic model on data simulated according to the delay differential equation model showed that the estimated donor bond half-life approximately equated the sum of the intronic transcription delay and the half-time of the first transesterification step (*Appendix 1—figure 5*, yet usually underestimating with a median relative level of 0.89). The second alternative model is a coupled differential equation model for the junction bonds that modeled junction formation as the outcome of a first-order kinetics splicing process rather than as a constant. Simulations showed that the data did not allow to easily distinguish this coupled kinetics from first-order kinetics (*Appendix 1—figure 3*). Moreover, the junction bond half-life estimated by the first order kinetics model approximately equated the sum of the splicing half-time and of the half-life of the processed RNA (*Appendix 1—figure 7*, yet usually overestimating with a median relative level of

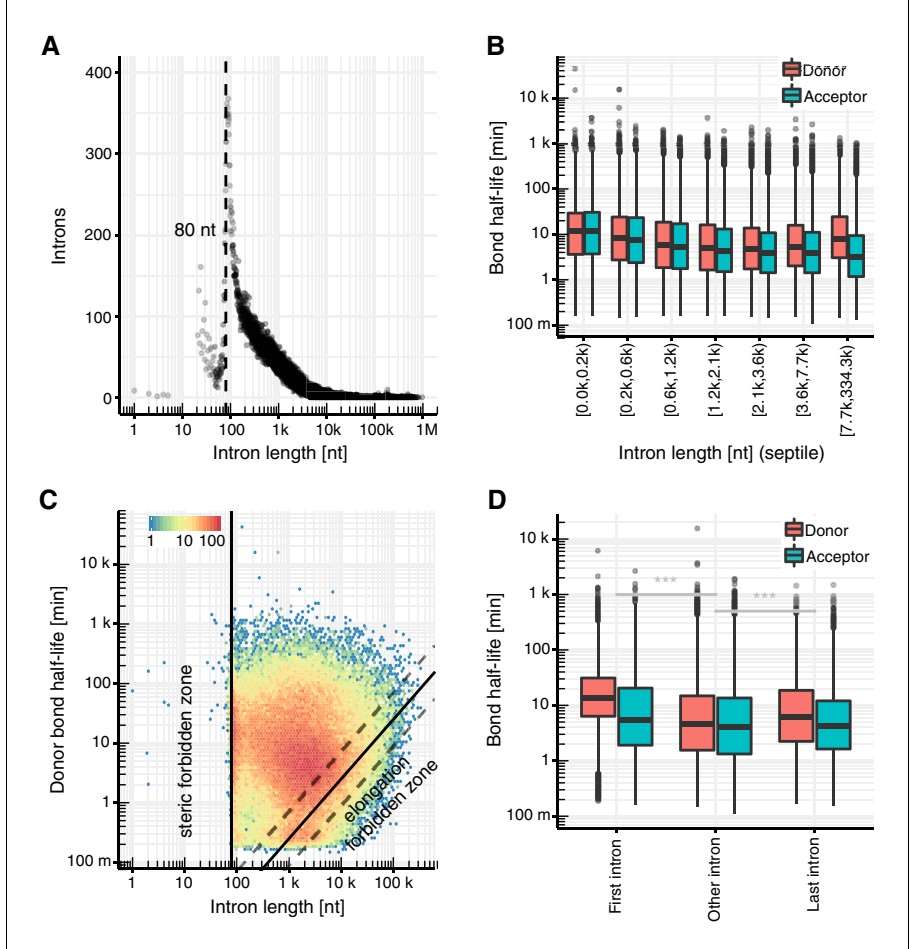

**Figure 3.** Intron length influences splicing kinetics. (**A**) Number of introns against intron length. Dashed line represents the minimum estimated intron length. (**B**) Distribution of donor and acceptor bond half-life for introns split by intron length septiles. Black bars represent the median values for each group. Lower and upper boxes are the first and third quartile, respectively. (**C**) Donor bond half-life against intron length for all observed introns, minimum estimated intron length (black vertical line), and theoretical elongation boundary for a intronic polymerase elongation rate of 4 kb min$^{-1}$ (black lines) and 95% confidence interval estimates (dashed lines, Materials and method). Colors encode data point density. (**D**) Distribution of half-life of donor (red) and acceptor (blue) bonds for first and last introns in major isoforms compared to other introns. (**A–D** based on *Supplementary file 2*).

DOI: https://doi.org/10.7554/eLife.45056.007

The following figure supplement is available for figure 3:

**Figure supplement 1.** Branchpoint-acceptor site distance.

DOI: https://doi.org/10.7554/eLife.45056.008

1.2). Unless specifically mentioned, we used the first order kinetics model in the remaining analyses because of its robustness and its approximate equivalence with alternative models (Appendix).

## Splicing times are in the range of minutes

Based on GENCODE annotation, the analyzed bonds mapped to 8,770 mRNAs, 162 RNAs antisense to protein-coding genes (asRNA), 204 long intergenic non-coding RNAs (lincRNA), and 290 other non-coding RNAs (*Figure 2—figure supplement 1C*). When averaged within major isoforms (Materials and methods), synthesis rates and half-lives of donors and acceptors ranged over two orders of magnitude (42-fold and 48-fold change, 90% equi-tailed range), whereas the junction bond half-life spanned only slightly over one order of magnitude (8.1-fold 90% equi-tailed range, *Figure 2B–D*). These major isoform aggregated rates generally agreed with previously reported splicing rates and

RNA half-lives (*Figure 2—figure supplement 1D*; *Mukherjee et al., 2017*). Moreover, mRNAs were spliced significantly faster (median of 7.2 min) than lincRNAs (median of 11 min, p<2 $\times$ 10$^{-16}$) and other non-coding RNAs (*Figure 2C*). Also, mRNAs had junction bonds with the longest half-lives (median of 316 min) (*Figure 2D*), consistent with previous studies (*Mukherjee et al., 2017*; *Schlackow et al., 2017*; *Schwalb et al., 2016*). Similar conclusions can be reached using site-specific rates (*Figure 2—figure supplement 1E–J*). The obtained apparent splicing times in the range of minutes agree with many previous estimates that were obtained using different methods, but argue against fast splicing, within less than a minute, that was suggested by some studies (*Carmo-Fonseca and Kirchhausen, 2014*).

## Intron length constrains splicing times

Intron length has been suggested to affect splicing kinetics (*Hicks et al., 2010*; *Khodor et al., 2011*; *Pai et al., 2017*; *Proudfoot, 2003*; *Windhager et al., 2012*), and we therefore investigated this further. First, our de novo annotation of introns is in agreement with a minimal intron length of about 80 nt (*Figure 3A*), as expected from the spatial needs within the spliceosome (*Ruskin et al., 1985*; *Wieringa et al., 1984*). Second, we find that among introns shorter than 2,000 nt, acceptor and donor bond half-life showed similar distributions and decreased with increasing intron length (*Figure 3B*). The reasons for why short introns are spliced more slowly than long ones remain to be investigated. It is possible that for longer introns the splice site definition by the following exon facilitates splicing and that there are less restraints for splicing for longer introns. This observation also strongly argues for pre-ordering of the spliceosome on the transcribing polymerase.

Our analysis also reveals that donor and acceptor bond half-lives differ for long introns. Among introns longer than 2,000 nt, acceptor bond half-life plateaued at a median value of about 4 min, whereas donor bond half-life increased with intron length up to a median value of about 8 min for introns larger than 7,700 nt (last septile). A possible explanation for this significant difference is that donor sites of long introns are transcribed much earlier than acceptor sites and splicing can only start when the intron is transcribed. Indeed, the donor bond half-life is determined by the elongation time needed to transcribe at least the branchpoint and by the first transesterification step, whereas the acceptor bond half-life is determined by both the time for the first transesterification step and for the second transesterification step to be completed. Assuming a maximum polymerase elongation velocity of 4 kb/min (*Fuchs et al., 2014*; *Gressel et al., 2017*; *Jonkers and Lis, 2015*; *Saponaro et al., 2014*; *Veloso et al., 2014*), we observed very few introns violating this predicted limit (*Figure 3C*). This limit for donor bond half-life affects only a small proportion of all introns (last septile) so that, overall, there is no positive correlation between donor bond half-life and intron length (*Figure 3C*). For shorter introns, the donor bond half-life and the acceptor bond half-life were similar (*Figure 3B*), indicating that the second transesterification step is fast compared to the overall splicing kinetics.

Another prediction of this model is that for slowly transcribed introns the donors should take longer to cleave than the acceptors. Consistent with this hypothesis, the median half-life was 2.5-fold (p<2$\times$10$^{-16}$) longer for donor bonds than for acceptor bonds of first introns (*Figure 3D*), which are known to be more slowly transcribed (*Danko et al., 2013*; *Fuchs et al., 2014*; *Jonkers and Lis, 2015*; *Saponaro et al., 2014*; *Veloso et al., 2014*). A small significant difference was also found for the last intron (1.4-fold, p<2$\times$10$^{-16}$), which could reflect slower polymerases near the transcript end or different kinetics of splicing of the last intron (*Davis-Turak et al., 2015*; *Rigo and Martinson, 2008*). In conclusion, these data show that donor half-life and thus the beginning of splicing is limited by transcription elongation for long introns. Taken together, our results are generally consistent with the co-transcriptional nature of splicing and reveal that the length of the intron influences splicing kinetics in at least two different ways.

## Several snRNA interactions are related to donor cleavage kinetics

Whereas overall trends in splicing kinetics can be explained by global features such as intron length and polymerase elongation velocity, the kinetics of splicing also critically depend on the RNA sequence context around the donor, acceptor, and branchpoint. To gain insights into the sequence determinants for splicing, we built a linear model (Materials and methods) that allowed us to estimate changes in donor bond half-life as a function of single nucleotide changes relative to the

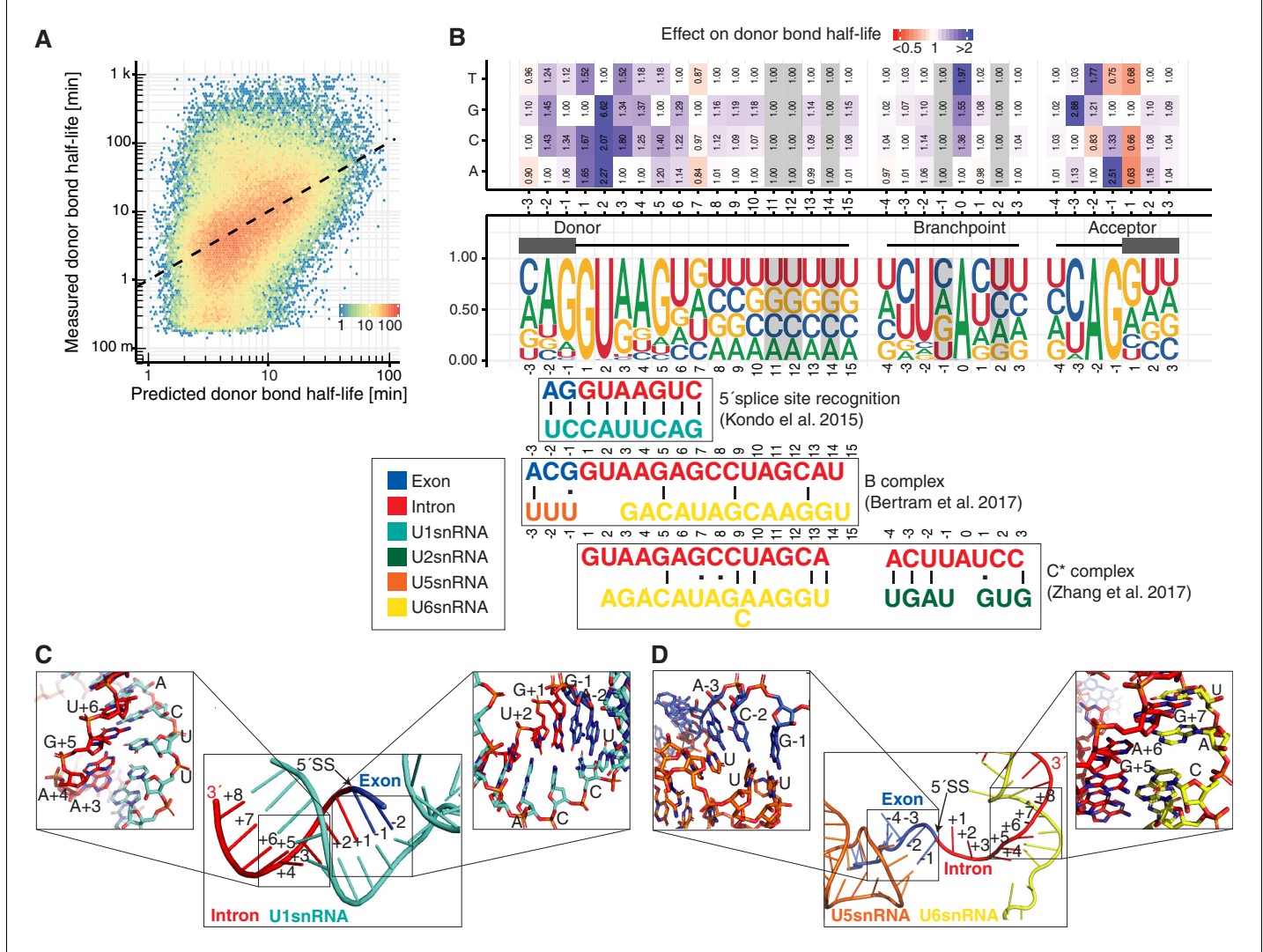

**Figure 4.** Sequence and structural contributions to donor bond half-life (related to the first catalytic step). (**A**) Measured donor bond half-life against single nucleotide model prediction (median relative error of 150%, Spearman's rank correlation ρ = 0.45, p<10$^{-16}$). (**B**) Nucleotide frequency (middle track) and prediction of the relative effect on donor bond half-life (upper track) for single nucleotide deviation from consensus sequence around donor, acceptor splice-site and branchpoint. Grey color marks positions predicted to have no effect (Materials and methods). RNA map of base-pairing interactions between four snRNAs (U1, U2, U5, U6) and precursor RNA sequences used in three published spliceosome structures of U1 snRNP binding to 5′splice-site, B and C* complex (bottom track) (exon (blue), intron (red), U1 snRNA (light blue), U2 snRNA (green), U5 snRNA (orange), U6 snRNA (yellow)). Canonical and non-canonical base-pairing interactions are depicted by black solid lines and black dots, respectively. (**B** based on *Supplementary file 4*). (**C**) Structure of U1 snRNA interactions with precursor RNA 5′splice-site (*Kondo et al., 2015*). (**D**) Structure of U5 and U6 snRNA interactions with precursor RNA in spliceosome B complex (*Bertram et al., 2017a*).

DOI: https://doi.org/10.7554/eLife.45056.009

The following figure supplement is available for figure 4:

**Figure supplement 1.** Effects of nucleotide changes on splicing kinetics as the spliceosome proceeds.
DOI: https://doi.org/10.7554/eLife.45056.010

consensus sequence. The single nucleotide model could explain 19% of the observed variance in log-transformed donor bond half-life and achieved a median relative error for individual sites of 150%, which is small compared to the dynamic range across sites spanning two orders of magnitude (*Figure 4A*). This analysis showed that nucleotide deviations from the consensus splice-site increase donor bond half-life. These findings are consistent with evolutionary pressure for donor sequences optimized for fast splicing.

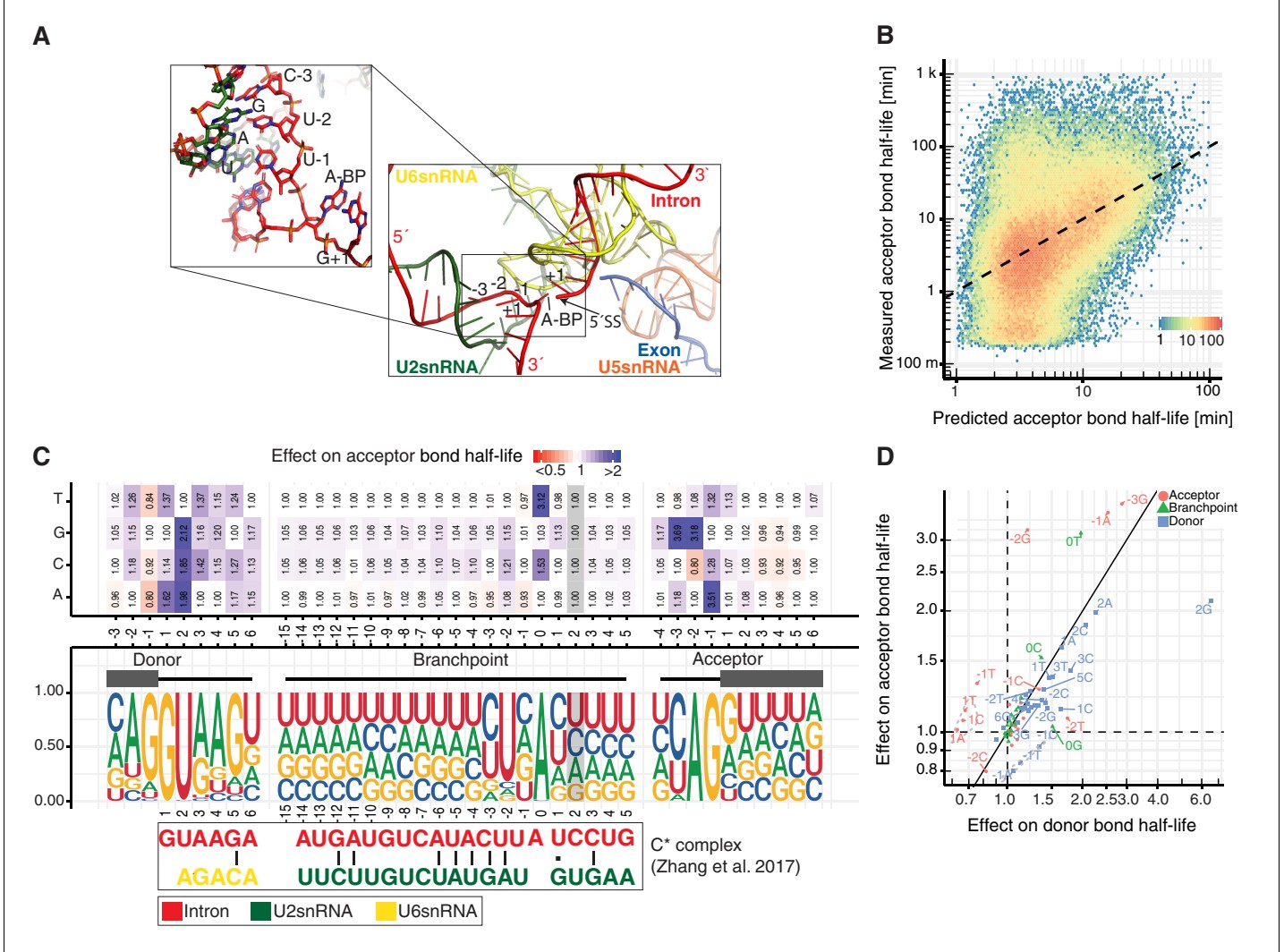

**Figure 5.** Sequence and structural contributions to acceptor bond half-life (related to the first and second catalytic steps). (A) Structure of U2, U5 and U6 snRNAs interactions with precursor RNA in C* spliceosome complex (*Zhang et al., 2017*). (B) Measured acceptor bond half-life against single nucleotide model prediction (median relative error of 150%, Spearman's rank correlation ρ = 0.45, p<10$^{-16}$). (C) Nucleotide frequency (middle track) and prediction of the relative effect on acceptor bond half-life (upper track) for single nucleotide deviation from consensus sequence in sequence around donor, acceptor splice-site and branchpoint. Grey color marks positions predicted to have no effect (Materials and methods). RNA map of base-pairing interactions between two snRNAs (U2 and U6) and precursor RNA sequences used in the published spliceosome structures of C* complex (intron (red), U2 snRNA (green), U6 snRNA (yellow)). Canonical and non-canonical base-pairing interactions are depicted by black solid lines and black dots, respectively. (D) Comparison of the effects of single nucleotide changes on donor bond half-life against effects on acceptor bond half-life. Donor (blue), acceptor (red), branchpoint (green) single nucleotide effects are shown. Positions one nt before the donor site and one nt after the acceptor site are encircled in grey. (C, D based on *Supplementary file 4*).

DOI: https://doi.org/10.7554/eLife.45056.011

In order to elucidate the contribution a single nucleotide change might have on interactions within the spliceosome, we compared our predicted single nucleotide effects with base interactions observed in three different spliceosome structures (*Figure 4B*, bottom). Recognition of the donor site by U1 snRNP plays a crucial role during early spliceosome assembly. RNA-RNA interaction between precursor RNA and U1 snRNA are mainly stabilized through Watson-Crick interactions (*Figure 4C*; *Kondo et al., 2015*). In our model, substitution of the highly frequent G at +1 or −1 from the donor site with a C resulted in an increase in bond half-life (*Figure 4B*), likely because C cannot form a stable interaction with a C in the U1 snRNA, in agreement with previous in vitro studies (*Kondo et al., 2015*). In contrast, at position +3 from the donor site, a change from A to G has

only a minor effect on the bond half-life (*Figure 4B*), likely because G can still form a non-canonical base pair with U in the U1 snRNA (*Kondo et al., 2015*). These results suggest that interactions of the precursor RNA donor region with U1 snRNA contribute to the observed donor bond half-lives.

After donor site recognition by U1 snRNP, the Prp28 RNA helicase mediates the exchange of U1 with U6 snRNP and the U4/U5/U6 tri-snRNP can stably bind the precursor RNA. In the resulting B complex, the U5 stem loop interacts with the three terminal nucleotides of the 5′-exon, whereas the U6 ACAGA helix is formed near the donor site (*Figure 4D*; *Bertram et al., 2017a*). Our results suggest that U5 interactions may contribute to the kinetics of donor cleavage. For example, an A in the position −3 relative to the donor site leads to faster donor bond half-life supposedly because this enables base-pairing with U5 snRNA in the extended precursor RNA-U5 snRNA duplex.

Completion of step-one results in the C complex that is then converted to the activated C* complex, which contains the two exon ends in close proximity for the step two reaction. RNA duplexes are formed between the intron region close to the donor site and U6 snRNA and between the branch site region and U2 snRNA (*Figure 5A*) (*Zhang et al., 2017*). We also found that interactions in the C* complex were predictive of donor bond half-life (*Figure 4D*). In particular, changes in the branchpoint adenine and at all positions −4 to +3 of the branchpoint show kinetic effects, except for the positions −1 and +2 that are predicted to not contribute to donor bond half-life (grey highlighting in *Figure 4D*). In agreement with the structural data, these nucleotides are also the only two nucleotides in the vicinity of the branchpoint that do not interact with U2 in the C* complex (*Zhang et al., 2017*). When compared to each other, the precursor RNA nucleotides interacting with snRNAs during 5' splice site recognition showed the strongest effects on donor bond half-life, followed by nucleotides interacting in the B complex, and to a lesser extent the nucleotides interacting in the C* complex (*Figure 4—figure supplement 1A*). Positions with no predicted contact in these structures showed least effects (*Figure 4—figure supplement 1A*). These observations support our kinetic modeling, but also argue that the structurally characterized spliceosomal complexes represent functional states.

Taken together, variation in the in vivo kinetics for the donor cleavage can in part be rationalized with early interactions of precursor RNA with U1 snRNA, and with later U5 snRNA interactions observed in structures of the B and Bact complexes. Moreover, the stability of the C* complex appears to also affect donor bond half-life, possibly because it prevents the reverse reaction of donor site cleavage (*Tseng and Cheng, 2008*). Since several precursor RNA positions are involved in different types of interactions in different splicing intermediates, the observed overall kinetics of donor cleavage reflect a combination of distinct microscopic rates, which cannot be distinguished by our in vivo approach. Furthermore, not all observed effects of nucleotide changes could be explained with available structures. For example, the first nucleotide of the downstream exon (acceptor +1 position) was important for donor cleavage kinetics. Although it remains unclear why, this could be related with co-transcriptional recruitment and recycling of splicing factors, maybe favored by Pol II 3′splice-site pausing, similar to that suggested in *Aitken et al. (2011)*.

## Sequence and structural contributions to acceptor bond half-life

We also built a regression model predicting log-transformed acceptor bond half-life from sequence (*Figure 5B*, 20% of variance explained, median relative error of 150%). Single nucleotide changes around the acceptor site generally had larger effects on acceptor bond half-life, reflecting effects on step two kinetics, whereas changes around the donor site had greater effects on donor bond half-life, reflecting effects on step one kinetics (*Figure 5C*). The post-catalytic complex (P complex), which is specific to step two splicing reaction, is not yet structurally characterized in human. Nevertheless, nucleotide changes that influence base pair interactions reported for the P-complex in *S. cerevisiae* (*Bai et al., 2017*) showed stronger effects on the acceptor bond half-life than for the donor bond half-life (*Figure 4—figure supplement 1A*). Nucleotide positions in the precursor RNA that are not involved in base pair interaction with snRNAs in B-type and C* complex structures were irrelevant for predicting acceptor bond half-life (grey highlighting in *Figures 4B* and *5C*, feature selection, Materials and methods).

Most nucleotides showed similar effects in the donor and acceptor bond half-life models but some noticeable differences were observed between them (*Figure 5D*). Our results indicate that a non-canonical G branchpoint does not affect acceptor bond half-life but increases donor bond half-life. We also observed that the predominant G at the donor −1 nucleotide leads to fast donor

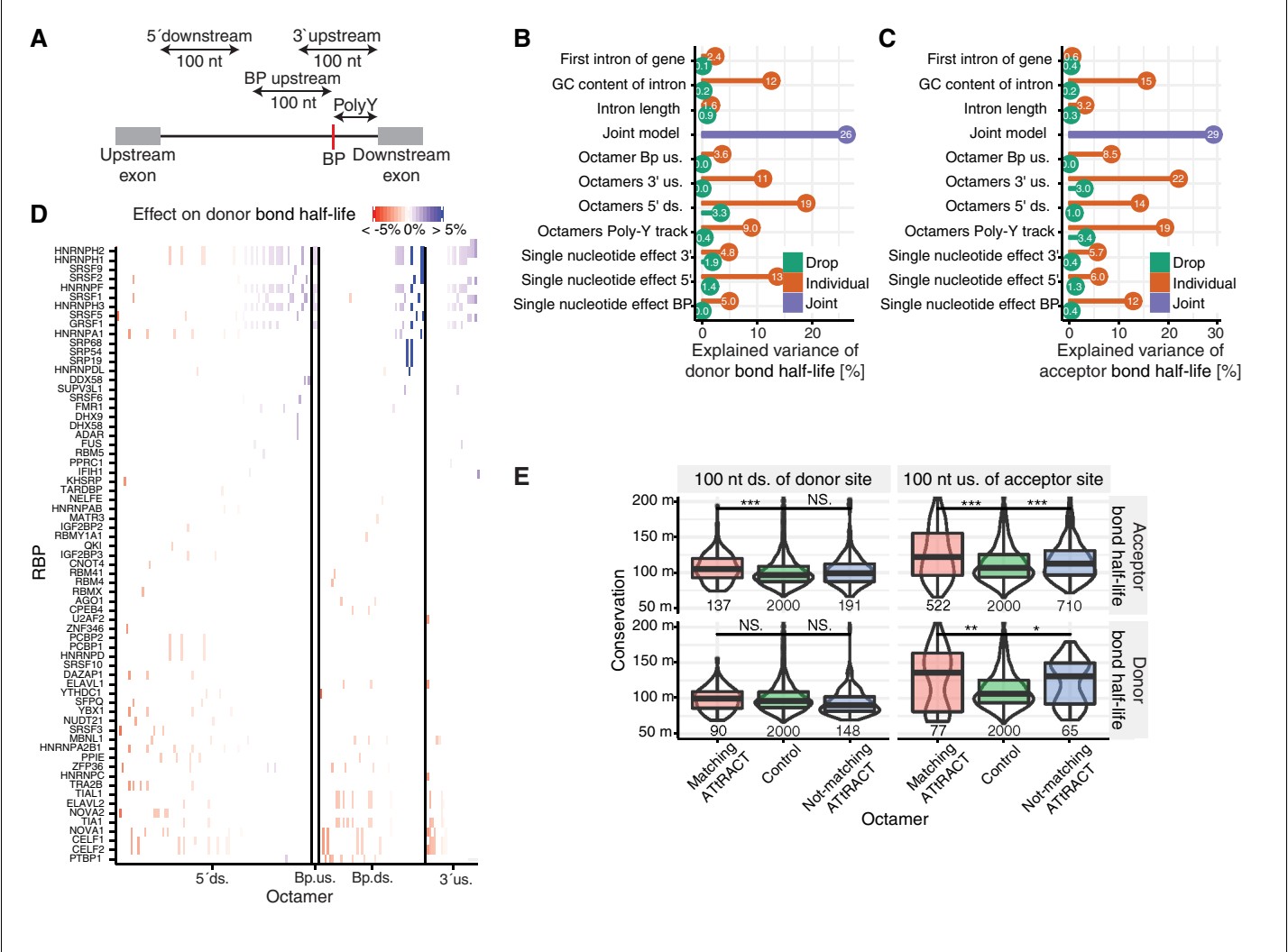

**Figure 6.** Precursor RNA sequences contributions to splicing kinetics. (**A**) Schematic representation of the regions used for the octamer search (BP = branchpoint, polyY = polyY track). In case of short intron, the regions 5' downstream, 3' upstream, and BP upstream were cropped to not extend into the exonic regions. Poly-Y track was defined as the region from the branchpoint up to the acceptor site. (**B–C**) Proportion of variance explained by the joint model (purple) for log-transformed donor (**B**) or acceptor (**C**) bond half-lives, as well as proportion of variance explained by individual features (orange) and drop of the proportion of variance explained when individual features are removed from the joint model (green). 5'ds = 5'downstream, 3' us = 3'upstream. (**D**) Effect on donor bond half-life of octamers matching to RNA-binding proteins (RBP, rows) motifs identified using the ATtRACT database. Each column represents one octamer; the color depicts strength and direction of the effect. (**E**) Distribution of the phylogenetic conservation score (PhastCons 100-way) of random octamers (green), significant octamers matching (red) and not matching (blue) the ATtRACT database of RNA-binding motifs, estimated by region (column) and model (row). Black bars represent the median values for each group. Lower and upper boxes are the first and third quartile, respectively. Stars above boxes depict pairwise significance levels by Wilcoxon signed rank test. (**D** based on *Supplementary file 5*).

DOI: https://doi.org/10.7554/eLife.45056.012

The following figure supplement is available for figure 6:

**Figure supplement 1.** Predicted octamers show significant regulatory effects on donor and acceptor bond half-life.

DOI: https://doi.org/10.7554/eLife.45056.013

cleavage kinetics but to slow acceptor cleavage kinetics, maybe because this interferes with positioning of the neighboring +1 donor nucleotide that serves as a nucleophile during step two. Despite this disadvantage in acceptor cleavage kinetics, the donor −1 position is predominantly G, presumably because this improves donor site recognition by base pairing with a C in U1 snRNA as described above. Taken together, available structural information on the spliceosome help to

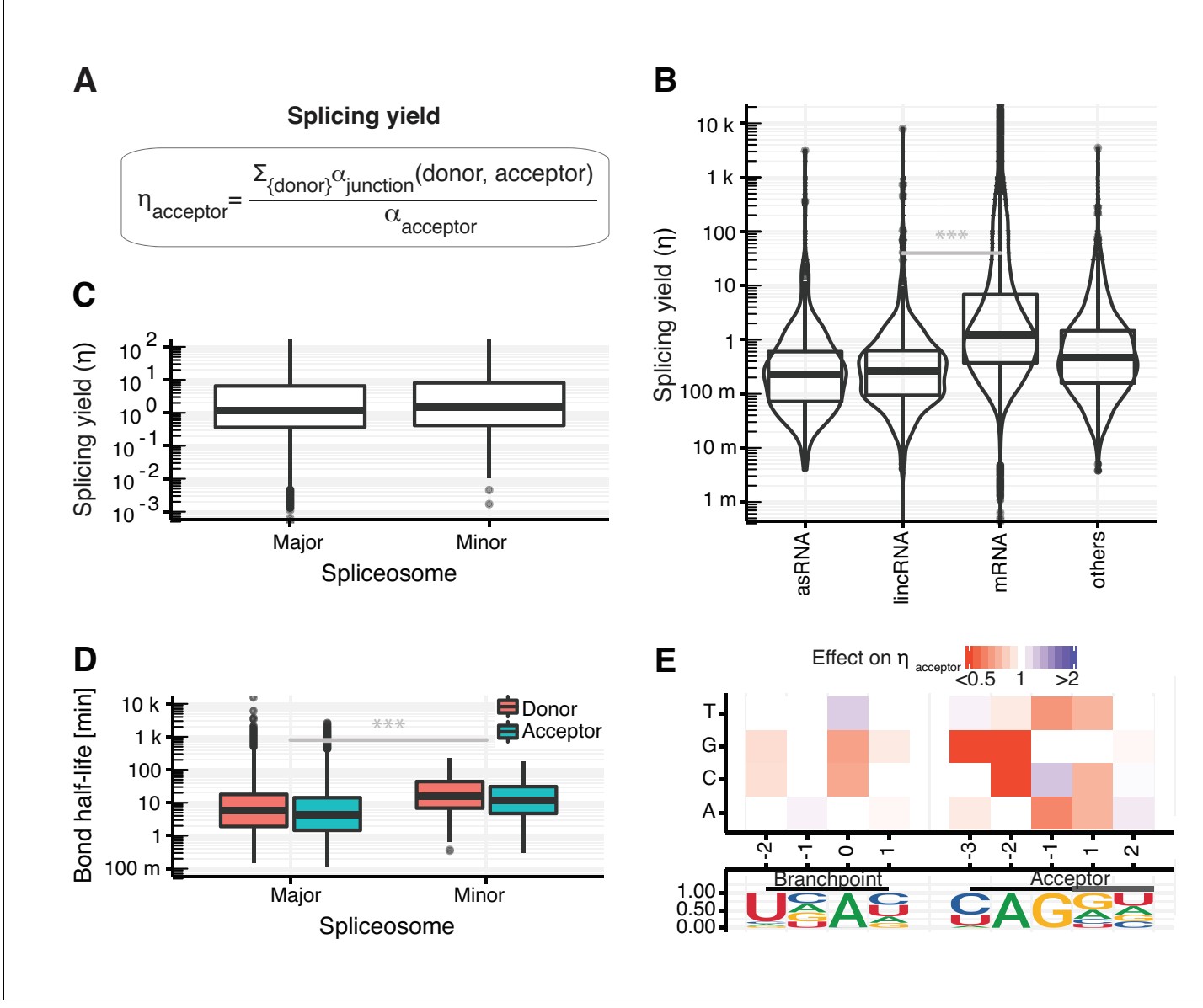

**Figure 7.** Splicing yield differs between RNA classes. (A) Definition of synthesis rate at individual phosphodiester bonds enables the estimation of splicing yield for acceptor splice sites ($\eta_{acceptor}$). (B) Distribution of splicing yield of acceptor splice sites for introns of mRNAs (n=64,555), lincRNAs (n=398) antisense RNAs (asRNA, n=265), and other ncRNAs (n=876) of major transcripts (Materials and methods) for GENCODE genes. Black bars represent the median values for each group. Lower and upper boxes are the first and third quartile, respectively. (C–D) Distributions of the observed acceptor spicing yield (C) and donor and acceptor bond half-life (D) of splice-sites split by major (U2-type) and minor (U12-type) spliceosome. (E). Single nucleotide effects for branchpoint and acceptor splice site on $\eta_{acceptor}$. Color depicts effect on yield relative to consensus nucleotide (Materials and methods). Relative frequency of the nucleotides is shown for all modeled introns. (B–E based on *Supplementary file 2*).

DOI: https://doi.org/10.7554/eLife.45056.014

The following figure supplement is available for figure 7:

**Figure supplement 1.** Splicing yield is affected by intron length.

DOI: https://doi.org/10.7554/eLife.45056.015

rationalize some of the effects of base changes around splice sites. Even though the contributions of several mechanistic processes to the observed kinetics cannot be disentangled, our results reveal which nucleotide positions around splice sites are critical for fast splicing kinetics.

## Regulatory precursor RNA motifs contribute to splicing kinetics

Splicing is modulated by auxiliary factors, including serine/arginine-rich proteins and hnRNPs (heterogeneous nuclear ribonucleoproteins), that bind to regulatory motifs around the splice sites (*Fu and Ares, 2014*; *Matlin et al., 2005*; *Wang and Burge, 2008*). We therefore aimed to identify putative regulatory motifs and to quantify their contribution to splicing kinetics. We derived two extended models for donor and acceptor bond half-life including the single nucleotide effects in the core regions investigated so far and all 65,536 possible RNA octamers in four extended intronic regions, 100 nt downstream of the donor site, 100 nt upstream of the acceptor site, and 100 nt upstream of the branchpoint and the region between branchpoint and acceptor site (*Figure 6A*, Materials and methods). We also included intron length and GC content, but did not include exonic regions, because this would require isoform annotations. Because of the very large number of octamers, we used a feature selection method (Lasso regression), which yielded 551 octamers jointly predicting donor bond half-life and 2,319 octamers jointly predicting acceptor bond half-life (Materials and methods).

Compared to the single nucleotide models, the extended models substantially increased the proportion of variance explained from 19% to 26% for the log-transformed donor bond half-life and from 20% to 29% for the log-transformed acceptor bond half-life (*Figure 6B and C*). These proportions of variance increased when we restricted the analysis to junctions of major isoforms, showing that the results are not over-estimated due to double counting of donors and acceptors belonging to multiple exon junctions (donor bond half-life model by 1.3% and the acceptor bond half-life model by 1.4%). Cumulatively, the proportions of variance explained of these non-overlapping regions largely exceed the proportion of variance explained by the joint model, indicating widespread co-occurrence of splicing-regulatory sequences across introns.

The improved prediction of the extended models over the single nucleotide models is mostly attributable to the octamers of the extended intronic regions. The largest number of predictive octamers identified by the donor bond half-life model was found in the 5' donor site region (*Figure 6—figure supplement 1A*). This set of octamers was the most predictive feature for donor bond half-life individually (19% of the variance) and the feature with the largest impact on variance when dropped from the joint model (3% of the variance). In the acceptor bond half-life model, the regions flanking the branchpoint and acceptor site contained most predictive octamers (*Figure 6—figure supplement 1A*) and associated with largest proportion of variance explained (*Figure 6C*). Moreover, the effects of octamers on bond half-life were of similar order of magnitude than the effects of single nucleotides in donor, acceptor and branchpoint sites for both categories (*Figure 6—figure supplement 1B*, median effect for octamer 1.4%, median effect for nucleotide 5.4%). The drop of proportion of variance explained when a feature was removed from the joint models were small (between 0.0 and 3.4, *Figure 6B and C*) indicating of substantial correlation between the features. These correlations could be technical in the case of overlapping regions, or the result of co-evolution. Altogether, these results show that the octamers in the extended intronic regions contribute to splicing kinetics.

To identify putative regulatory factors that could bind to the predicted RNA octamers, we scored the octamers binding affinities to the 159 human RNA-binding proteins of the ATtRACT database (*Giudice et al., 2016*). We found 258 octamers predicted by the donor bond half-life model (47% versus 42% of non-selected octamers, p=0.017, Fisher test) associating with 69 RNA-binding proteins and 1,039 octamers identified by the acceptor bond half-life model (45% versus 42% of non-selected octamers, p=0.007, Fisher test) associating with 99 RNA-binding proteins motifs (*Figure 6D*, *Figure 6—figure supplement 1C*, relative position weight matrix score >0.9 and selecting for the 5% highest absolute scores, Materials and methods). Our results suggest that several serine/arginine-rich and hnRNP proteins (*Supplementary file 5*) regulate donor and acceptor bond half-life in both positive and negative fashions, depending on the location of their binding site. Octamers associated with the binding site of the polypyrimidine tract-binding protein Ptpb1 are predictive of short donor bond half-life when present between branchpoint and acceptor site but they prolong the donor bond half-life when located near the donor site (*Figure 6D*). The remaining octamers may reflect cis-regulatory elements bound by splicing factors that remain to be characterized. To address the evolutionary conservation of the identified octamers, we aligned them to conserved sequences across 99 mammalian and other vertebrate genomes. Except for octamers

predicted to affect donor bond half-life in the region 100 nt downstream of the donor site, the remaining ones show significantly higher phylogenetic conservation compared to a negative control of random octamers (*Figure 6E*), providing evidence of their biological significance.

## Splicing yield differs between RNA classes

Cleavage of the phosphodiester bonds at the donor and acceptor sites can lead to ligation of the two exon ends, thus completing splicing. However, cleavage of these bonds may also be non-productive in the sense that exon ligation can fail and RNA may be degraded after cleavage. To account for this, we defined the 'splicing yield' as the proportion of precursor RNA successfully converted into spliced RNA (*Figure 7A*). A splicing yield of 1 means that all precursor RNAs that are synthesized are also successfully spliced, whereas a splicing yield less than one means that only a fraction of the precursor RNA is converted to spliced product. We estimated the splicing yield using the junction bonds modeled with the coupled model and the acceptor bonds modeled with first-order kinetics, because alternative kinetic models or using the donor bonds led to systematic biases (Appendix). Hence, we did not computationally constrain our splicing yield estimates to be bounded by 1. Due to estimation errors of the synthesis rates, yields sometimes turn out to be greater than 1.

We found that the splicing yield across sites was much higher for mRNAs (median = 1.2, *Figure 7B*) than for antisense RNAs (median = 0.2), lincRNAs (median = 0.3), and other non-coding RNAs (median = 0.5), suggesting that degradation pathways are competing with splicing more intensively for non-coding RNAs than for coding RNAs. Moreover, the higher yield of mRNAs compared to lincRNAs also held when stratifying by cumulative read coverage across all samples and by half-life (*Appendix 1—figures 26* and *27*), two possible confounders associating with synthesis rate estimation biases in simulations (Appendix). Furthermore, splicing yield was the same for the 139,344 (99.9%) introns harboring the canonical terminal dinucleotides GU and AG recognized by the major spliceosome (U2-type) than for the 182 (0.1%) introns harboring the terminal dinucleotides AU and AC recognized by the minor spliceosome (U12-type) (*Figure 7C*). Although introns targeted by the minor spliceosome showed two-fold slower donor and acceptor bond half-lives compared to those targeted by the major spliceosome (*Figure 7D*, *Figure 7—figure supplement 1B*), the minor spliceosome nonetheless reached similar splicing yields.

To analyze how the nature of the splice sites contributes to splicing yield, we built a model that allow us to predict splicing yield based on sequence (*Figure 7E*). Similar to the effects on bond half-lives (*Figure 4B*, *Figure 5C*), deviations from the consensus sequence led to lower splicing yield. Furthermore, nucleotides near a splice site showed stronger effects than more distant ones, suggesting that the early recognition of donor and acceptor splice sites is a determinant for splicing yield. Taken together, these results indicate that rate and yield are distinct aspects of splicing that may have evolved independently and that the sequence around splice sites determines both the rate and the yield of splicing.

## Discussion

RNA splicing is an essential step of eukaryotic gene expression, but the in vivo kinetics of this two-step process and its dependence on transcription remain poorly understood. Here, we have coupled a metabolic RNA labeling time series to TT-seq analysis of new and total RNA to investigate RNA metabolism in human cells. We have then used kinetic modeling based on a definition of RNA metabolic rates at the level of individual phosphodiester bonds to provide rate estimations for cleavage of phosphodiester bonds at donor and acceptor splice sites in human introns. The obtained splice site cleavage rates, expressed as donor and accept bond half-lives, are free of ambiguities introduced by other methods and are related to the independent contributions of the two splicing steps in vivo.

The donor and acceptor bond half-lives were found to be generally in the range of minutes, although we cannot exclude that we are missing a small population of quickly spliced introns. The donor and acceptor bond half-lives were found to depend on intron length, on the nucleotide sequence surrounding splicing sites, including the branchpoint, and on flanking octamer sequences that may bind regulatory factors. This is consistent with a complex relationship between the splicing machinery and its nuclear environment, in which splicing rates can be influenced not only by RNA sequence but also by gene structure and chromatin landscape (*Davis-Turak et al., 2018*; *Davis-*

*Turak et al., 2015*). In addition, we define the yield of successful splicing and show that it differs dramatically between different RNA classes.

Our results also provided insights into the nature and evolution of co-transcriptional splicing. Previous studies of splicing kinetics in mouse (*Rabani et al., 2014*) and fission yeast *S. pombe* (*Eser et al., 2016*) all found that splicing is faster for shorter introns. A recently published study in *Drosophila* (*Pai et al., 2017*) demonstrated that splicing is faster for intron-defined, short introns (60–70 nt), whereas for exon-defined introns, splicing was faster for introns longer than 2,944 nt, suggesting a complex relationship between splicing kinetics and intron length. We find here that exon-defined splicing in human cells is fastest for introns with a length of around 2,000 nt, whereas short introns (<140 nt) on average take about twice as long to be spliced. High spatial and temporal resolution kinetics data coupled with focused analyses of these very short introns remain to be performed for other species to understand how universal these observations are.

Another observation we made was that longer human introns (>10,000 nt) show increased donor bond half-life, apparently because donor cleavage requires RNA polymerase II to first transcribe the intron. This effect of transcription-limited splicing is observed at ~10% of human precursor RNA introns, whereas most human introns are short enough so that their splicing kinetics are not limited by transcription. How the polymerase elongation rate depends on the nature of the intron and how this influences splicing remains to be investigated. The observed relationship between transcription and splicing of coding RNAs extends to non-coding RNAs. We found that spliced coding RNAs and spliced non-coding RNAs showed similar RNA synthesis rates (*Figure 2C*), whereas previous studies reported considerably slower synthesis rates for mainly unspliced, non-coding RNAs (*Mukherjee et al., 2017*).

Our definition of splice site-specific RNA cleavage rates also allowed for a comparison of kinetic information in vivo with detailed structural knowledge of the spliceosome in different functional states obtained in vitro. Structure-based interpretation of our nucleotide-resolution kinetic results highlights the importance of interactions between snRNAs in the spliceosome and the precursor RNA substrate and provides guidance for interpreting interactions in structures of the spliceosome yet to be obtained. Our results suggest that the predicted interactions between the precursor RNA and snRNAs in the P complex, which have presently only been obtained in yeast (*Bai et al., 2017*; *Wilkinson et al., 2017*) may be similar in human. We show that different types of RNA-RNA interactions observed at various stages of the process are related to the splice site cleavage time. In particular, RNA-RNA interactions must be of high enough affinity to allow for sufficiently specific recognition of splice sites, yet the affinities must be in a range that also allows for rapid conversion between subsequent states, which can show strongly altered RNA-RNA interactions. However, many processes and factors contribute to the observed apparent splicing rates, and these must be disentangled in the future.

Taken together, we have analyzed the metabolism of individual donor and acceptor splice sites in vivo and provided quantitative models for how RNA splicing kinetics may be encoded in the human genome. As we looked at a single growth condition, our models are derived from comparisons across genes and essentially reflect the affinity of precursor RNAs to the core splicing machinery. However, the experimental and computational methodology presented here could be applied to different cell types or under dynamic responses to reveal and quantify the role of splicing regulatory factors and of their related binding sites. Another interesting future direction is the modeling of alternative splicing, which is understood to be the outcome of competitions of alternative donor or acceptor sites with various strengths. Our distinct models of donor and acceptor site kinetics may help to build up such quantitative competition models. Eventually, quantifying the contribution of individual bases to splicing rates, backed by structural and functional studies, may explain the numerous contributions of splicing to the genetics of rare (*López-Bigas et al., 2005*) and common (*Li et al., 2016*) diseases.

## Materials and methods

**Key resources table**

*Continued*

| Reagent type (species) or resource | Designation | Source or reference | Identifiers | Additional information |
|---|---|---|---|---|
| Reagent type (species) or resource | Designation | Source or reference | Identifiers | Additional information |
| Cell line (*Homo sapiens*; female) | K562 chronic myeloid leukemia in blast crisis | DSMZ | DSMZ Cat# ACC-10, RRID:CVCL_0004 | |
| Commercial assay or kit | Plasmo Test Mycoplasma Detection Kit | InvivoGen, San Diego, CA USA | rep-pt1 | |
| Commercial assay or kit | Ovation Universal RNA-Seq System | NuGEN, Leek, The Netherland | 0343–32 | |
| Chemical compound, drug | 4-thiouracil | Carbosynth, UK | NT06186 | CAS 13957-31-8 |
| Software | STAR | https://github.com/alexdobin/STAR | RRID:SCR_015899 | |
| Software | Picard | http://broadinstitute.github.io/picard/ | RRID:SCR_006525 | |
| Software | Salmon | https://combine-lab.github.io/salmon/ | RRID:SCR_017036 | |
| Software | glmnet | https://cran.r-project.org/web/packages/glmnet/index.html | RRID:SCR_015505 | |
| Software | PyMOL | https://pymol.org/2/ | RRID:SCR_000305 | |
| Software | LaBranchoR | https://kipoi.org/models/labranchor | | |
| Software | CleTimer | https://kipoi.org/models/CleTimer | | |
| Software | rCube | https://github.com/gagneurlab/rCube | | Last commit number: 463119 |

## Cell culture

K562 cells were obtained from DSMZ (DSMZ no.: ACC-10) and grown in RPMI 1640 medium (Thermo Fisher Scientific, 31870–074) supplemented with 10% heat-inactivated fetal bovine serum (Thermo Fisher Scientific, 10500–064) and 2 mM GlutaMAX (Thermo Fisher Scientific, 35050087) at 37°C and 5% $CO2$. Cells were routinely verified to be free of mycoplasma contamination using Plasmo Test Mycoplasma Detection Kit (InvivoGen, rep-pt1). K562 cells were authenticated at the DSMZ Identification Service according to standards for STR profiling (ASN-0002).

## TT-seq time series

TT-seq was performed as described (*Schwalb et al., 2016*), with minor modifications. Specifically, $2.5 \times 10^7$ cells from two biological replicates were used for each time point. Cells were exposed to 500 µM of 4-thiouracil (4sU, Carbosynth, NT06186) for 2, 5, 10, 15, 20, 30, 60 min at 37°C and 5% $CO_2$. Cells were harvested by centrifugation at 600 g for 2 min at 37°C. Cell pellets were lysed in 5 mL of QIAzol (Qiagen) and 150 ng of RNA spike-ins mix were added to each sample. RNA spike-ins were produced in house, based on ERCC-RNA sequences (sequences of spike-ins are described in *Supplementary file 6*). RNA spike-ins were produced as described (*Schwalb et al., 2016*). RNAs were extracted using QIAzol according to the manufacturer's instructions. RNAs were sonicated to obtain fragments of <6 kbp using AFAmicro tubes in a S220 Focused-ultrasonicator (Covaris Inc, parameters: 10 s, peak power 100, cycles 200, duty cycle 1%). The quality of RNAs and the size of fragmented RNAs were checked using Fragment Analyzer. 1 µg of each of the sonicated RNAs was stored at −80°C as total RNA (RNA-seq) and later eluted with miRNAeasy Micro Kit (Qiagen, 217084) together with 4sU-labeled purified RNAs.

4sU-labeled RNAs were purified from 300 µg of each of the fragmented RNAs. Biotinylation and purification of 4sU-labeled RNAs was performed as described (*Dölken et al., 2008*; *Schwalb et al., 2016*). Biotinylated 4sU-labeled RNAs were separated from unlabeled RNAs with streptavidin beads (Miltenyi Biotec, Bergisch Gladbach, Germany) and eluted in 100 mM DTT as described in

*Dölken et al. (2008)* and *Schwalb et al. (2016)*. 0.3M sodium acetate was added to 4sU-labeled purified RNAs and to total RNAs prior RNA extraction. RNAs were extracted and eluted using miR-NAeasy Micro Kit (Qiagen, 217084). The on-column DNAse I treatment (Qiagen, 79254) was performed for 15 min at 25°C. Prior to library preparation, total RNAs and 4sU-labeled purified RNAs were quantified using Qubit. Enrichment of 4sU-labeled *versus* unlabeled RNAs was analyzed by RT-qPCR using oligonucleotides amplifying selected regions of 4sU-labeled and unlabeled spike-ins (sequences of oligonucleotides are described in *Supplementary file 6*). Only 4sU-labeled purified samples showing ΔΔCt changes from 4 to 6 were subjected to library preparation (total RNAs were used as a control for normalization). 100 ng of input RNA was used for strand-specific library preparation according to the Ovation Universal RNA-seq System (NuGEN). Libraries were prepared using random hexamer priming only. The size-selected libraries were analyzed on a Fragment Analyzer before sequencing on the Illumina HiSeq 4000.

## Read alignment and counting

Paired-end 150 bp reads with additional 6 bp of barcodes were obtained for each sample. Reads were aligned using STAR version 2.5.0a (*Dobin et al., 2013*) in single pass mode. The genome Index was built against the full GENCODE version 24 annotation and the hg38 (GRCh38) genome assembly (Human Genome Reference Consortium) using 150 bp overhang size. Additional specified parameters were alignSJDBoverhangMin 2, chimSegmentMin 15, chimScoreMin 15, chimScoreSeparation 10, and chimJunctionOverhangMin 15. The aligned reads were filtered for duplicates using Picard tools version 2.5.0 (https://broadinstitute.github.io/picard/) using the option MarkDuplicates REMOVE_DUPLICATES = true. In average, each TT-seq sample yielded about 250 M reads and each RNA-seq sample about 55 M reads. For each sample, ~90% of the reads could be uniquely mapped to the reference genome. The duplication ratio was estimated to 55% by FastQC (https://www.bioinformatics.babraham.ac.uk/projects/fastqc/).

Using the rCube package (https://github.com/gagneurlab/rCube), all split reads (containing N stretches in Cigar string) were extracted to create a database of potential introns (~341 k). The obtained introns were classified relative to annotated introns and genetic elements from the GENCODE annotation (version 24 obtained from https://www.gencodegenes.org/releases/24.html). For each intron three characteristic counts were calculated: The numbers of reads starting in the upstream exon and extending into the intron ('donor'), the number of reads starting in the intron and extending into the downstream exon ('acceptor'), and all split reads matching the introns coordinates ('junction'). The reads were filtered using a bam quality score of 255. Reads having secondary alignment flag were discarded.

## Estimation of sample normalization factors and cross-contamination

To estimate the sample normalization factors $F_j$ that account for variations in sequencing depth as well as the overall newly synthesized RNA fraction and the fraction of cross-contamination $\chi_j$ of non-labeled reads in the TT-seq data, we modeled the expectation of counts $E_{ij}$ of spike-in $i$ in sample $j$ using a statistical model similar to the one of *Schwalb et al. (2016)*.

$$E_{ij} = F_j p_{ij} \left( \chi_j - \delta_i \chi_j + \delta_i \right) \tag{1}$$

$\chi_j$ is set to 1 for all RNA-seq samples, $\delta_i$ is 0 for labeled spike-ins and 1 for unlabeled spike-ins. The parameter $p_{ij}$ is the condition and spike-in specific extraction probability. The difference with (*Schwalb et al., 2016*) is to allow the parameter $p_{ij}$ to be condition-specific (TT-seq or RNA-seq), which turned out to model better cross-contamination of unlabeled RNA in the short duration TT-seq libraries. We set $p_{ij} = p_{ik}$ for all sets of $j$ and $k$ belonging to either RNA-seq samples, TT-seq samples or if $i$ belongs to a labeled spike-in. We assumed read count data to follow a negative binomial distribution with a common dispersion parameter for all data. The model parameters and the dispersion parameter were fitted as generalized linear model using maximum likelihood.

## Kinetic rate modeling and estimation

For each detected intron $i$ we modeled the concentrations $c_{i,l}$ of each of three characteristic bonds (donor, acceptor, junction) independently following a first order kinetic rate equation. Without loss

of generality, we consider in the following just one of the three equations - the other two behave the same.

$$\frac{d}{dt}c_i(t) = \alpha_i - \beta_i c_i(t) \tag{2}$$

We assume that all newly synthesized RNAs are labeled. The concentration of labeled bonds, assuming an initial concentration of 0, follows:

$$c_i(t, \text{labeled}) = \frac{\alpha_i}{\beta_i}\left(1 - e^{-t\beta_i}\right) \tag{3}$$

Also, the old, non-labeled RNA decays exponentially as $c_i(t, un\text{labeled}) = \frac{\alpha_i}{\beta_i}e^{-t\beta_i}$. Using the normalization factor $F_j$ of sample $j$, the labeling time $t_j$ and $\chi_j$ the cross-contamination of unlabeled RNAs in the purified fraction, the concentration can be mapped to its expected count $E_{i,j}$:

$$E_{i,j} = \frac{F_j\alpha_i}{\beta_i}\left(1 + e^{-t_j\beta_i}\left(\chi_j - 1\right)\right) \text{for TT} - \text{seq}; E_{i,j} = \frac{F_j\alpha_i}{\beta_i} \text{for RNA} - \text{seq} \tag{4}$$

We modeled read counts using the negative binomial distribution, a count distribution often used for RNA-seq data because it captures sampling noise and further sources of variations. The kinetic parameters $\alpha_i, \beta_i$ are estimated by maximizing the log likelihood $l = \sum_{i,j} \log\left(NB\left(k_{i,j}|E_{i,j}(\alpha_i,\beta_i), \theta\right)\right)$, where $k_{i,j}$ are the observed counts, using the BFGS numerical optimization algorithm and using the dispersion parameter obtained from the spike-ins analysis. The optimization was initialized 10 times with independent random parameters; the final solution comprises the median of all $\alpha_i, \beta_i$ over the different runs to compensate for numerical instabilities. We removed all donors, acceptors, junctions with too few counts ($\sum_j k_{i,j} < 100$) from the modeling.

Using the table in *Figure 2—figure supplement 1A* we map the rates $\alpha, \beta$ to the characteristic kinetic parameters of donor, acceptor, and junction. The whole modeling approach was implemented in R and is available as apackage called rCube (https://github.com/gagneurlab/rCube). Because the donor and acceptor bond half-life models work on a logarithmic scale, we present model errors as multiplicative errors given by the equation $median\left(\exp\left(\left|\log\left(\frac{y}{\hat{y}}\right)\right|\right)\right)$, with $y$ as the observation and $\hat{y}$ the prediction. More details about kinetic models are provided in the Appendix.

## Determination of the major isoforms

We applied the software Salmon (*Patro et al., 2017*) (index kmer size = 31) to all RNA-seq samples and mapped them against the full transcriptome of the GENCODE (Ver. 24) annotation. For each gene, we selected the isoform with the maximum mean TPM value across all RNA-seq samples as the major isoform. The major isoform was only used in the analyses in *Figures 2B–D*, *3D* and *7B*. Elsewhere, analyses were only relying on individual junction annotations.

## Estimation of the relative uncertainty for the kinetic parameters

To estimate relative uncertainty of the kinetic parameters in a conservative way we assumed that all donors and acceptors of the major isoform of a given GENCODE gene shared the same synthesis rate equal to the transcription rate of the gene. We further assumed that all products ('junctions') shared the same half-life equal to the mature RNA half-life. Because noise of these rate estimates is typically multiplicative, we computed the standard errors of the logarithm of these rates and reported relative uncertainties as the exponential of these standard errors.

## Comparison of 4sU-seq and TT-seq

Alignment, counting and estimation of normalization and cross-contamination factors of the RNA-seq data sets of *Schwalb et al. (2016)* was done as described above for our data. Counts for 4sU-seq and TT-seq was normalized using $\hat{K}_{i,j} = \frac{K_{i,j}}{F_j} - \frac{\chi_j K_{i,RNA-seq}}{F_{RNA-seq}}$, where $i$ denotes the split / unsplit reads as shown if *Figure 2A* for each intron of the major transcripts and $j$ is the sample (4sU- / TT-seq and replicate). Both replicates were pooled together.

## Branchpoint identification

Due to the limited availability of experimental branchpoint measurements, the prediction algorithm LaBranchoR (*Paggi and Bejerano, 2018*) was utilized to predict branchpoint positions within introns. We applied the model within the kipoi framework (http://kipoi.org/) to score the last 100 nucleotides of each intron and took the nucleotide with the maximum score for being the utilized branchpoint. The results were validated using experimental data of *Mercer et al. (2015)* where available.

## Estimation of single nucleotide effects

We identified nucleotide positions not predictive of donor or acceptor bond half-life and estimated the effect of the remaining single nucleotides on the donor bond half-life and on the acceptor bond half-life by regression. To this end, we modeled log-transformed half-lives of the donor bonds (and with a separate model of the acceptor bonds) as a weighted sum of each of the 20 nucleotides upstream or downstream of the donor, acceptor site and branchpoint, as well as of the GC frequency of the whole intron, the donor site, and the acceptor site. In this linear model, the reference sequence was chosen to be the consensus sequence so that the coefficients can be interpreted as the effects of substituting a consensus nucleotide to an alternative nucleotide. Lasso regression (*Tibshirani, 1996*) is a regularized linear regression method that can estimate some of the coefficients to be exactly 0 and that is therefore often used to select explanatory variables. We performed Lasso regression as implemented in glmnet (*Friedman et al., 2010*), choosing the largest shrinkage parameter at which the mean squared error (MSE) was within one standard error of the minimal MSE using 10-fold cross-validation. For donor bond half-life, as well as for acceptor bond half-life, the Lasso regression fit led to several nucleotide coefficients to be exactly 0. We then removed all the nucleotide positions where all single nucleotide effects had a coefficient equal to 0. Next, we estimated the single nucleotide effects of all remaining positions as well as the effect of GC frequency of the whole intron, the donor site, and the acceptor site on log-transformed donor and acceptor bond half-lives using ordinary least squares regression.

## Structures modeling

Images of spliceosome structures (PDB code 4PJO, 5O9Z, 5XJC) were drawn using Pymol (https://pymol.org/).

## Estimation of octamer effects and multivariate model

The number of occurrences of all 65,536 nucleotide octamers in the regions 15–100 nt downstream of the donor site, 100 nt upstream of the branchpoint, all nucleotides between the branchpoint and the five nt upstream of the acceptor site and 5–100 nt upstream of the acceptor site were counted allowing for two mismatches. The 15 nt immediately downstream of the donor site or 5 nt upstream of the acceptor site were excluded from the octamer search space because they were already incorporated in the single nucleotide model. Regions extending in the upstream or downstream exon were cropped to keep them within the intron. The base 2 logarithm of octamer pseudo-counts $\log_2$ (count +1) were used as covariates together with the GC frequency of the intron and the GC frequency of each region. The log-transformed donor/acceptor bond half-lives were the response variable. Lasso regression was applied to each region independently with 5-fold cross-validation to choose the optimal shrinkage parameter and select potential significant octamers. In a second step all selected octamers of each region were used together with the single nucleotide model as well as the GC frequency of the different regions, intron length and whether an intron is the first within the major transcript in a joint model to refine the selection of octamers (Lasso 10-fold cross-validation).

## Octamer match to ATtRACT database

We compared each octamer to all reported PWMs with at least 5 nucleotides of the ATtRACT database and calculated the ratio between the probability of the best matching position (PWM-score) and the highest possible probability for any octamer (RPM-score, *Cook et al., 2011*). Each octamer was padded with an equal number of 'N's at both sides if the PWM was longer than the octamer. We ranked all matches based on their RPM-score and kept only the best 5% for each PWM and removed afterwards all matches with a RPM-score less than 0.9. The remaining matches were considered as hits.

## Phylogenetic conservation of octamers

To calculate the phylogenetic conservation score for each octamer, we retrieved the PhastCons 100-way track (http://hgdownload.cse.ucsc.edu/goldenpath/hg38/phastCons100way/), which reports conservation across 99 vertebrates aligned to the human genome, and extracted the mean of all nucleotides for all matching positions. Octamers of the region 100 nt downstream of the donor site found to be predictive for donor site or acceptor bond half-life were also searched in the region 100 nt downstream of the donor site. Octamers of the region 100 nt upstream of the branchpoint or acceptor site or between the branchpoint and the acceptor site found to be predictive for donor or acceptor bond half-life were jointly searched in the region 100 nt upstream of the acceptor site, since these three regions were strongly overlapping. We also included as list of 2000 random octamers to estimate the background distribution in the same regions.

## Calculation of splicing yield

We define the splicing yield of donor $\eta_{donor}$ and acceptor $\eta_{acceptor}$ as follows:

$$\eta_{donor} = \sum_{\{acceptor\}} \frac{\alpha_{junction}(donor, acceptor)}{\alpha_{donor}}$$
$$\eta_{acceptor} = \sum_{\{donor\}} \frac{\alpha_{junction}(donor, acceptor)}{\alpha_{acceptor}}$$

(5)

where $\alpha_{donor}$ and $\alpha_{acceptor}$ denote the synthesis rates of the donor site and of the acceptor site phosphodiester bond, respectively, and $\alpha_{junction}(donor, acceptor)$ denote the synthesis rate of the spliced exon-exon phosphodiester bond utilizing the specified donor and acceptor. Since the first-order kinetic model does systematically underestimate junction synthesis rates and overestimate donor synthesis rates, we switched to the alternative kinetic models to estimate these rates. However, since the first-order kinetic model is more robust and the acceptor kinetics do not include a delay we used the first-order kinetic model for the estimation of the acceptor synthesis rate. We defined the intron splicing yield $\eta$ as the acceptor site splicing yield because its estimation is more robust compared to the donor site splicing yield.

## Code availability

All the code used for counting donor site, acceptor sites, and junction reads as well as estimating the kinetic rates is available in the R package rCube (https://github.com/gagneurlab/rCube; *Wachutka et al., 2017*). The single nucleotide model is shared in the model repository Kipoi (http://kipoi.org/models/CleTimer/; *Avsec et al., 2019*).

## Accession code

The sequencing data and processed files were deposited in NCBI Gene Expression Omnibus (GEO) database under accession code GSE129635.

# Acknowledgements

We thank Jun Cheng for initial pre-processing of the data and stimulating discussions. We thank Carina Demel for help in developing the rCube package and Carina Demel and Björn Schwalb for support during the pre-processing of the data. We thank Hauke Hillen for help with structural modeling. LC was funded by EMBO Long-Term Postdoctoral Fellowship (ALTF-1261–2014). LW and JG were supported by EU Horizon2020 Collaborative Research Project SOUND (633974). PC was funded by Advanced Grant TRANSREGULON of the European Research Council and the Volkswagen Foundation.

# Additional information

### Funding

| Funder | Grant reference number | Author |
| --- | --- | --- |
| European Molecular Biology Organization | ALTF-1261-2014 | Livia Caizzi |

| Horizon 2020 | SOUND 633974 | Leonhard Wachutka Julien Gagneur |
| European Research Council | | Patrick Cramer |
| Volkswagen Foundation | | Patrick Cramer |

The funders had no role in study design, data collection and interpretation, or the decision to submit the work for publication.

## Author contributions

Leonhard Wachutka, Conceptualization, Data curation, Software, Formal analysis, Validation, Investigation, Visualization, Methodology, Writing—original draft, Writing—review and editing, Designed and carried out the bioinformatics analysis; Livia Caizzi, Conceptualization, Data curation, Validation, Investigation, Visualization, Methodology, Writing—original draft, Writing—review and editing, Optimized and carried out TT-seq experiments, contributed to the design of the bioinformatics analysis, and used molecular modeling to interpret results; Julien Gagneur, Conceptualization, Resources, Software, Supervision, Funding acquisition, Investigation, Visualization, Methodology, Writing—original draft, Project administration, Writing—review and editing; Patrick Cramer, Conceptualization, Resources, Supervision, Funding acquisition, Investigation, Visualization, Methodology, Writing—original draft, Project administration, Writing—review and editing

## Author ORCIDs

Leonhard Wachutka https://orcid.org/0000-0002-5959-040X
Livia Caizzi https://orcid.org/0000-0001-9723-6893
Julien Gagneur https://orcid.org/0000-0002-8924-8365
Patrick Cramer https://orcid.org/0000-0001-5454-7755

## Decision letter and Author response

Decision letter https://doi.org/10.7554/eLife.45056.066
Author response https://doi.org/10.7554/eLife.45056.067

# Additional files

## Supplementary files

• Supplementary file 1. Coordinates (start, stop, strand, chromosome) of all junctions (exon – exon boundaries) with at least 10 reads in all samples.
DOI: https://doi.org/10.7554/eLife.45056.016

• Supplementary file 2. Synthesis rate and half-life time of donor, acceptor and junction bonds estimated using different models as well as splicing yield estimates for junction bonds. Each bond is annotated based on GENCODE and major isoform estimation was done using the software Salmon.
DOI: https://doi.org/10.7554/eLife.45056.017

• Supplementary file 3. Synthesis rate, mean donor-bond and acceptor-bond half-life and junction-bond half-life aggregated by median of all introns belonging to the major isoform determined by Salmon.
DOI: https://doi.org/10.7554/eLife.45056.018

• Supplementary file 4. Single nucleotide multiplicative effects on the donor-bond or acceptor-bond half-life. For each position, it is annotated whether there is a known contact between a spliceosomal structure and the processed RNA.
DOI: https://doi.org/10.7554/eLife.45056.019

• Supplementary file 5. Octamer multiplicative effects on the donor-bond or acceptor-bond half-life and their match to ATtRACT database.
DOI: https://doi.org/10.7554/eLife.45056.020

• Supplementary file 6. Spike-in sequences used in the experiment and oligonucleotides sequences used in RT-qPCR to amplify selected regions of 4sU-labeled and unlabeled spike-ins.

DOI: https://doi.org/10.7554/eLife.45056.021
• Transparent reporting form
DOI: https://doi.org/10.7554/eLife.45056.022

## Data availability

The sequencing data and processed files were deposited in NCBI Gene Expression Omnibus (GEO) database under accession code GSE129635.

The following dataset was generated:

| Author(s) | Year | Dataset title | Dataset URL | Database and Identifier |
|---|---|---|---|---|
| Wachutka L, Caizzi L, Gagneur J, Cramer P | 2019 | Global donor and acceptor splicing site kinetics in human cells | https://www.ncbi.nlm.nih.gov/geo/query/acc.cgi?acc=GSE129635 | NCBI Gene Expression Omnibus, GSE129635 |

The following previously published datasets were used:

| Author(s) | Year | Dataset title | Dataset URL | Database and Identifier |
|---|---|---|---|---|
| Schwalb B, Michel M, Zacher B, Frühauf K, Demel C, Tresch A, Gagneur J, Cramer P | 2016 | TT-seq maps the human transient transcriptome | https://www.ncbi.nlm.nih.gov/geo/query/acc.cgi?acc=GSE75792 | NCBI Gene Expression Omnibus, GSE75792 |
| Mercer TR, Clark MB, Andersen SB, Brunck ME, Haerty W, Crawford J, Taft RJ, Nielsen LK, Dinger ME, Mattick JS | 2015 | Genome-wide discovery of human splicing branchpoints | https://www.ncbi.nlm.nih.gov/geo/query/acc.cgi?acc=GSE53328 | NCBI Gene Expression Omnibus, GSE53328 |
| Kondo Y, Oubridge C | 2015 | Crystal structure of human U1 snRNP, a small nuclear ribonucleoprotein particle, reveals the mechanism of 5' splice site recognition | https://www.rcsb.org/structure/4PJO | RCSB Protein Data Bank, 4PJO |
| Bertram K, Agafonov DE, Dybkov O, Haselbach D, Leelaram MN, Will CL, Urlaub H, Kastner B, Luhrmann R, Stark H | 2017 | Cryo-EM Structure of a Pre-catalytic Human Spliceosome Primed for Activation | http://www.rcsb.org/structure/5O9Z | RCSB Protein Data Bank, 5O9Z |
| Zhang X, Yan C, Hang J, Finci LI, Lei J, Shi Y | 2017 | An Atomic Structure of the Human Spliceosome | https://www.rcsb.org/structure/5XJC | RCSB Protein Data Bank, 5XJC |

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

## Appendix 1

DOI: https://doi.org/10.7554/eLife.45056.023

# Notations, definitions, and relation to splicing quantities found in literature

## 1.1 Definitions of parameters used in this study

We consider RNA metabolism in unsynchronized cells at steady state. Synthesis balances out degradation at steady-state for any molecular species, independently of the kinetics. Denoting the steady state concentration $c$ and the steady-state synthesis rate $\alpha$ (amount produced per unit of time) of such a molecular species of interest, we define the steady-state degradation rate constant $\beta$, such that $c = \frac{\alpha}{\beta}$. We underscore that these constants are defined at steady-state without any assumption on the kinetics.

As molecular species of interest, we considered the donor (exon-intron), acceptor (intron-exon) and junction (exon-exon) phosphodiester bonds. We defined for each of them their corresponding rates (*Appendix 1—table 1*).

**Appendix 1—table 1.** Rate definitions of phophodiester bonds.

| Phosphodiester bond | Steady-state synthesis rate | Steady-state degradation rate |
|---|---|---|
| Donor | $\alpha_D$ | $\beta_D = \sigma_D + \lambda_D$ |
| Acceptor | $\alpha_A$ | $\beta_A = \sigma_A + \lambda_A$ |
| Junction | $\alpha_J$ | $\beta_J = \lambda_J$ |

DOI: https://doi.org/10.7554/eLife.45056.024

We assume that the donor bond can only disappear by splicing with rate $\sigma_D$ or by precursor RNA degradation with rate $\lambda_D$. Similarly, we assume that the acceptor bond can only disappear by splicing with rate $\sigma_A$ or by precursor RNA degradation with rate $\lambda_A$. We assume that the junction bond only disappears by mature RNA degradation with rate $\lambda_J$.

We define the 'splicing yield' of an acceptor bond $\eta_A$ as the fraction of acceptor bonds that get spliced $\eta_A := \frac{\sigma_A}{\sigma_A + \lambda_A}$. For an acceptor that is spliced with a single donor, this is equivalently defined by the relation $\alpha_J = \eta_A \alpha_A$ because of flux balance at steady-state. Generally, as acceptors can splice with multiple donors, we sum up the synthesis rates of the different resulting junctions $J$, leading to:

$$\eta_A := \frac{\sum_J \alpha_J}{\alpha_A} \tag{1.1}$$

The splicing yield for donor bonds $\eta_D$ is analogously defined.

## 1.2 Relation to steady-state quantities used in literature

In this section we are relating our parametrization defined in section 1.1 with other RNA-seq based steady-state quantities typically found in the literature. Many studies used quantities that combine donor and acceptor reads. A reasonable assumption is that RNA polymerase II does not drop off during the transcription of an intron. With this assumption, steady-state synthesis rates of the donor and of the acceptor of one intron are equal to the transcription rate $\mu$ ($\mu := \alpha_D = \alpha_A$). Moreover, the steady-state amount of donor and acceptor unsplit RNA-seq reads of a same intron are often roughly equal, implying that the relative difference of degradation rates of the donor and acceptor bonds are small compared to their average. We define the average degradation rate of the donor and of the acceptor bond

as $\sigma := \frac{\beta_A + \beta_D}{2} \approx \beta_A \approx \beta_D$. **Appendix 1—table 2** expresses several splicing quantities often used in literature using these parameters and these assumptions.

**Appendix 1—table 2.** Relations of different splicing quantities. Square brackets stand for concentrations. The lariat is a spliced-out intron, known to be quickly degraded. Because lariat degradation rate may not be faster than splicing rate, we therefore consider it in the model of the concentration of intronic reads.

| Steady-state quantity | Value |
| --- | --- |
| $[exon - intron\ bond]$ | $\frac{\alpha_D}{\beta_D} = \frac{\alpha_D}{\sigma_D + \lambda_D} \approx \frac{\mu}{\sigma}$ |
| $[intron - exon\ bond]$ | $\frac{\alpha_A}{\beta_A} = \frac{\alpha_A}{\sigma_A + \lambda_A} \approx \frac{\mu}{\sigma}$ |
| $[exon - exon\ bond]$ | $\frac{\alpha_J}{\beta_J} = \frac{\mu\eta}{\lambda_J}$ |
| $[exon]$ | $\approx \frac{\mu}{\sigma} + \frac{\mu\eta}{\lambda_J}$ |
| $[intron]$ | $\approx \frac{\mu}{\sigma} + [lariat]$ |
| 3'SS or 5'SS intron-exon ratio $\frac{[intron]}{[exon]}$ (**Khodor et al., 2011**) | $\approx \frac{1 + [lariat]\frac{\sigma}{\mu}}{1 + \frac{\sigma\eta}{\lambda}}$ |
| Splicing efficiency $\frac{[exon-exon]}{[exon-intron]}$ (**Převorovský et al., 2016**) | $\approx \frac{\sigma}{\lambda_J \eta}$ |
| Splicing Index $\frac{[exon-exon]}{[intron-exon]}$ (**Schlackow et al., 2017**) | $\approx \frac{\sigma}{\lambda_J \eta}$ |

DOI: https://doi.org/10.7554/eLife.45056.025

Note that these steady-state quantities combine effects of splicing, degradation and splicing yield and are therefore not ideal to characterize splicing kinetics. For instance, splicing efficiency can be affected by mature RNA degradation rate.

## 1.3 Percent spliced-in

One common metric used to quantify alternative splicing is the percent splice-in (psi) (**Schafer et al., 2015**). Given three successive exons (1,2,3), PSI ($\Psi$ is defined as the ratio of inclusion split reads ($n_{1,2}, n_{2,3}$) and the sum of inclusion split reads and exclusion split reads ($n_{1,3}$).

$$\Psi = \frac{n_{1,2} + n_{2,3}}{n_{1,2} + n_{2,3} + 2n_{1,3}} \tag{1.2}$$

We assume that we only have the two isoforms, including and excluding the middle exon (2), such that $n_{1,2} = n_{2,3}$. Moreover, we assume no polymerase drop off such that all introns are transcribed at the same rate. Further assuming that each junction is spliced at its splicing rate constant $\sigma$ and degraded at the mature RNA degradation rate constant $\lambda$, and that the splicing yield is 1 for all sites, we obtain:

$$\Psi = \frac{\frac{\sigma_{1,2}}{\lambda_{1,2}}}{\frac{\sigma_{1,2}}{\lambda_{1,2}} + \frac{\sigma_{1,3}}{\lambda_{1,3}}} \tag{1.3}$$

The simple form of Equation 1.3 is a particular case that is based on many assumptions. Nonetheless, it shows that $\psi$ is not only determined by the relative splicing rates but also by the relative stabilities of the isoforms. Steady-state data does not allow untangling these quantities. Moreover, it shows that splicing yield is not a quantity that is redundant with $\Psi$.

## 2 Kinetic models

### 2.1 Models for the donor bonds and for the acceptor bonds

#### 2.1.1 Constant degradation rate

We used a model that assumes first order kinetics, that is with constant parameters $\alpha_D$ and $\beta_D$. Denoting $[D]$ the concentration of donor bonds, we modelled:

$$\frac{d}{dt}[D] = \alpha_D - \beta_D[D] = \mu_D - \sigma_D[D] - \lambda_D[D] \tag{2.1}$$

This leads to:

$$[D] = \frac{\alpha_D}{\beta_D}\left(1 - e^{-\beta_D t}\right) \tag{2.2}$$

#### 2.1.2 Constant splicing rate with a fixed delay

We also used a more complex model that takes into account that splicing can only be completed after the transcription of the corresponding acceptor site, which introduces a delay $\delta$. Assuming for the sake of simplicity that degradation exhibits the same time delay we find:

$$\beta_D = \left\{ \begin{array}{l} 0: t<\delta \\ \tilde{\beta}_D: t \geq \delta \end{array} \right\} \tag{2.3}$$

The resulting delayed differential equation model solves as:

$$[D] = \left\{ \begin{array}{l} \alpha_D t : t<\delta \\ \alpha_D \delta + \frac{\alpha_D}{\beta_D}\left(1 - e^{-\tilde{\beta}_D(t-\delta)}\right) : t \geq \delta \end{array} \right\} \tag{2.4}$$

We show below that this model is difficult to fit on our data. Although there is no obvious biophysical motivation for the delay to be equal for degradation and splicing, a more complex model would need more parameters and be even more difficult to be fitted on our data.

### 2.2 Models for the acceptor site

Since the acceptor site can be spliced immediately after its transcription, we did not consider delayed models for the acceptor bonds. Instead, we only considered a first order kinetic model with constant synthesis rate as in Equation (2.2).

### 2.3 Model for the junction bonds

#### 2.3.1 Constant junction formation rate model

We used a model that assumes first order kinetics, that is with constant parameters $\alpha_J$ and $\beta_J$. Denoting $[J]$ the concentration of junction bonds, we modelled:

$$\frac{d}{dt}[J] = \alpha_J - \beta_J[J] \tag{2.5}$$

which leads to:

$$[J] = \frac{\alpha_J}{\beta_J}\left(1 - e^{-\beta_J t}\right) \tag{2.6}$$

### 2.3.2 Coupled model

We also used a more complex model that describes the formation of junction bonds as the outcome of splicing with a coupled first order kinetic model. Assuming that each acceptor bond that disappears through splicing creates a junction, we have $\alpha_J = \sigma_A[A]$ in (2.6).

This leads to:

$$[J] = \frac{\alpha_J}{\beta_J(\beta_J - \beta_A)}\left(\left(1 - e^{-\beta_A t}\right)\beta_J - \left(1 - e^{-\beta_J t}\right)\beta_A\right) \tag{2.7}$$

Because of possible alternative splicing, this model was fitted to the split reads data only, that is without considering the unsplit reads of the corresponding donor and acceptor sites. Below we show that the coupled model is difficult to fit and therefore we worked with the constant rate model during our whole analysis except when calculating splicing yield.

To calculate the splicing yield $\eta_A$ of an acceptor as in Equation (1.1), we (i) estimated $\alpha_J$ by fitting Equation (2.7) to the split reads of all junctions the acceptor is involved in, and (ii) estimated $\alpha_A$ using the first order kinetic model on intron-exon reads.

## 3 Comparison of the models

### 3.1 Parameter estimation

The constant rate models use only two parameters whereas the more complex models (the delay model and the coupled splicing model) used three parameters. To compare how well we can retrieve the model parameters from observed data, we simulated counts based on typical synthesis, splicing, degradation and delay constants using typical negative binomial noise and parameter ranges. We then tried to retrieve the parameters using a maximum likelihood approach with random initialization. We compared the results to the ground truth and found that the two-parameter models (*Appendix 1—figure 1*) were more accurately fitted than the three-parameter models (*Appendix 1—figures 2,3*). All models showed an unbiased estimate for the synthesis rate, which is important to calculate the splicing yield.

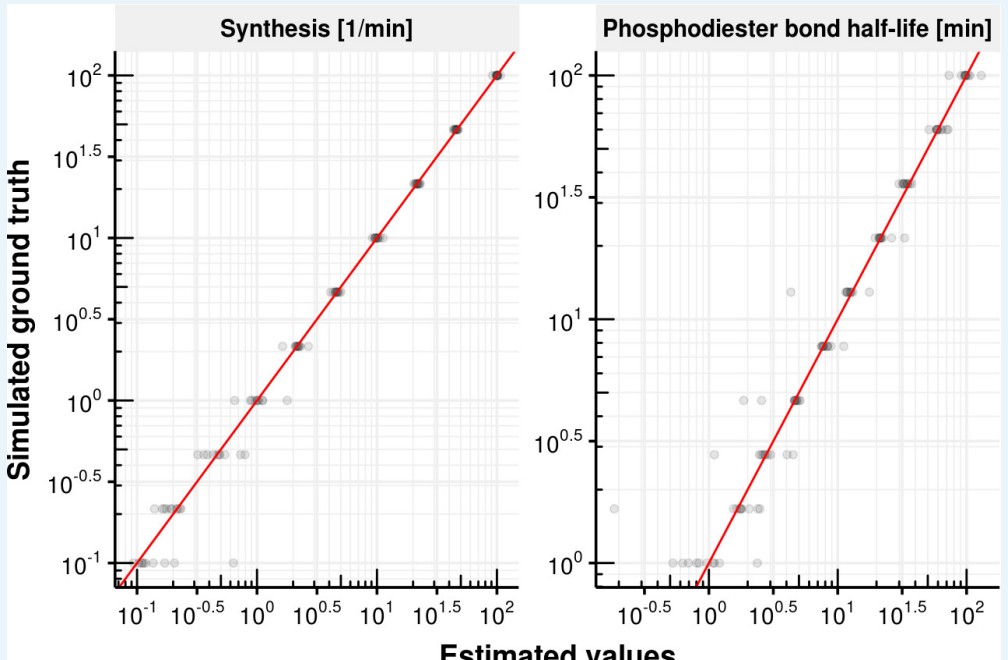

**Appendix 1—figure 1.** Evaluation of first order kinetic model fitting. Estimated synthesis rate $\alpha$ (left) and phosphodiester bond half-life ($\frac{log(2)}{\beta}$, right) on x-axis vs. simulated ground truth synthesis rate and phosphodiester bond half-life. The red line shows the identity line y=x.
DOI: https://doi.org/10.7554/eLife.45056.026

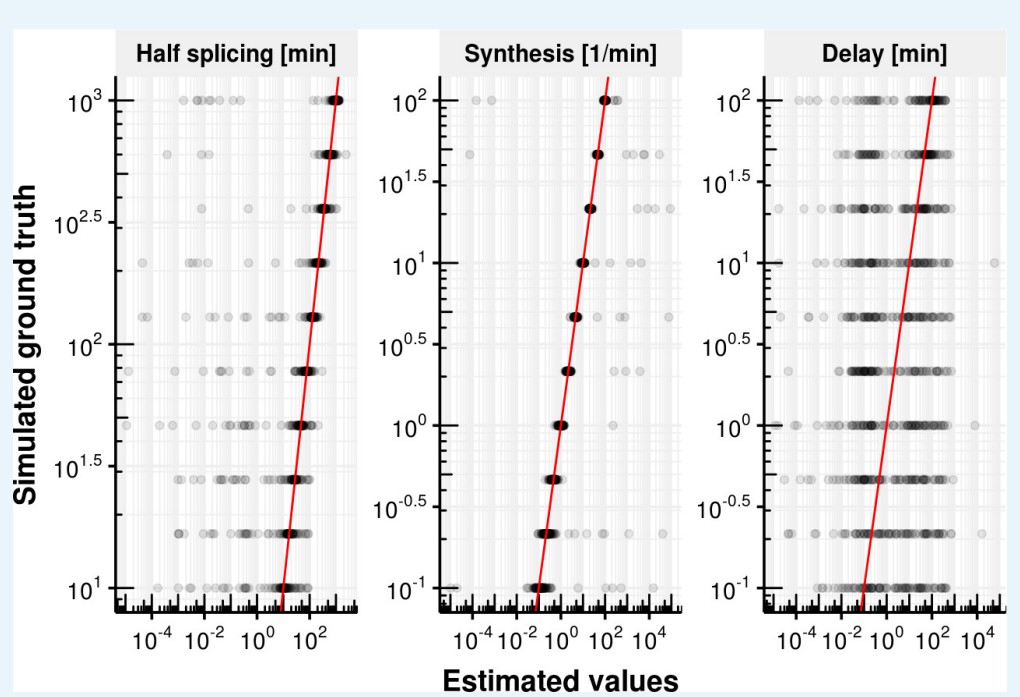

**Appendix 1—figure 2.** Evaluation of delay model fitting. Estimated splicing half-time ($\frac{log(2)}{\beta}$), synthesis rate $\alpha$ and delay time $\delta$ on the x-axis vs. simulated ground truth half-life, synthesis rate and delay. The red line shows the identity line. Most estimates are close to the identity line except for the delay.

DOI: https://doi.org/10.7554/eLife.45056.027

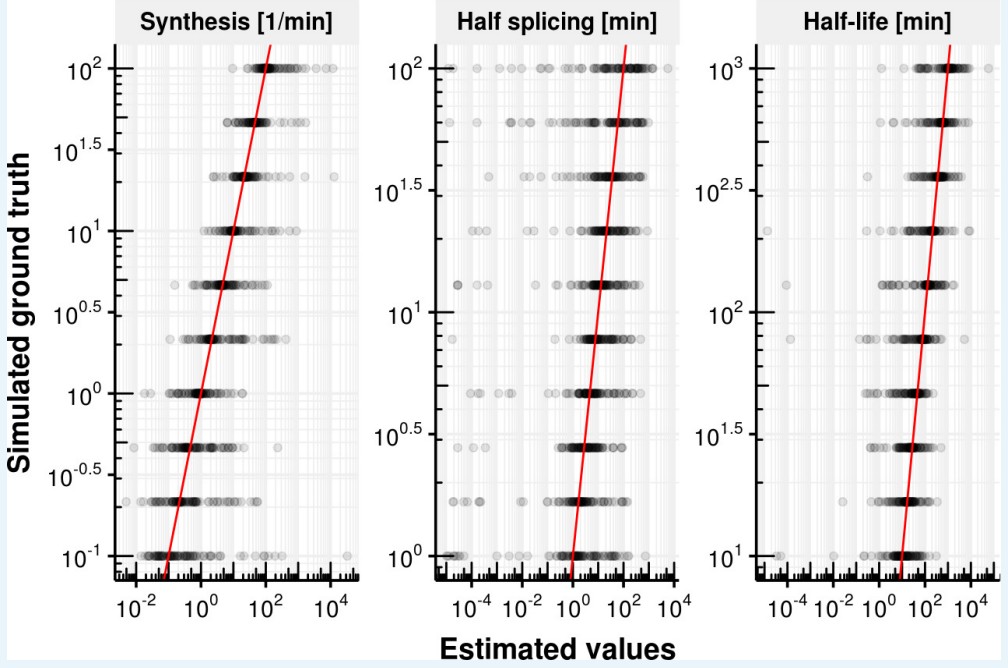

**Appendix 1—figure 3.** Evaluation of coupled model fitting. Estimated synthesis rate $\alpha$, splicing half-time ($\frac{log(2)}{\beta_A}$) and junction bond half-life ($\frac{log(2)}{\beta_J}$) on the x-axis vs. simulated ground

truth synthesis rate, splicing half time and half-life time. The red line shows the identity. Most estimates are unbiased but show a high degree variation.

DOI: https://doi.org/10.7554/eLife.45056.028

## 3.2 Relation between the parameters of the different models

Since the two-parameter models are more accurately fitted than the three-parameter models, we investigated the validity of approximations relating the parameter of the two-parameter models to those of the three-parameter models.

For the donor bonds, we asked how well the sum of the delay $\delta$ and the splicing half-time $\frac{log(2)}{\beta_D}$ is approximated by the donor bond half-life $\frac{log(2)}{\beta_D}$ estimated by the constant rate model.

For the junction bonds, we asked how well the sum of the splicing half-time $\frac{log(2)}{\beta_A}$ and the mature RNA half-life $\frac{log(2)}{\beta_J}$ is approximated by the junction bond half-life estimated by the constant rate model. To assess how many introns are affected by potential biases of using the constant rate model instead of the more complex models, we created histograms of the estimated error for each model parameter given the observed distribution of synthesis rates, splicing times and half-lives (*Appendix 1—figures 4–7*). To get a distribution for the delay, the intron length was divided by an assumed polymerase velocity of 4 kb/min (*Jonkers et al., 2014*; *Gressel et al., 2017*). Compared to our estimated measuring precision of 180% fold for synthesis and 32% fold for half-life, the absolute errors are negligible for the synthesis rates and for the splicing half-time.

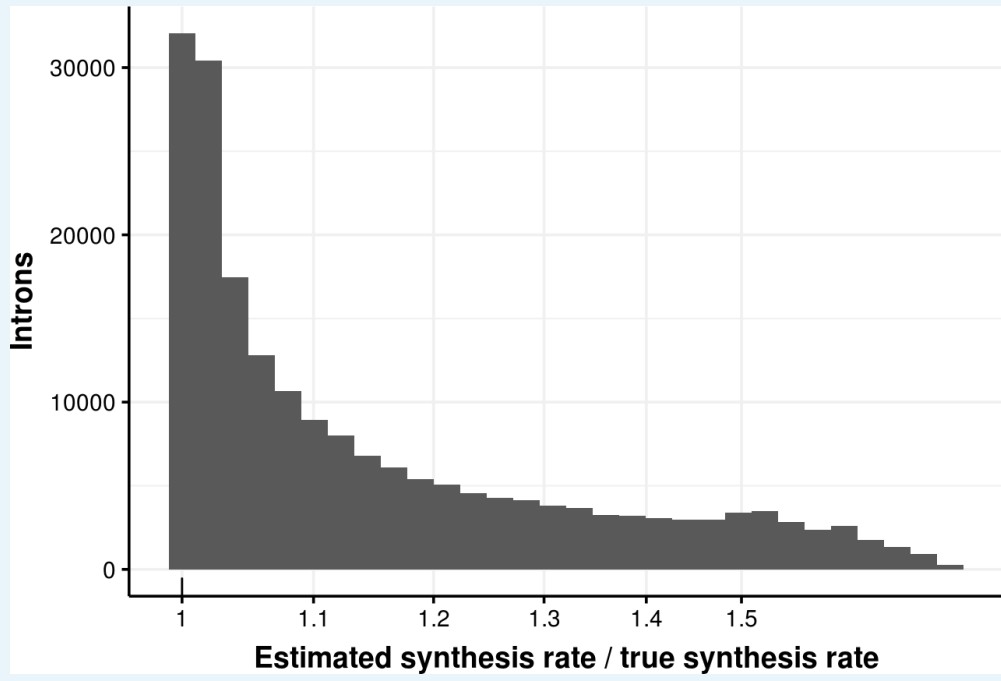

**Appendix 1—figure 4.** Estimated bond synthesis rate error for delay model. Histogram of the ratio of estimated and true synthesis rate of donors simulated with the delay model. The median error is 1.08.

DOI: https://doi.org/10.7554/eLife.45056.029

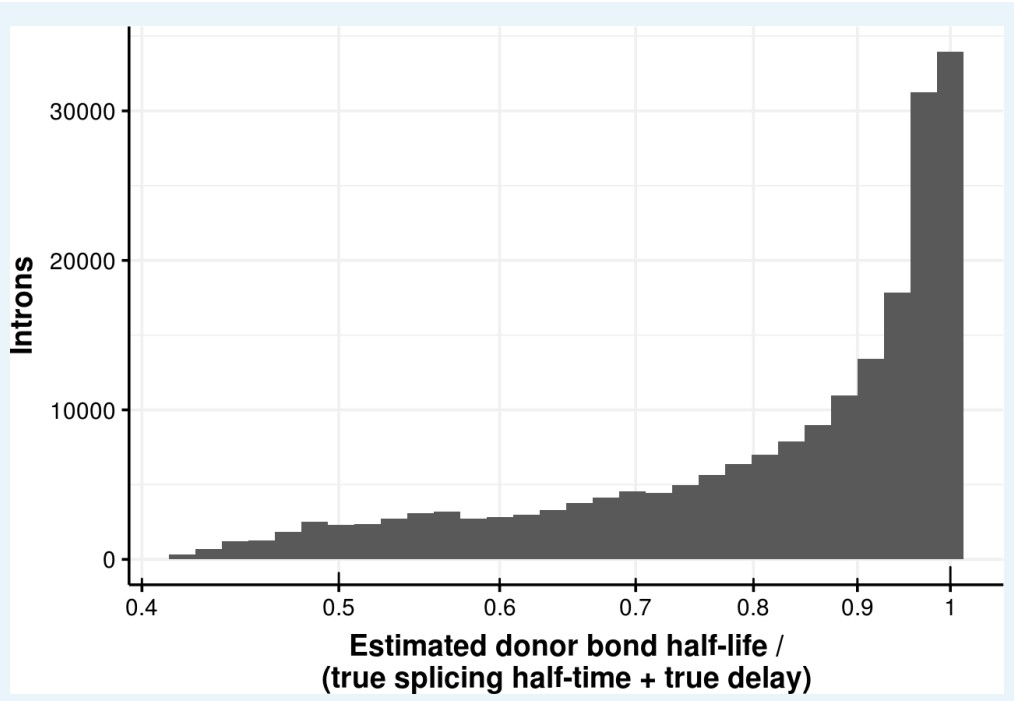

**Appendix 1—figure 5.** Estimated bond half-life error for delay model. Histogram of the ratio of donor bond half-life estimated using a first order kinetic model and the sum of the true splicing half-time and delay of donors simulated using a delay model. The median error is 0.89.

DOI: https://doi.org/10.7554/eLife.45056.030

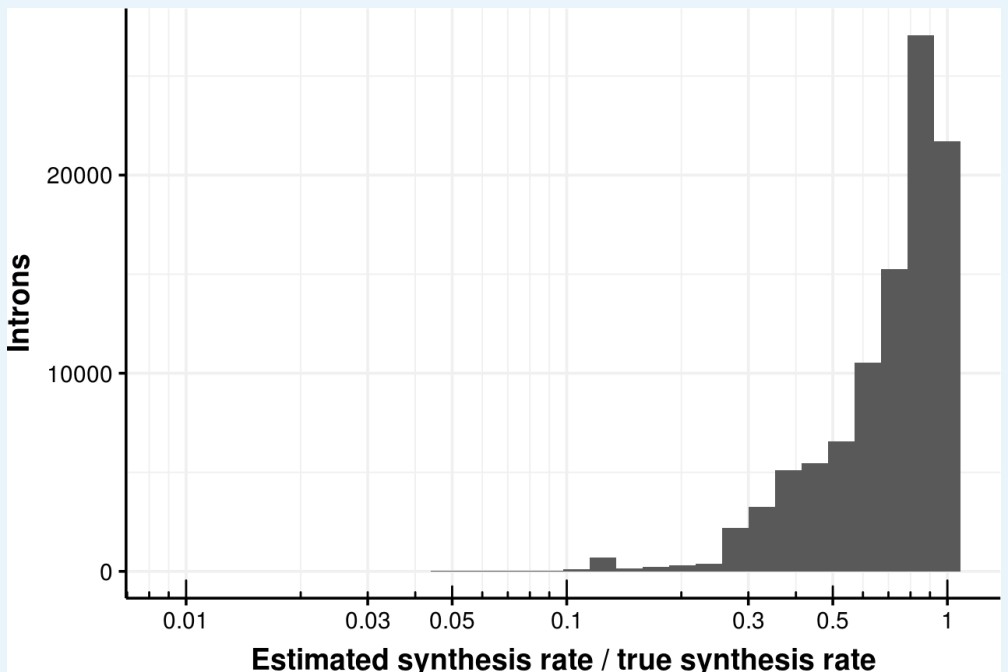

**Appendix 1—figure 6.** Estimated bond synthesis rate error for the coupled model. Histogram of the ratio of estimated and true synthesis rate of junctions based on the coupled model. The median error is 0.8.

DOI: https://doi.org/10.7554/eLife.45056.031

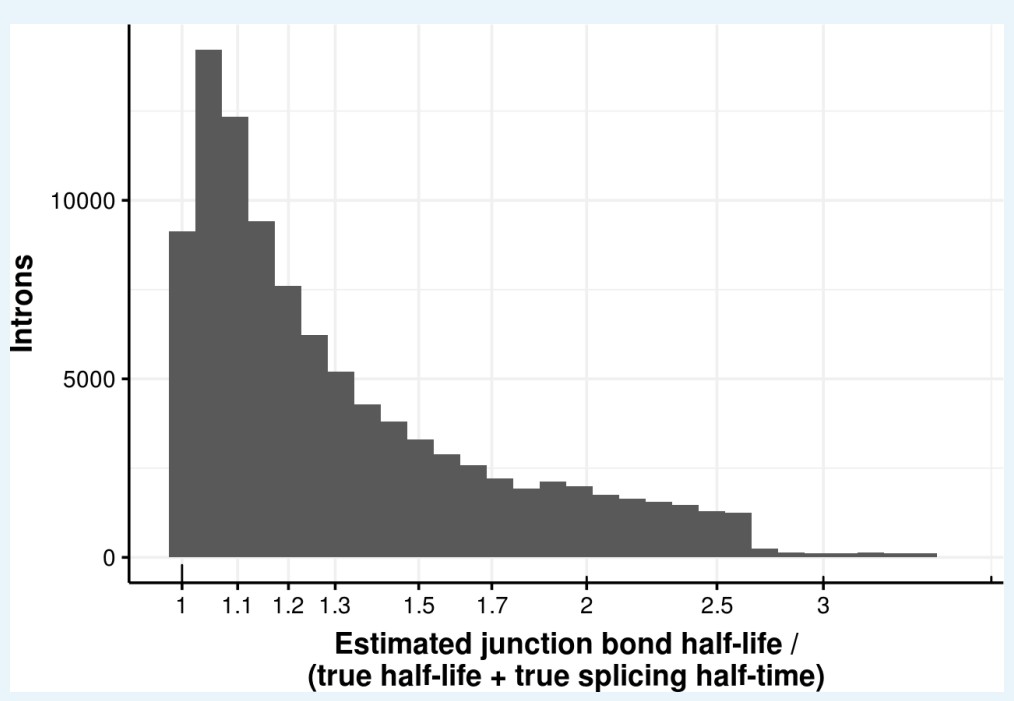

**Appendix 1—figure 7.** Estimated bond half-life error for coupled model. Histogram of the ratio of junction bond half-life estimated with a first order kinetic model and the sum of the true processed RNA half-life and of the splicing half-time of junction data simulated with a coupled model. The median error is 1.2.

DOI: https://doi.org/10.7554/eLife.45056.032

## 3.3 Experimental data

To further assess whether all three models give similar estimations on real data, we applied the donor models to all donor sites as well as the junction models to all junctions (*Appendix 1—figures 8–15*). The results were similar in all models and showed only marginal differences. However, synthesis rates and half-lives were systematically underestimated or overestimated. This has no relevance if rates are only compared within one model, but needs to be taken into account if acceptor and junction rates are compared, as in the case of splicing yield. Therefore, we based our splicing yield estimates on the more complex models.

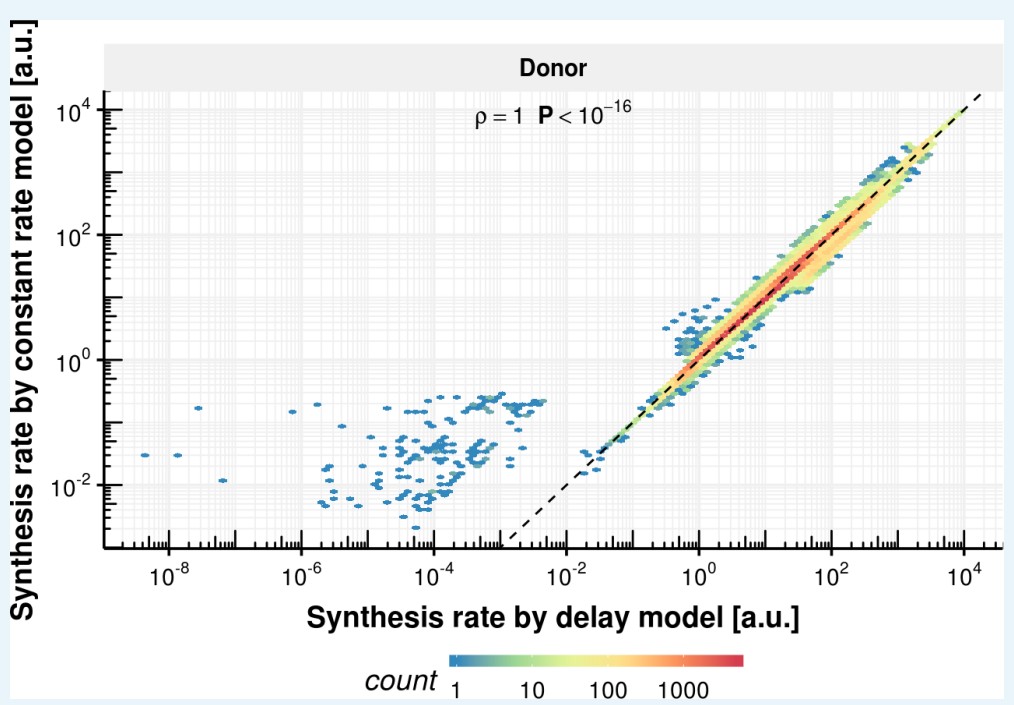

**Appendix 1—figure 8.** Synthesis rate comparison of delay and constant rate model. Synthesis rate estimated with the constant splicing rate model (y-axis) vs. synthesis rate estimated with the delay model (x-axis) for the donor bond based on the observed experimental data.

DOI: https://doi.org/10.7554/eLife.45056.033

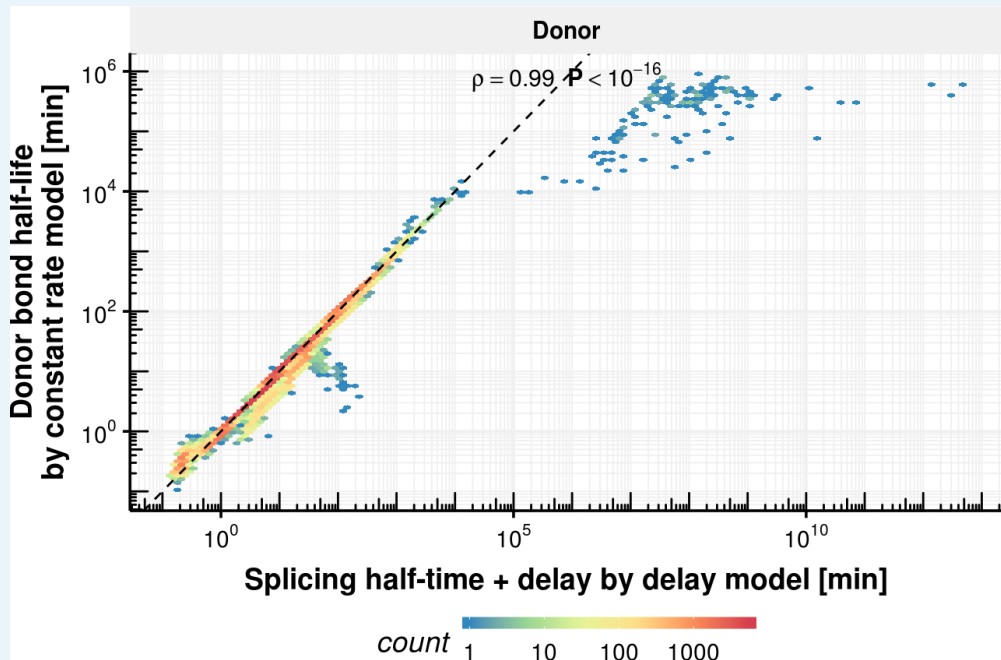

**Appendix 1—figure 9.** Donor bond half-life comparison of delay and constant rate model. Donor bond half-life estimated with the constant splicing rate model (y-axis) vs. the sum of the splicing half-time and delay estimated with the delay model (x-axis) for the donor bond (right) based on the observed experimental data.

DOI: https://doi.org/10.7554/eLife.45056.034

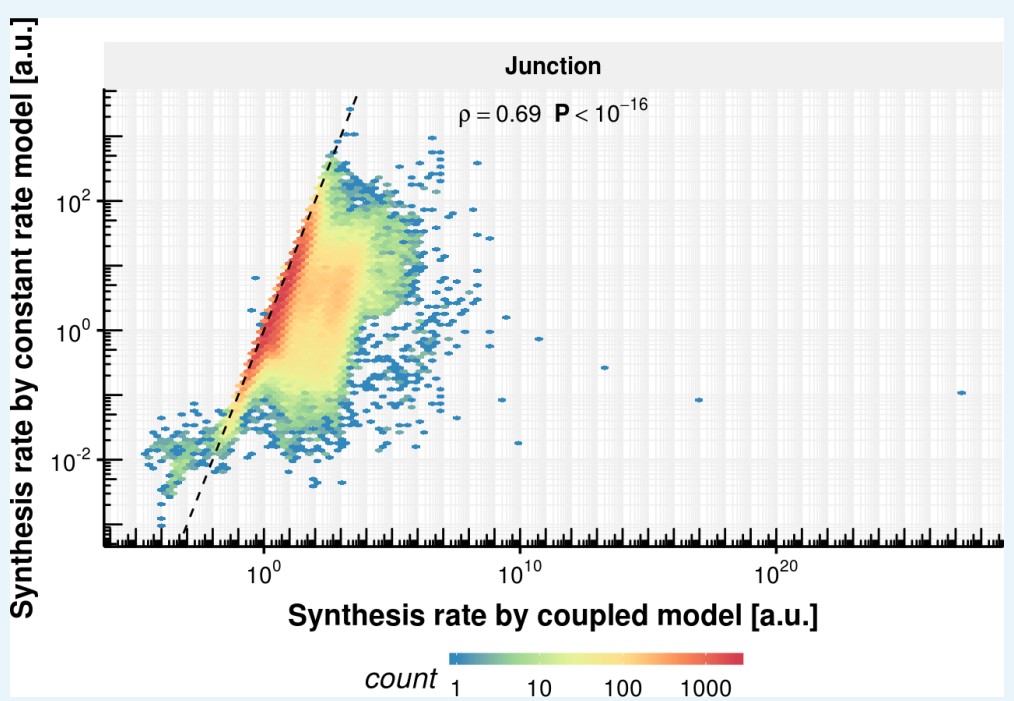

**Appendix 1—figure 10.** Synthesis rate comparison of coupled and constant rate model. Estimated synthesis rate of the constant splicing rate model (y-axis) vs estimated synthesis rate of the coupled model based on the observed experimental data.

DOI: https://doi.org/10.7554/eLife.45056.035

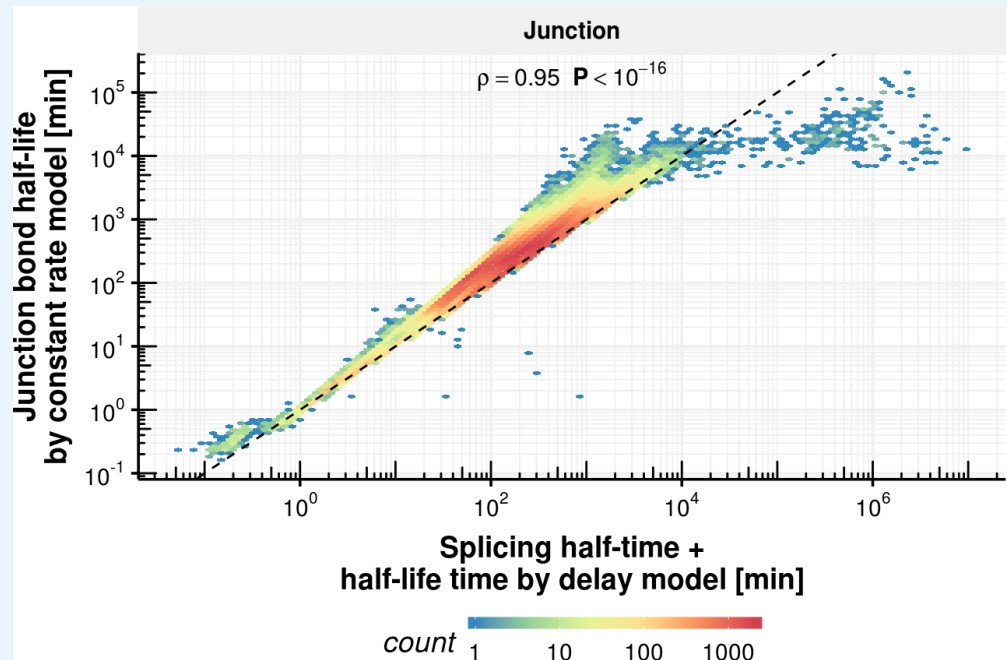

**Appendix 1—figure 11.** Junction bond half-life comparison of delay and constant rate model. Junction bond half-life estimated with the constant splicing rate model (y-axis) vs. the sum of splicing half-time and half-life estimated with the coupled model (x-axis) based on the observed experimental data.

DOI: https://doi.org/10.7554/eLife.45056.036

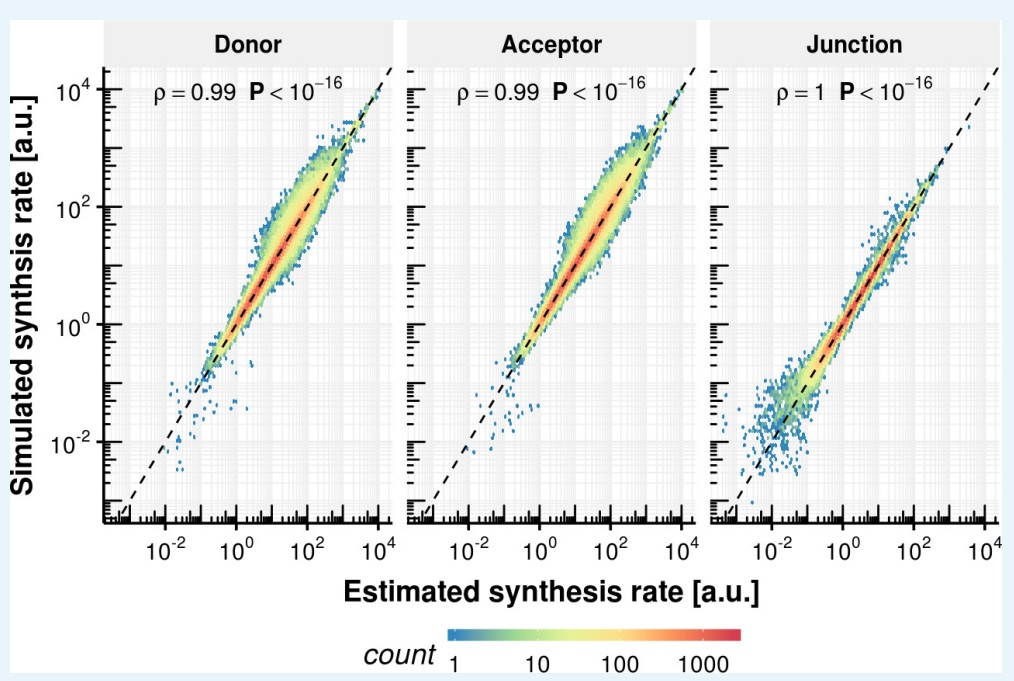

**Appendix 1—figure 12.** Synthesis rate precision of rate estimation procedure. Estimated synthesis rate based on simulated counts on the x-axis vs. the ground truth synthesis rate for donor, acceptor and junctions.

DOI: https://doi.org/10.7554/eLife.45056.037

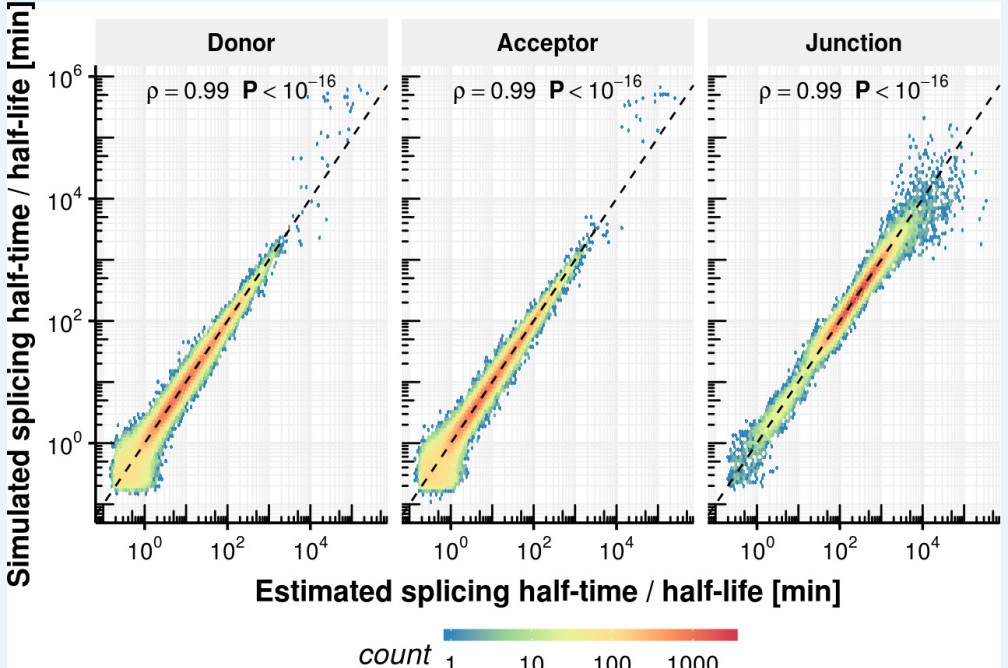

**Appendix 1—figure 13.** Splicing half-time precision of rate estimation procedure. Estimated half splicing time for donor and acceptor as well as half-life for junctions based on simulated counts on the x-axis vs. the ground truth.

DOI: https://doi.org/10.7554/eLife.45056.038

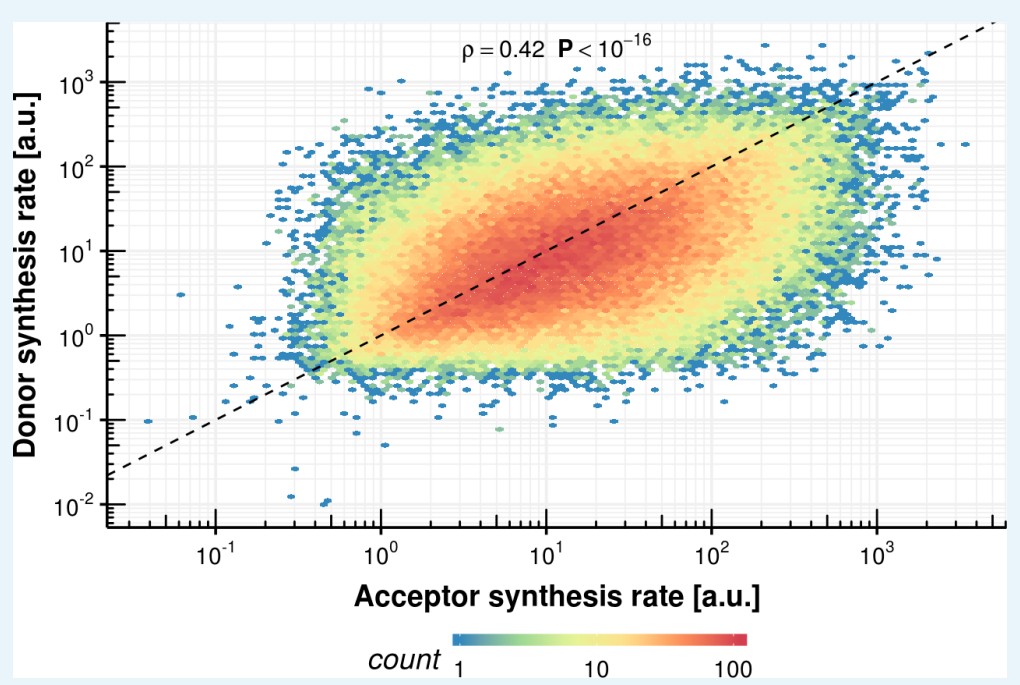

**Appendix 1—figure 14.** Donor synthesis rate vs. acceptor synthesis rate within the same intron.

DOI: https://doi.org/10.7554/eLife.45056.039

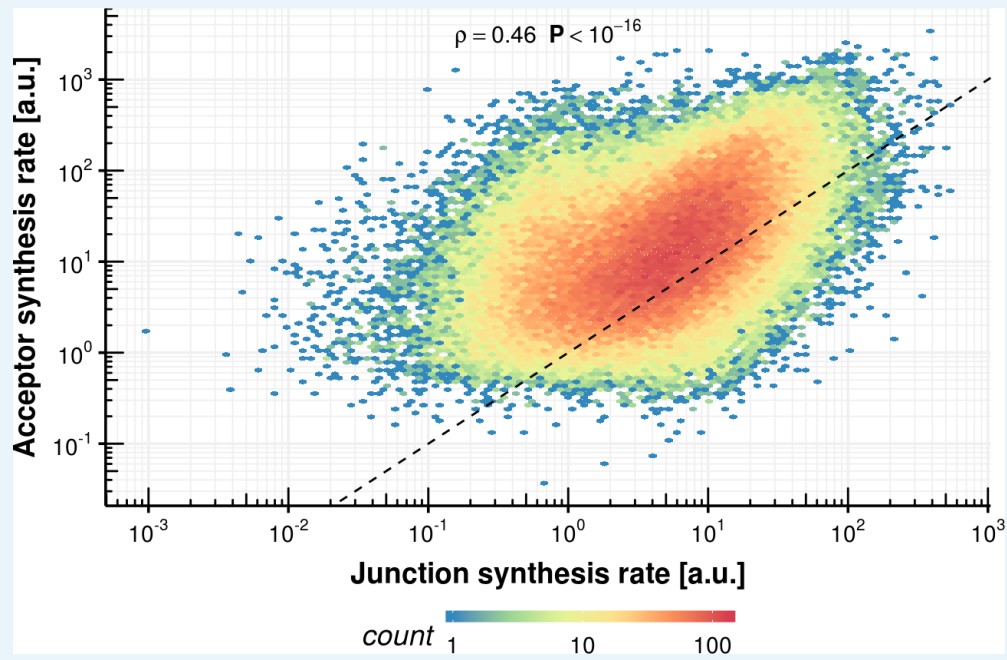

**Appendix 1—figure 15.** Acceptor synthesis rate vs. junction synthesis rate within the same intron.

DOI: https://doi.org/10.7554/eLife.45056.040

## 4 Quality of fits

### 4.1 Simulated vs. fitted rates

To assess whether our method is able to estimate synthesis and degradation rates from ground truth, we simulated counts based on the estimated distributions of synthesis rates, splicing half times and half-lives based on the experimental data. Based on simulated data, our method is an unbiased estimator of ground truth synthesis rates (*Appendix 1—figure 12*), splicing half-time and half-life (*Appendix 1—figure 13*) with high precision compared to the dynamic range.

### 4.2 Agreement of kinetic parameters donor, acceptor and junction model

Under the assumption that RNA polymerase II does not drop off during the transcription of one intron, the donor and acceptor synthesis rate are equal (*Appendix 1—figure 14*), whereas the junction synthesis rate reduced by the splicing yield (*Appendix 1—figure 15*).

### 4.3 Expected vs. observed counts

Comparisons of the predicted expected counts of the constant rate model with our observed experimental counts are shown in *Appendix 1—figures 16* (donor), *Appendix 1—figures 17* (acceptor) and *Appendix 1—figures 18* (junctions).

Normalization was based on spike-ins. Therefore, errors in spike-ins quantification possibly led to off centred the density plots. Indeed, the non-centred scatterplot showed deviation compatible with the bins they were created, {2 min}, {5 min, 10 min}, {15 min, 20 min}, {30 min, 60 min}.

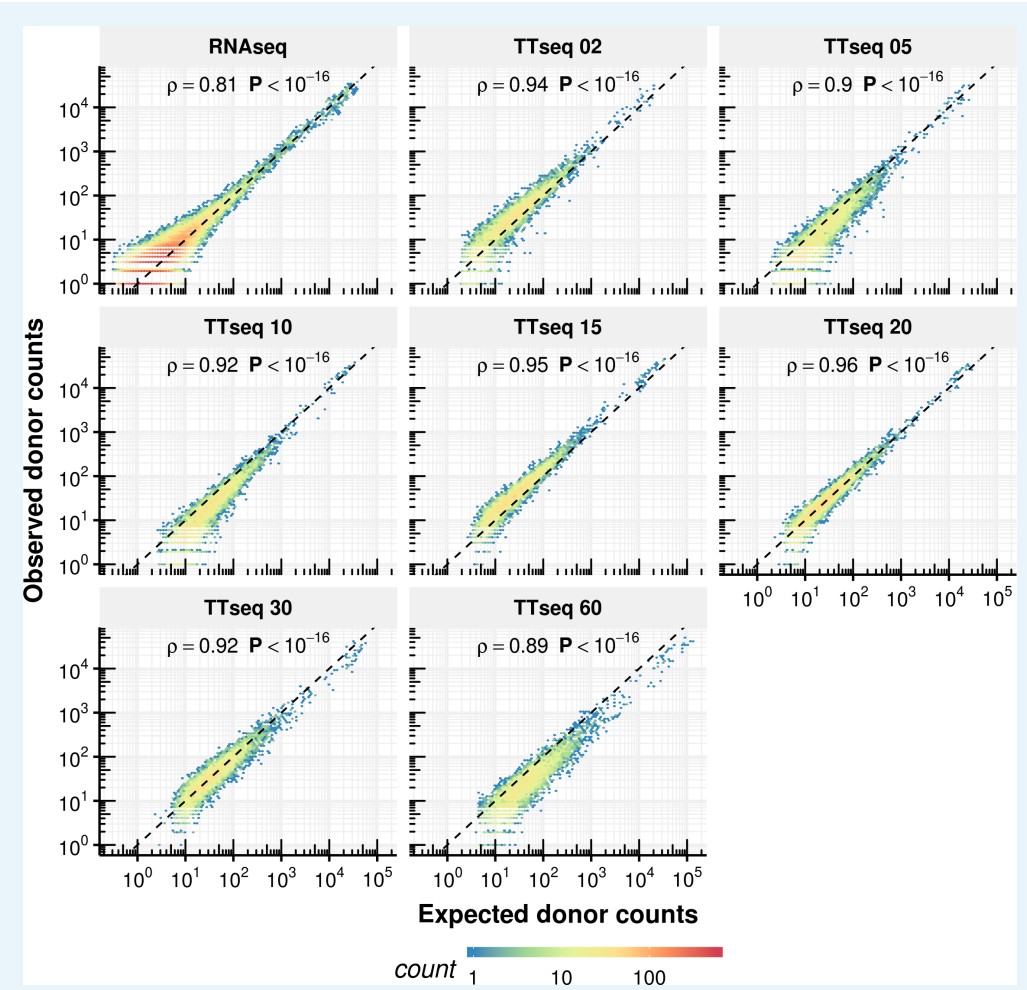

**Appendix 1—figure 16.** Comparison of expected counts of the constant rate model with observed experimental counts. Expected donor counts based on the constant rate model vs. the experimentally observed counts.

DOI: https://doi.org/10.7554/eLife.45056.041

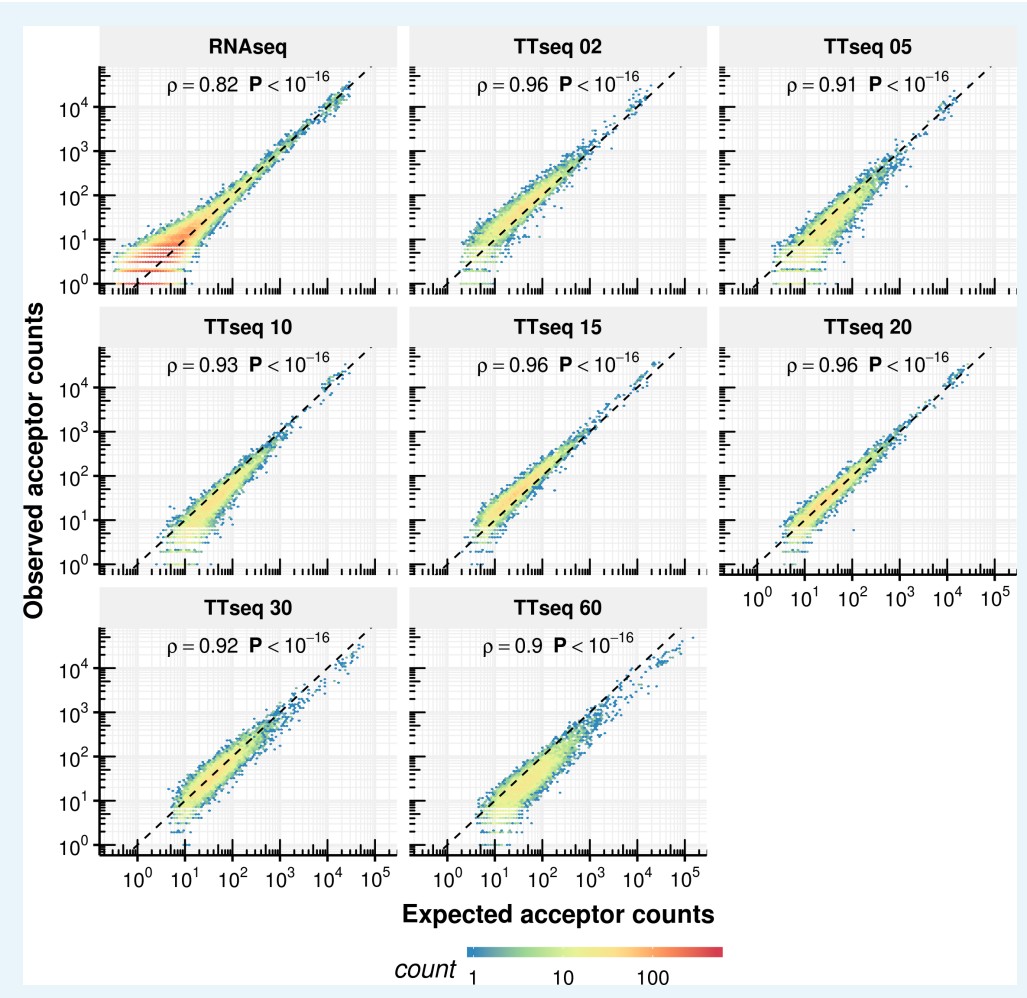

**Appendix 1—figure 17.** Comparison of expected counts of the constant rate model with observed experimental counts. Expected acceptor counts based on the constant rate model vs. the experimentally observed counts.

DOI: https://doi.org/10.7554/eLife.45056.042

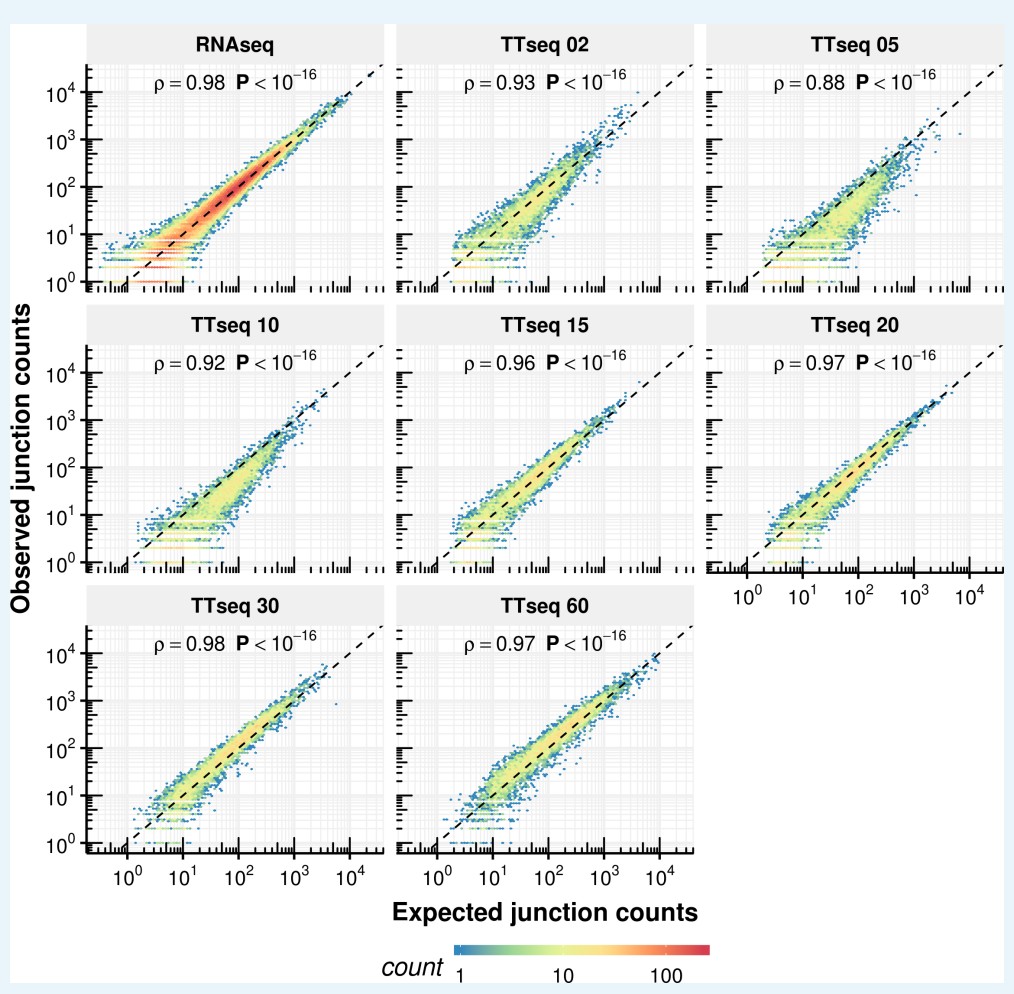

**Appendix 1—figure 18.** Comparison of expected counts of the constant rate model with observed experimental counts. Expected junction counts based on the constant rate model vs. the experimentally observed counts.

DOI: https://doi.org/10.7554/eLife.45056.043

## 4.4 Poor identifiability of splicing half-time and delay

In order to estimate splicing half-time and delay using our experimental data, we fitted a delay model. Although from theory we should be able estimate half splicing time and delay based on the delay model, the low numbers of data points and the noise of our data is too limiting to distinguish between linear or exponential start of the donor read population. This is supported by the bimodal distribution of the maximum likelihood estimate for the delay (*Appendix 1—figure 2*). If we fit a delay model to the donor site we would ideally expect that the donor delay correlates with intron length and the splicing half-time of the donor and acceptor sites correlate. Only for this analysis, we fitted the delay model also to the acceptor site. However, we found that the estimated delay or splicing half-time of the donor and acceptor sites of one junction correlates only little (*Appendix 1—figures 19–21*). Their sum however correlates much more, showing that we actually can measure the sum of delay and splicing half-time but not each one alone (*Appendix 1—figure 22*).

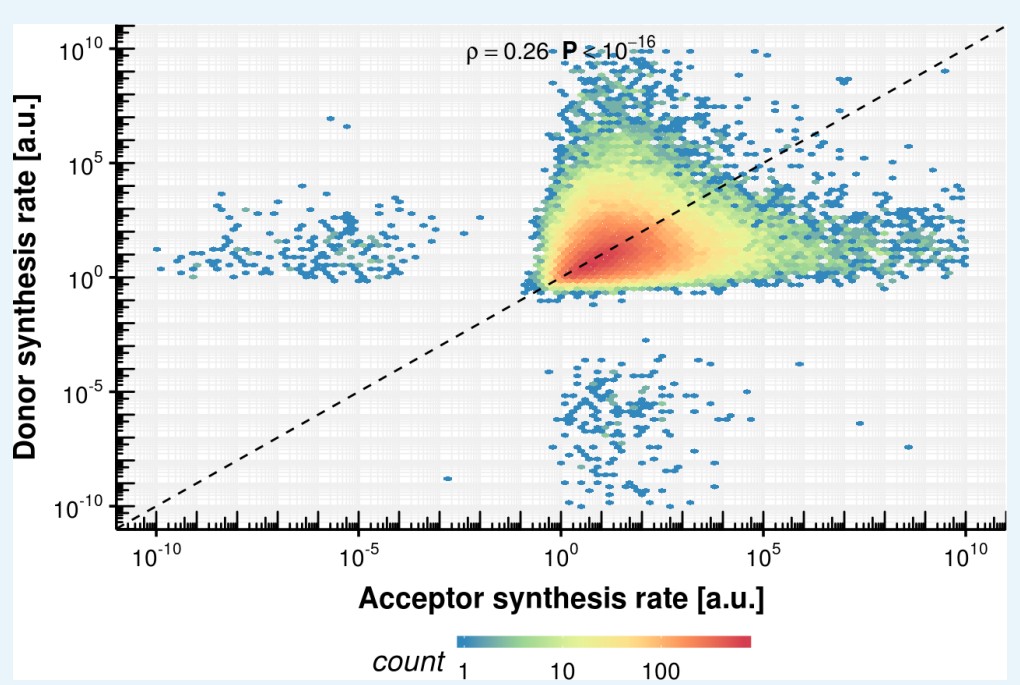

**Appendix 1—figure 19.** Acceptor vs. donor synthesis rate estimation based on the fixed delay model.

DOI: https://doi.org/10.7554/eLife.45056.044

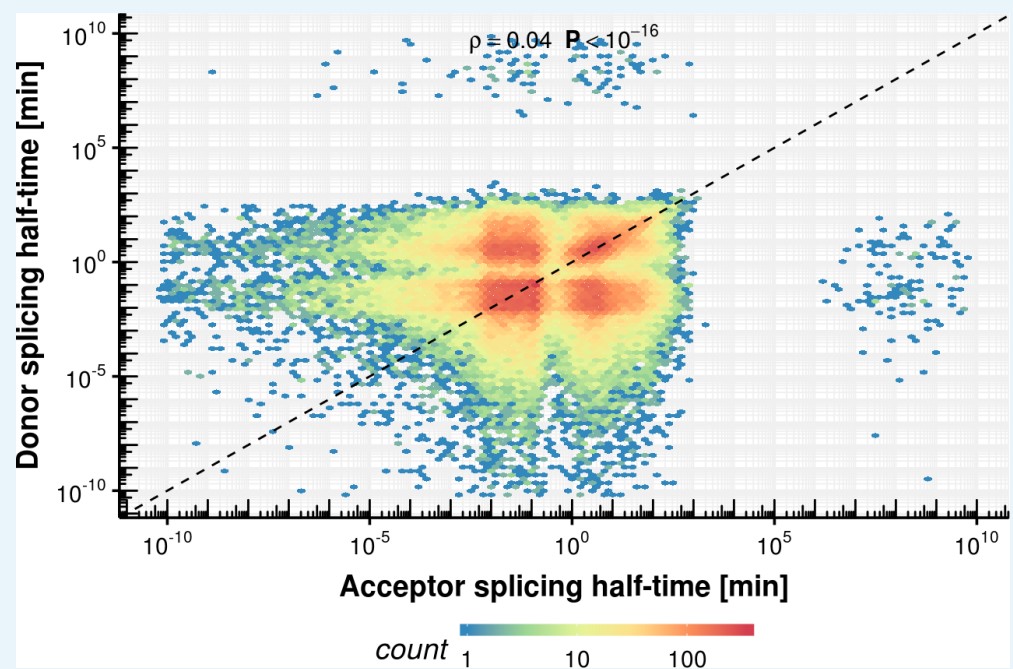

**Appendix 1—figure 20.** Acceptor vs. donor splicing half-time estimation based on the fixed delay model.

DOI: https://doi.org/10.7554/eLife.45056.045

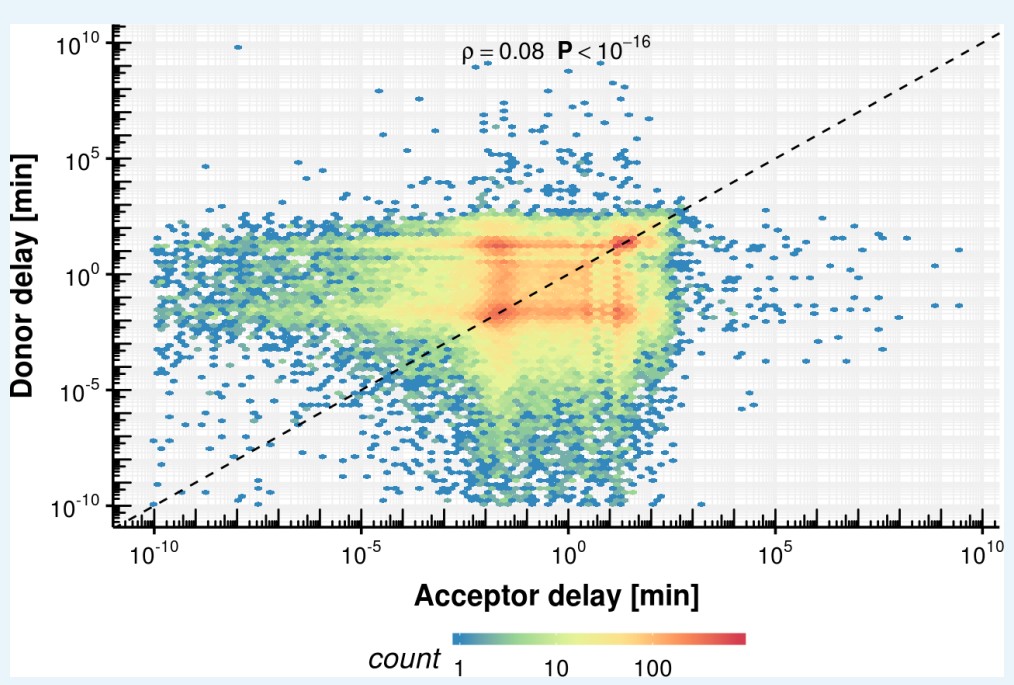

**Appendix 1—figure 21.** Acceptor vs. donor delay estimation based on the fixed delay model.
DOI: https://doi.org/10.7554/eLife.45056.046

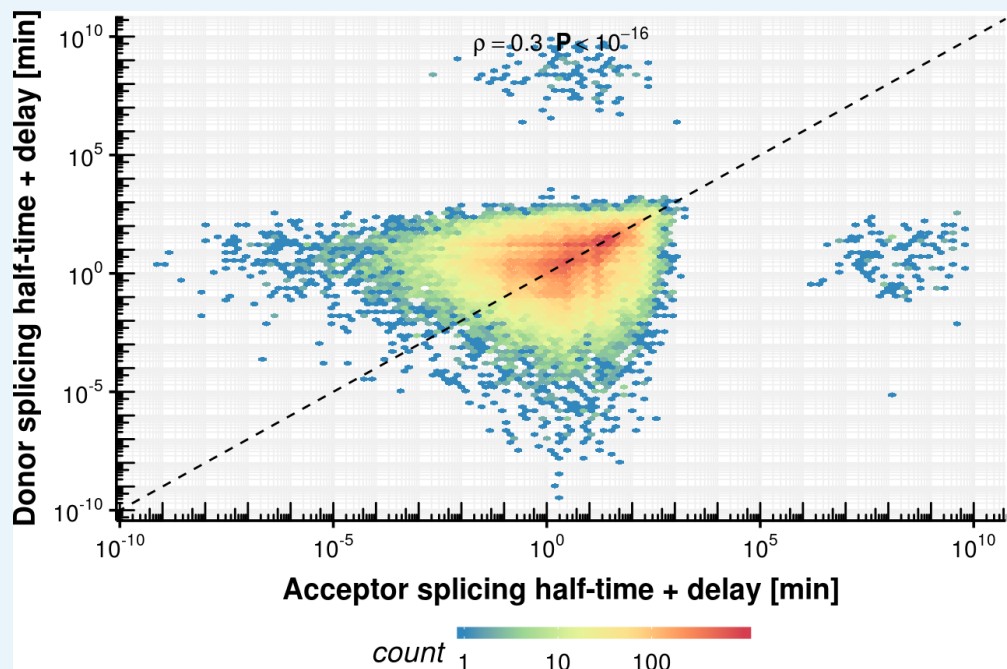

**Appendix 1—figure 22.** Acceptor vs. donor sum of splicing half-time and delay estimation based on the fixed delay model.
DOI: https://doi.org/10.7554/eLife.45056.047

## 5 Limits of the model

We simulated counts based on the constant rate model given a large range of log-uniform distributed bond synthesis rates ($10^{-5}$ – 100 1/cell/ min) and bond half-lives ($10^{-3}$ - $7 \times 10^{-6}$ min). Based on this data we investigated the typical lower bound of reads necessary in all samples to proper model the kinetics a bond (*Appendix 1—figure 23*). We find that using a

lower cut-off of 100 reads (after the 6<sup>th</sup> ventile) allows us to estimate the kinetics with a typical error below 100%.

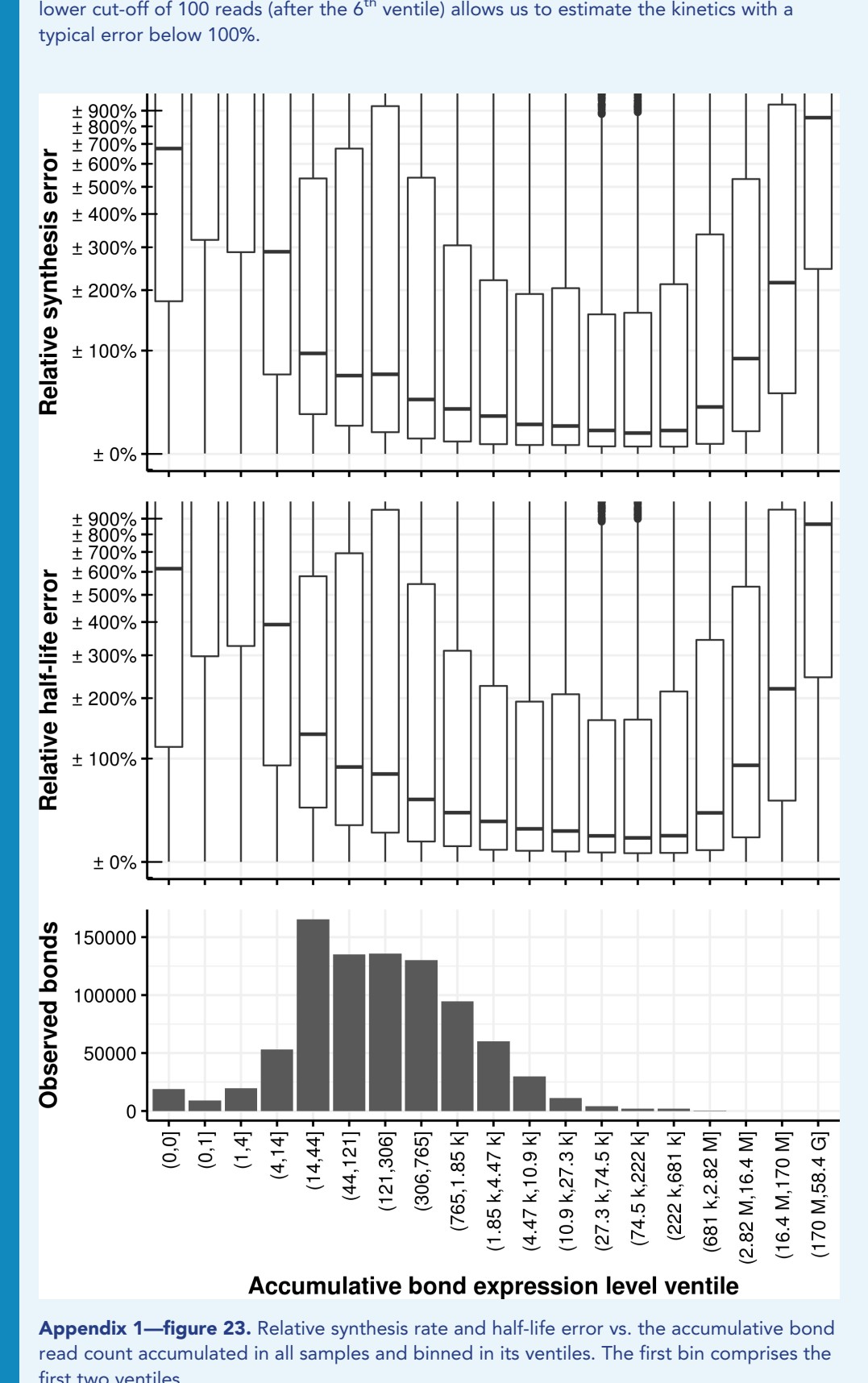

**Appendix 1—figure 23.** Relative synthesis rate and half-life error vs. the accumulative bond read count accumulated in all samples and binned in its ventiles. The first bin comprises the first two ventiles.

DOI: https://doi.org/10.7554/eLife.45056.048

To test the shortest and longest bond half-lives our experimental approach is able to capture and model properly we stratified the results also by bond half-live. We proceeded similarly with the bond synthesis rates. We found that most of our modelled data lies within a range where the median relative error of synthesis rates and half-lives is below 100% or 30% respectively (*Appendix 1—figure 24, 25*). We note that the errors given for the stratifications by bond half-life and synthesis are inflated for real world data, because the simulation is based on half-lives and synthesis rates that were drawn independently and more extreme than our real world data.

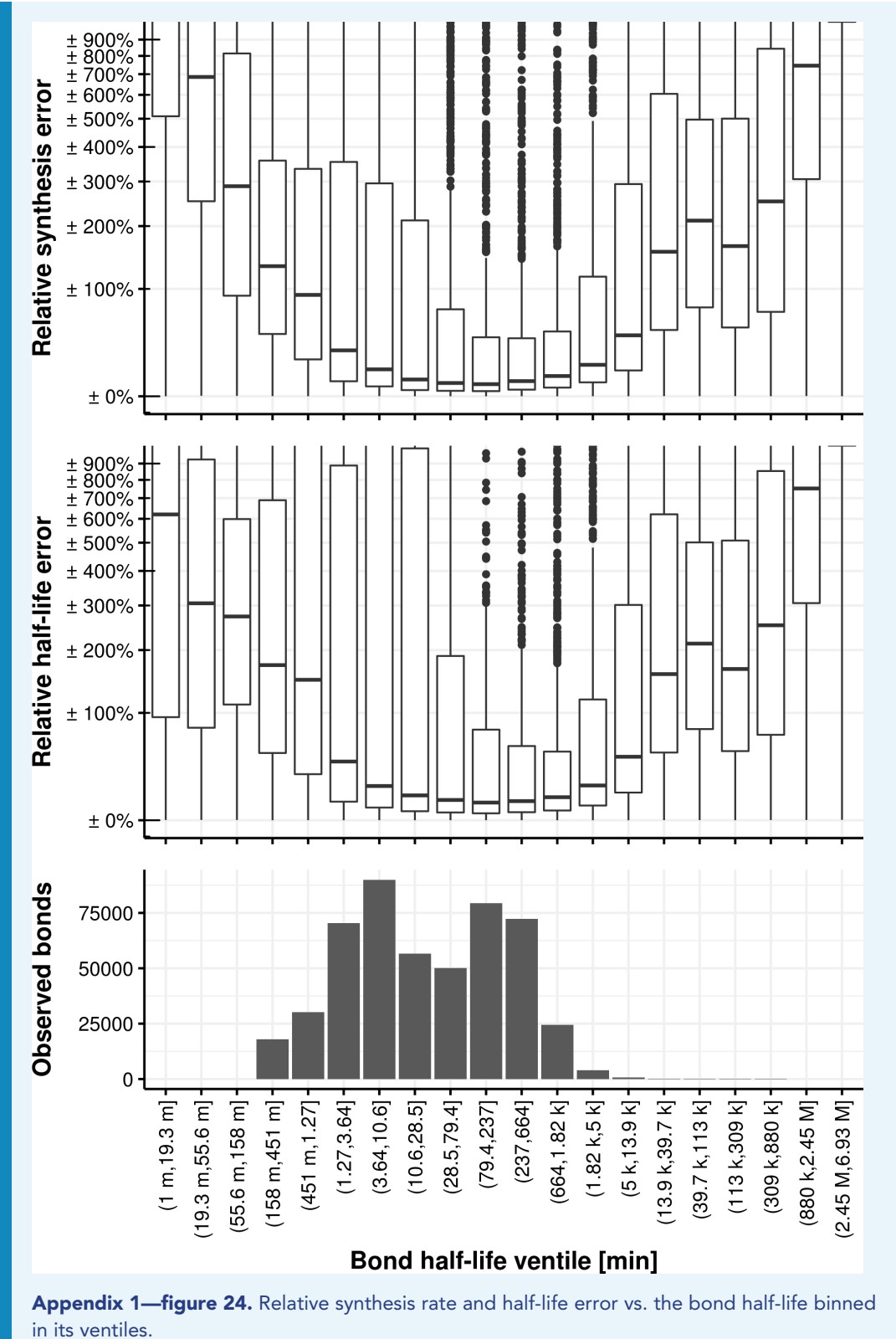

**Appendix 1—figure 24.** Relative synthesis rate and half-life error vs. the bond half-life binned in its ventiles.

DOI: https://doi.org/10.7554/eLife.45056.049

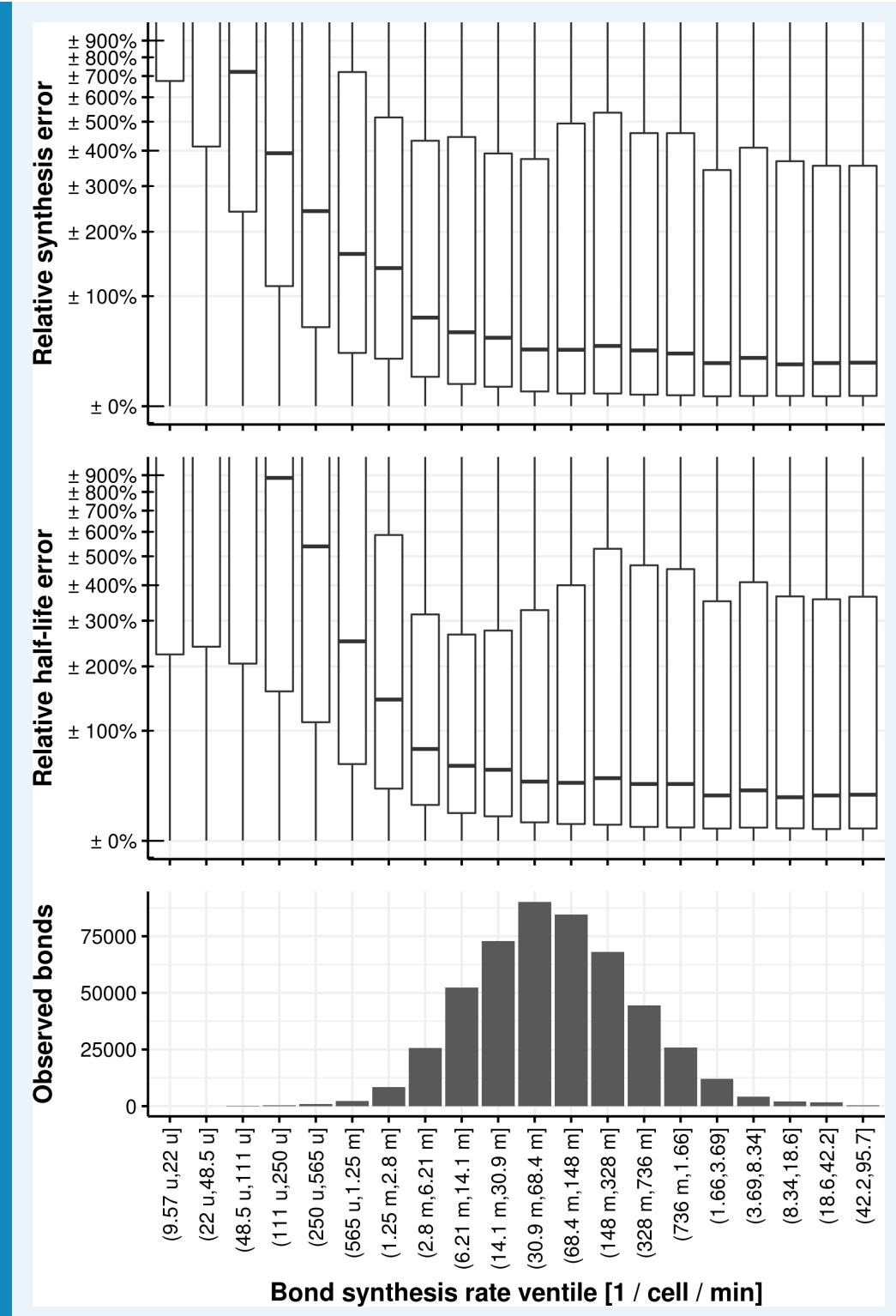

**Appendix 1—figure 25.** Relative synthesis rate and half-life error vs. the bond synthesis rate binned in its ventiles.

DOI: https://doi.org/10.7554/eLife.45056.050

## 6 4sU incorporation

For our analysis we did not consider the time until the labeled uracil 4sU gets available to the transcription machinery (by diffusion and import). The lag is a constant that is the same for all

genes. Note that this time has to be very short since labeled RNA was detected after 2 min labeling.

Assuming that only a fraction of all transcripts incorporates 4sU, this would lead to an underestimation of the synthesis rates (e.g. a labeling efficiency of 90% would results in a 10% underestimation of the rate), given that the labeling efficiency is constant over all samples. However, it does not affect estimation of the half-lives nor of the splicing yields. Indeed, the kinetic models are then modeling the kinetics of the labelled fraction of the RNA species rather than the overall RNA species. These kinetics differ from the kinetics of the overall RNA species only by different synthesis rates.

## 7 Splicing yield

Splicing yield is computed by independently estimating the synthesis rate of the precursor (using unsplit reads of the acceptor bond) and of the mature RNA (using split reads) and taking the ratios (Materials and methods). Due to estimation errors, these ratios may turn out to be greater than 1. Simulations (section 5) showed that cumulative read coverage across all samples as well half-life can lead to bias estimates of synthesis rates. We therefore investigated whether this could confound our observation that non-coding RNAs show lower yield than mRNAs. However, the higher yield of mRNA versus non-coding RNA was recapitulated when stratifying by these possible confounders (*Appendix 1—figures 26*, *27*).

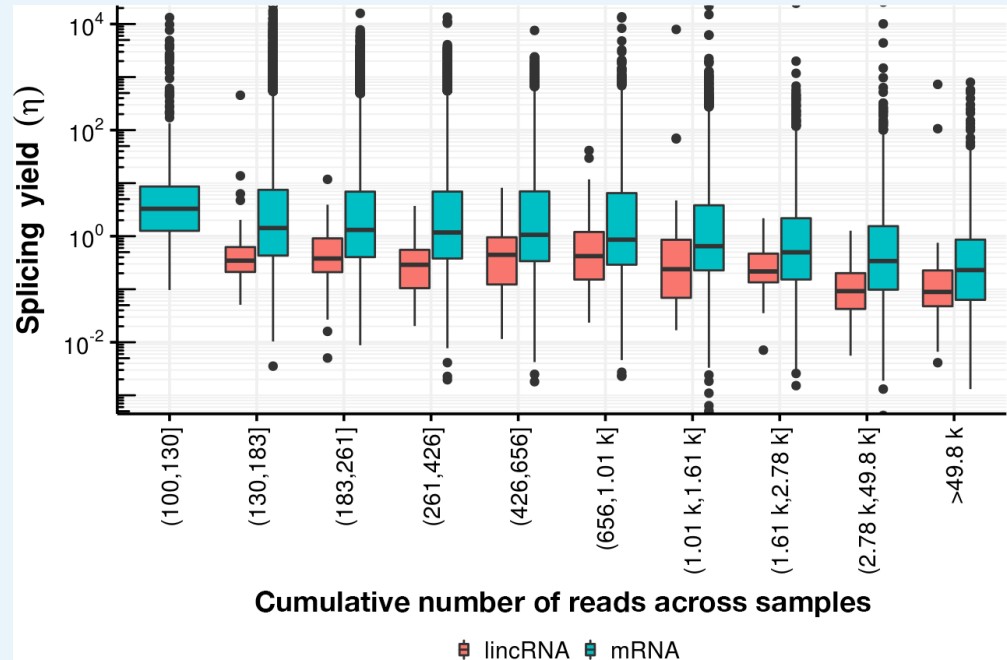

**Appendix 1—figure 26.** Splicing yield distribution (boxplot) of lincRNA (red) and mRNA (blue) stratified by bins of cumulative number of reads across all samples. Although yield correlates negatively with the cumulative number of reads, indicative of potential estimation bias, the mRNA yield remains higher than the lincRNA yield in every stratum.
DOI: https://doi.org/10.7554/eLife.45056.051

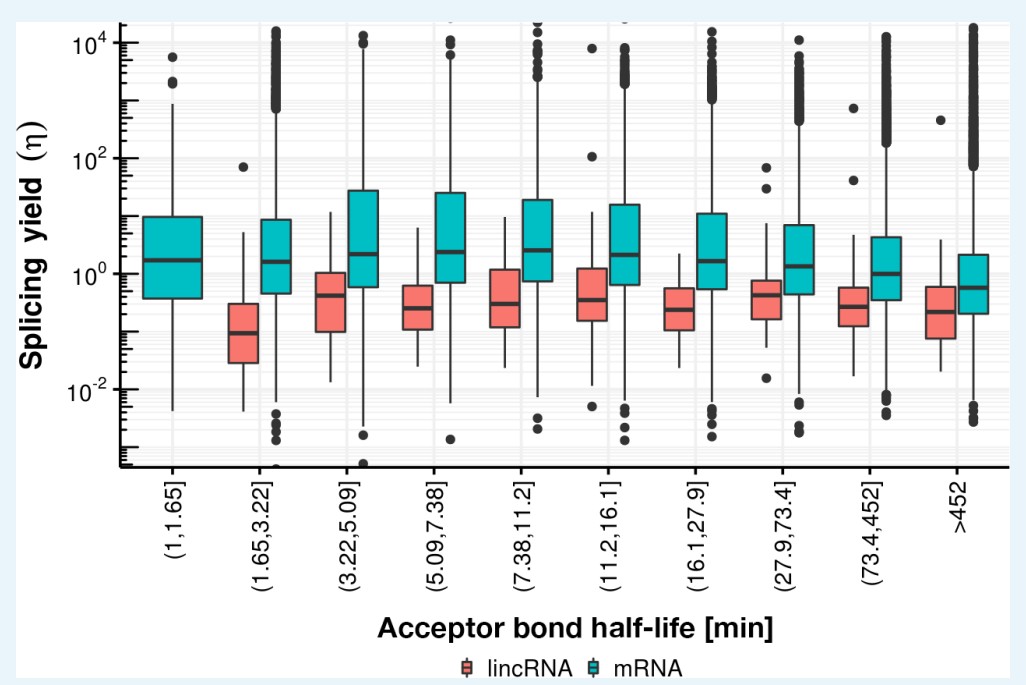

**Appendix 1—figure 27.** Splicing yield distribution (boxplot) of lincRNA (red) and mRNA (blue) stratified by bins of acceptor bond half-life. Although yield correlates negatively with acceptor bond half-life (for half-life larger than 16 min), indicative of potential estimation bias, the mRNA yield remains higher than the lincRNA yield in every stratum.
DOI: https://doi.org/10.7554/eLife.45056.052

