## [Decision Letter]

[Editors’ note: a previous version of this study was rejected after peer review, but the authors submitted for reconsideration. The first decision letter after peer review is shown below.]

Thank you for submitting your work entitled "Global two-step RNA splicing kinetics in human cells" for consideration by *eLife*. Your article has been reviewed by three peer reviewers, and the evaluation has been overseen by a Reviewing Editor and a Senior Editor. The reviewers have opted to remain anonymous.

Our decision has been reached after consultation between the reviewers. Based on these discussions and the individual reviews below, we regret to inform you that your work will not be considered further for publication in *eLife*.

There was some interest from the reviewers and editors in this new approach to kinetic analysis of splicing. However, the reviews point out that the study does not demonstrate how well the new model actually fits the observations, either on a global scale or at the level of individual genes. This would be a key step for a study presenting a new approach to such questions. A number of other major concerns were raised regarding the kinetic modeling.

Reviewer #1:

Wachutka et al. describes a method to assess splicing kinetics independently for each of the two steps of splicing, with downstream analyses to understand what may contribute to variation in the rates of splicing donor and acceptor cleavage during the 1st and 2nd steps of splicing respectively. I found that their approach to splicing kinetics – breaking it up into the individual steps rather than focusing on the entire intron as one piece – to be very interesting and innovative. The time course TT-seq labeling data presented in the manuscript is perfectly suited to answer these questions and their conclusions seem to support the idea that we can use this approach to gain insights on how splice sites are being recognized and efficiently used. Overall, I found the manuscript to be fairly well written, though I had a hard time following exactly what the authors were describing or their rationale in many places, as highlighted below. Specifically, I have the following concerns:

1) This manuscript is both technically and biologically dense, with the mathematical modeling and structurally based interpretation of the resulting kinetic data. I think it is necessary to provide more details about the methods in order to give readers an understanding of (1) the precise normalization, scaling, and modeling steps with regard to how to interpret each value being used for these steps, (2) the assumptions about kinetic relationships that underlie these models, and (3) how to assess the appropriateness and validity of this model (as described below).

2) I think it would be very useful to have an assessment of how well their model is fitting to their data, and the limitations of this model (and the data used). Ideally this would be provided by an orthogonal approach, but I sympathize that the experiments necessary for this may be logistically prohibitive. However, a computational assessment of this – either by simulations (accounting for how well their model picks up the ground truth after accounting for all the assumptions being made) or assessment of their model fit – is crucial. This would relate to why they choose a first order kinetic model, given that they seem to be modeling multiple potential first order processes (synthesis, bond cleavage, and RNA degradation) simultaneously.

3) One assumption that seems to be have been made in their kinetic modeling is that the site (either donor/acceptor) was created exactly at the beginning of labeling – while the experimentation fragmentation inherent to the TT-seq protocol allows us to know that the site being sequenced was indeed transcribed sometime during the labeling period, it may have been at the very beginning or the very end of this period. How can the authors account for this (i.e. should the assumption, given a uniform probability of transcribing during every finite time in the labeling period, be that the site was transcribed at the beginning or the middle of the labeling period on average)?

4) The authors make assumptions about a maximum polymerase elongation velocity for a few of their analyses, however studies have also shown substantial variability in polymerase elongation rates, across genes and potentially across regions within a gene. How might this affect their conclusions about intron length affecting the time to splicing or other observations?

5) The authors seem to suggest that previous studies may have overestimated splicing rates and not accounted for biases in these rates across the gene, however, this study seems to get similar splicing rate values. How do they account for this agreement of their measurements? Furthermore, how might they make these comparisons given that most other studies estimated rates while considered the entire intron entity rather than each individual splicing site? It seems like the junction formation rate may be the best metric for comparison, however, they do not focus very much analysis or time on this particular metric (which is indicative of completed splicing).

6) I really don't understand the splicing yield measurement and, particularly, how it differs or compares to a standard percent-spliced-in value that is commonly used to evaluate splicing patterns in RNA-seq datasets. What happens when only look at read abundance over these sites – is using the rates telling us anything different? Would be useful to put this in context of a metric that readers may be more used to thinking about in order to give it the proper interpretation.

Reviewer #2:

In this manuscript by Wachutka et al., the authors use modeling of 4sU pulse-chase RNA-sequencing data to infer rates of splicing across introns genome-wide. They then use sequence features and comparison to structural data to draw conclusions about first and second step transesterification rates. Experimentally, cells are exposed to a brief pulse of 4 thiouracil, and RNA made during the pulse will incorporate the 4sU and can be purified through biotin pull downs. So a junction (exon-intron, intron-exon, exon-exon) which is still around at some variable chase time provides kinetic information about the rates of removal of that particular junction. These removal processes could be splicing, non-sense mediated decay, or mRNA decay. Usually, the entire RNA is pulled down, giving the complete history of that particular transcript. The innovation from the Cramer lab (previously published TT-seq method) is to fragment this RNA to < 6kb, which removes some of the memory effect, although one is still analyzing RNA which was transcribed before the pulse. The present work is based entirely on analysis of this one data set in K562 cells. Overall, I have deep concerns about the analysis which are further exacerbated by the complete lack of perturbations or any effort at comparing to a ground truth measure of splicing kinetics. For these reasons, I don't think the paper is suited to *eLife*.

1) The following example illustrates the problem I have with this approach. Take the exon-intron donor junction reads as an example. Assume that the only way for this collection of reads to disappear is through the first-step transesterification step of splicing, which means the polymerase must at least reach a branchpoint, followed by the branchpoint nucleophilic attack. It is this second part which is the rate of the first step of splicing, not the time to reach the branchpoint. In addition, this elongation time to the branchpoint is not exponentially-distributed because RNAP is a processive enzyme. This time simply cannot be modeled as a single-step reaction but is rather a series of many exponential reactions. Moreover, even if one were to somehow include this information, meaning the local context of this donor junction, and then compute an elongation time to the branchpoint, it is impossible to know from this assay which branchpoint was chosen. This omission is a critical, and in my mind, fatal flaw of this analysis. I cannot see how one can possibly determine a rate for the first step of splicing without knowing which branchpoint was used. Going further, it would even be inappropriate in this analysis to use the annotated branchpoints because the authors are purporting to finds all sorts of unannotated introns and splicing events. Indeed, I think it is for this combination of reasons why so many labs are exerting great efforts into branchpoint sequencing and analysis.

2) Therefore, statements about "donor and acceptor cleavage times" are meaningless in the splicing context. It could be that the authors intend to report a 'donor cleavage time' which is indeed a combined measure of elongation, branchpoint selection, and the first chemical step of splicing, but I can't see what biochemical insight is gained from that time. There is an inclination that they are measuring something related to splicing based on the results in Figure 4B: donor cleavage time varies with departures from the consensus sequence. However, these changes don't seem to be consistent with the known biochemistry of 5' ss.

3) The theoretical treatments are in general lacking in rigor.

- Several papers have treated the kinetics of splicing and modeled kinetic data in single cells or even genome-wide (PMIDs: 25271374, 22022255, 25541195), yet the authors seem to be unaware of this work. Although one paper they do cite (Schmidt et al., 2011) indicates the importance of the correct theoretical treatment of elongation.

- There is no measure for the goodness of fits or examination of other models.

- They use a worm-like chain model based on RNA persistence length to compute the energy of looping. With the energy, they apply the Arrhenius equation to get a rate and compare this value to the donor cleavage time (Figure 3E). However, whether there is any trend in this data and whether this model explains that trend is difficult to determine.

4) I found the splicing yield part of the paper (Figure 7) to be the most interpretable of the results presented. Again, however, none of this was validated with any more quantitative measure such as digital droplet PCR.

In summary, I see this analysis (with the underlying parameterization) as a reasonably good way to explain 'donor cleavage time' and 'acceptor cleavage time' as measured in this assay. And there may be some utility in such metrics. My overall difficulty is that I don't think this model parameterized in such a way is telling us much about splicing and transcription in a way that could be validated or interpreted in other assays. So I see this paper as a bioinformatic exercise lacking in biochemical restraint or biophysical insight.

Reviewer #3:

The authors present human genome-wide estimates of cleavage times at a single splice site resolution (donor and acceptor) along with estimates of synthesis and mRNA half-life using transient transcriptome sequencing (TTseq) which uses pulse labeling of 4SU followed by RNA fragmentation prior to purification of the labelled RNA – this reduces the 5' bias. They also assess the effect of different features to splicing time, such as intron position, length and sequence, and then attempt to develop correlations between structural aspects of spliceosomal interactions and kinetics.

Given that the optimization methods used for estimating synthesis/formation and cleavage/degradation of the splice sites/junction is a local algorithm, is 10 random initialization enough to be confident that the global minimum has been reach?

A key assumption for the model fitting is that 100% of the label is incorporated. The authors should explicitly explore how lower levels of labelling (e.g. 90, 50, 10%) affect their results and conclusions.

Some comparison and correlation of synthesis rate estimated has been done separately for different donor sites and different acceptor sites (Figure 2—figure supplement 2B), but as mentioned, these should also be consistent for donor and acceptor of the same gene; however, donor synthesis rate vs acceptor synthesis rate is not shown.

Model fits should be shown for individual genes, highlighting different classes: e.g. single vs multi-intron, highly expressed vs low expressed.

Kinetic language is not precise and causes confusion: For example "Cleavage times" should be referred to as "half cleavage times" or possibly "cleavage half-times". They do not refer to completion times, or characteristic times.

"Cleavage Rates in the range of minutes" is also imprecise. They seem to refer to cleavage half-life times being in in the minute range. More precise would be to report the actual cleavage rate constants, in addition to the cleavage half-life times. ("Rate" = rate constant x abundance).

The part on the single nucleotide model methods would benefit of a more detailed description. It seems to be a statistical model based on the previously derived cleavage time and consensus splice site sequence but the methods is not very clear to me.

[Editors’ note: what now follows is the decision letter after the authors submitted for further consideration.]

Thank you for submitting your article "Global donor and acceptor splicing site kinetics in human cells" for consideration by *eLife*. Your article has been reviewed by three peer reviewers, and the evaluation has been overseen by Douglas Black as Reviewing Editor and James Manley as the Senior Editor. The reviewers have opted to remain anonymous.

The reviewers have discussed the reviews with one another and the Reviewing Editor has drafted this decision to help you prepare a revised submission. We believe the paper is in principle more appropriate as a "Tools and Resources" paper than as a Research Article, and therefore request that a revised manuscript be submitted in that category.

Summary:

Gagneur and colleagues describe a new approach to measuring rates of intron excision in vivo that combines a metabolic labeling method with RNAseq and mathematical modeling. Focusing the analysis on the appearance and disappearance of reads for individual exon-intron, intron-exon, and exon-exon junctions, the authors show that they can extract rate information for the chemical steps at particular splice sites genomewide. They then correlate these rates with a variety of features in the pre-mRNA including consensus sequence match, intron length, and local sequence context. Elucidating the determinants of splicing kinetics is important to understanding gene regulatory programs in metazoans and has been understudied. A new method for measuring splicing rates is thus potentially very valuable.

The reviewers all felt that the paper was much improved from the previously version, which we had declined. There was universal interest in the method of combining metabolic labeling with analyses of single phosphodiester junctions to extract individual rates of decay and formation for particular splice sites and exon-exon junctions. This method presents an original approach to measuring splicing kinetics in vivo, and a potentially valuable advance for the field. As such, the paper could make a good addition to *eLife*, if it can be appropriately revised. The primary interest of the paper is in the technology and analytical techniques. New biological or mechanistic insights arising from the study are limited, with values for intron excision rates similar to previous studies. It is very important that the method be well described and that the data not be overinterpreted. While improved, the paper is still very dense and difficult to follow. The reviewers raised a number of issues and points of needed clarification that must be addressed.

Essential revisions:

1) The reviewers remain unconvinced that the measured decay and formation rates of phosphodiester linkages can be directly attributed to individual recognition steps and enzymatic rates. By the time a chemical step causes the loss or gain of a junction read, dozens of events have occurred that could contribute to the rate of that step. For a 5' splice site these include initial recognition by U1, transcriptional elongation/pausing in the downstream intron, branchpoint recognition (which cannot be measured here), myriad spliceosome assembly steps, and finally catalysis. While the effects of splice site mutations agree with the known interaction of the 5' splice site by the U1 snRNP, other moieties will be interacting with these residues, and there is no evidence that the U1 interaction is determinative of the rate of cleavage. It is very simplistic to claim this is so. The rate of exon-intron junction loss will be the aggregate function of many different rates that likely vary from site to site and are not being measured. It should be noted that U1 is not involved in catalytic steps, only initial recognition. To say that "we developed a computational approach that models rates of RNA metabolism at the level of single phosphodiester bonds" is misleading. The authors need to examine their interpretations of their data and be careful not to claim that the rates they are observing can be broken down into steps that are not being measured.

2) Subsection “New and alternative splice sites”, last paragraph: The authors reported numbers (177,322 and 164,533) of non-split *reads* mapping to exon-intron boundaries (5' splice sites) sum to the total number of putative introns reported in the paragraph above (341,855, identified by split reads across an exon-exon boundary). This seems wrong, or did they throw out all examples of alternative splicing? They don't report numbers for intron-exon boundaries (3' splice sites), but farther down in the same paragraph they treat the read numbers as if these sites are included. In that later discussion, the authors state that of the split reads, "18% contained a non-annotated donor site, 22% a non-annotated acceptor site and another 38% represented new putative splice sites". What is the difference between a non-annotated donor or acceptor site and a new putative splice site? Furthermore, if aiming to differentiate between annotated and non-annotated sites, wouldn't it make more sense to start with the percentage of *split* reads mapping to non-annotated regions, where all the sites are likely unannotated, and then discuss new previously unannotated sites in GENCODE?

3) The authors indicate that they are fitting the model independently to every donor-acceptor combination that they observe. If so, what is causing the differences in the numbers of donor (162,134), acceptor (177,543), and junction (156,825) bonds that they are able to model? Don't they need sufficient read coverage across all three to confidently estimate the synthesis/cleavage rates on a per-junction level? Or does this include some cases of alternative splice site pairings?

4) The use of simulations to estimate the robustness of the model fitting, across both the first order kinetic model and alternative models is a helpful addition to the paper. It should be highlighted more in the main text how this supports the validity of the method. An assessment should also be provided for how the method performs and where it doesn't work well. For example, the authors indicate that "the robustness of our approach… consistent with high accuracy of the fitting procedure on simulated data." However, all models or experimental systems are underpowered in certain cases. The authors should include a short discussion of where their model falls short (ultra-fast rates, length considerations, etc.) and thus more fully inform the reader about the biological interpretations that can be drawn from the data. Simulations are often the best way to push the model and test these extreme cases.

5) In the section entitled "Intron length constrains co-transcriptional splicing times." It is not clear what the authors mean by "co-transcriptional". The authors describe how donor bond half-life increases with intron length, presumably indicating that donor bond half-life is dependent on the time to reach the branchpoint and acceptor site. This would imply that the first step of splicing occurs soon after the acceptor site is transcribed and available. If so, then why is the acceptor bond half-life so long? A median acceptor half-life of 4 min indicates that the Polymerase is on average 16kb away by the time the second step of splicing occurs. Do they propose that the first step is close in time to the second step, with a lag between transcription of the branchpoint and the loss of the donor? Or are the two chemical steps separated by 4 minutes? This seems unlikely in that it implies that spliceosome assembly up to C complex formation is very fast compared to the transition from the first to the second catalytic steps. If the authors think that donor loss occurs closer in time to the second step – well after the appearance of the branchpoint – then the definition of "co-transcriptional" that involves immediate splicing upon transcription/availability of the acceptor site doesn't seem to hold. All of this needs better explanation.

6) It is not clear how the splicing yield calculation can produce a value of > 1, with the median splicing yield reported for mRNAs as 1.2. Does this mean that the measured levels of precursor mRNAs are consistently underestimated? If it's not bounded as described (0-1), the splicing yields are difficult to assess as it seems one or more of the measured values is incorrect.

---

## [Author Response]

[Editors’ note: the author responses to the first round of peer review follow.]

Reviewer #1:[…] 1) This manuscript is both technically and biologically dense, with the mathematical modeling and structurally based interpretation of the resulting kinetic data. I think it is necessary to provide more details about the methods in order to give readers an understanding of (1) the precise normalization, scaling, and modeling steps with regard to how to interpret each value being used for these steps, (2) the assumptions about kinetic relationships that underlie these models, and (3) how to assess the appropriateness and validity of this model (as described below).

We thank the reviewer for the comment. (1) We have described the spike-in normalization procedure in main text. (2) We provided more details about our kinetic modeling in the Appendix (subsections “Notations, definitions, and relation to splicing quantities found in literature” and “Kinetic models”). (3) To assess the appropriateness and validity of our model, we compared our model with other two alternative, more complex splicing models (Appendix subsection “Comparison of the models”). As a result, our model (two free parameters constant rate) estimates synthesis, splicing, and degradation rate in a more accurate way than the other models by at least one order of magnitude (Appendix subsection “Parameter estimation”, Appendix—figures 13). When we applied the model to our experimental data, the results were similar in all models (Appendix subsection “Experimental data”, Appendix—figures 8-11). To assess the appropriateness and validity of our model we added a subsection in the Appendix (“Quality of fits”) where we tested our quality of fitted rates (see answer below).

2) I think it would be very useful to have an assessment of how well their model is fitting to their data, and the limitations of this model (and the data used). Ideally this would be provided by an orthogonal approach, but I sympathize that the experiments necessary for this may be logistically prohibitive. However, a computational assessment of this – either by simulations (accounting for how well their model picks up the ground truth after accounting for all the assumptions being made) or assessment of their model fit – is crucial. This would relate to why they choose a first order kinetic model, given that they seem to be modeling multiple potential first order processes (synthesis, bond cleavage, and RNA degradation) simultaneously.

We thank the reviewer for the insightful comment. To assess whether our model estimates kinetic rates from ground truth, we have now generated simulated datasets under the assumptions of the first order kinetic models, using realistic ranges of parameters to assess how accurate the estimation of the parameters were. Our estimated rates recovered accurately the simulated ones (Appendix subsection “,Simulated vs. fitted rates”, Appendix—figures 12-13). Comparison of the expected versus observed counts of donor (Appendix—figure 16), acceptor (Appendix—figure 17), junctions (Appendix—figure 18) demonstrates the precision of our model. Moreover, we have also assessed how robust our fitting is to deviations to the first order kinetics assumption by simulated data with different kinetics (delayed process and coupled ODEs). This showed that i) these more complex kinetics models are difficult to fit with our experimental design (Appendix—figures 1-3) and ii) the 1st order ODEs can be robustly fitted to such data yielding parameters that can be interpreted (Appendix—figures 4-11).

3) One assumption that seems to be have been made in their kinetic modeling is that the site (either donor/acceptor) was created exactly at the beginning of labeling – while the experimentation fragmentation inherent to the TT-seq protocol allows us to know that the site being sequenced was indeed transcribed sometime during the labeling period, it may have been at the very beginning or the very end of this period. How can the authors account for this (i.e. should the assumption, given a uniform probability of transcribing during every finite time in the labeling period, be that the site was transcribed at the beginning or the middle of the labeling period on average)?

No, a first order kinetic model does not assume that each molecule is created at the beginning of the time series but that they are continuously synthesized and degraded during the time series. Related to this point, our model did not consider time until 4sU gets available to the transcription machinery (by diffusion and import). The lag is a constant that is the same for all genes. Note that this time has to be very short since labeled RNA was detected after 2 min labeling. We now mention this point in the Appendix.

4) The authors make assumptions about a maximum polymerase elongation velocity for a few of their analyses, however studies have also shown substantial variability in polymerase elongation rates, across genes and potentially across regions within a gene. How might this affect their conclusions about intron length affecting the time to splicing or other observations?

We agree with the reviewer and we are aware that Pol II elongation rate is gene specific and it shows variability across the regions within the genes. However, precise calculation of Pol II elongation rate would require the integration of TT-seq data with local Pol II occupancy profiling which is difficult and would justify a manuscript on its own. It has been shown that Pol II elongation velocity varies from 0.5 to 4kb/min (Gressel et al., 2017, Jonkers et al., 2014, Ardehali and Lis, 2009), with an average elongation velocity of 2.3kb/min (Gressel et al., 2017, Fuchs et al., 2014; Jonkers et al., 2014; Saponaro et al., 2014, Veloso et al., 2014). Pol II accelerates in the gene body up to 4kb/min, indicating a faster transcription of introns (Gressel et al., 2017, Jonkers et al., 2014). Because of this, we reported an overall trend assuming a 4kb/min Pol II elongation rate in introns. Investigation of intron-specific relationship of Pol II elongation rate and splicing could be an interesting future research direction. We now comment on this in the Discussion.

5) The authors seem to suggest that previous studies may have overestimated splicing rates and not accounted for biases in these rates across the gene, however, this study seems to get similar splicing rate values. How do they account for this agreement of their measurements? Furthermore, how might they make these comparisons given that most other studies estimated rates while considered the entire intron entity rather than each individual splicing site? It seems like the junction formation rate may be the best metric for comparison, however, they do not focus very much analysis or time on this particular metric (which is indicative of completed splicing).

The closest dataset to ours is the one of Mukherjee et al., 2016, which uses 4sU-seq in human cells. Mukherjee et al., 2016, did not compute splicing kinetics of individual introns but only classified them as fast, medium or slow, by clustering the time series data. We therefore could not directly compare the splicing rates of individual introns. Mukherjee et al., 2016, provided gene level splicing rates, which were computed by fitting first order kinetics on the complete precursor RNA, using RSEM to estimate mature and precursor RNA abundance. To compare our results with their results, we averaged donor and acceptor cleavage times within major isoforms as these reads are specific to the precursor. While our measures of half-life correlate well across genes, those of synthesis rates and splicing rates were more modest (Figure 2—figure supplement 1D). The orders of magnitudes match. Junction formation rate estimation is indicative of completed splicing but is not the most informative metric to estimate the time needed to splice donor and acceptor splice-site (cleavage time). Assuming perfect yield. i.e. that every precursor RNA leads to a mature RNA, the junction formation rate equals the precursor synthesis rate. It is thus roughly a constant across one transcript. We now explain this in the Appendix (Appendix 1—table 1).

6) I really don't understand the splicing yield measurement and, particularly, how it differs or compares to a standard percent-spliced-in value that is commonly used to evaluate splicing patterns in RNA-seq datasets. What happens when only look at read abundance over these sites – is using the rates telling us anything different? Would be useful to put this in context of a metric that readers may be more used to thinking about in order to give it the proper interpretation.

We apologize for not being clear enough in the text and we thank the reviewer to point it out. In Appendix we now relate our metric to existing ones, including Psi. In Appendix subsection “Notations, definitions, and relation to splicing quantities found in literature”, we added more detailed explanation of how we defined splicing quantities and of the difference between percentage-spliced-in and splicing yield. Splicing yield cannot be computed from steady-state RNA-seq data because synthesis and degradation are entangle in steady-state data. We have also edited the text extensively to make sure this is now easier to understand and we trust the reviewer agrees.

Reviewer #2:In this manuscript by Wachutka et al., the authors use modeling of 4sU pulse-chase RNA-sequencing data to infer rates of splicing across introns genome-wide. They then use sequence features and comparison to structural data to draw conclusions about first and second step transesterification rates. Experimentally, cells are exposed to a brief pulse of 4 thiouracil, and RNA made during the pulse will incorporate the 4sU and can be purified through biotin pull downs. So a junction (exon-intron, intron-exon, exon-exon) which is still around at some variable chase time provides kinetic information about the rates of removal of that particular junction. These removal processes could be splicing, non-sense mediated decay, or mRNA decay. Usually, the entire RNA is pulled down, giving the complete history of that particular transcript. The innovation from the Cramer lab (previously published TT-seq method) is to fragment this RNA to < 6kb, which removes some of the memory effect, although one is still analyzing RNA which was transcribed before the pulse. The present work is based entirely on analysis of this one data set in K562 cells. Overall, I have deep concerns about the analysis which are further exacerbated by the complete lack of perturbations or any effort at comparing to a ground truth measure of splicing kinetics. For these reasons, I don't think the paper is suited to eLife.

We thank the reviewer for acknowledging the advantages of TT-seq. However, we wish to point out that there is no reason to believe that we do analyze RNA that was transcribed before the pulse. There is no cross-contamination of labeled RNA fragments with non-labeled RNA, and the RNA fragment length is much shorter than the distance of polymerase progression during the labeling time. Thus the assumption of the reviewer is not substantiated. We also feel there may be a misunderstanding: we are not conducting a pulse-chase experiment but rather apply labeling pulses that vary in length. Analysis of in vivo pulse-chase experiments is made difficult due to recycling of labelled uracil after RNA degradation. Our approach does not have this limitation. We think it is the best currently available experimental protocol to infer in vivo rates that are tightly coupled to splicing events. Since we also provide splicing yields and can related our in vivo data to in vitro structural data, we think our manuscript is novel and timely and highly suitable for *eLife*. We kindly ask the reviewer to reconsider his/her opinion in the light of our additional analysis and modeling.

1) The following example illustrates the problem I have with this approach. Take the exon-intron donor junction reads as an example. Assume that the only way for this collection of reads to disappear is through the first-step transesterification step of splicing, which means the polymerase must at least reach a branchpoint, followed by the branchpoint nucleophilic attack. It is this second part which is the rate of the first step of splicing, not the time to reach the branchpoint.

We apologize if we did not explain it clear enough in the text: what we define as “donor cleavage time” is the sum of two different processes, elongation time to the branchpoint and first transesterification step, which together will determine the time needed to cleave the phosphodiester bond of the donor splice site. We had actually used this extensively in our manuscript to provide insights into the co-transcriptional nature of the process. Because of this, we could investigate the role of intron length, and therefore elongation time, in the paragraph of the paper “Intron length constrains co-transcriptional splicing times” where we explicitly stated that donor splicing kinetics are regulated by the time the polymerase needs to transcribe the intron. To avoid misunderstanding, we now describe the processes determinant of the cleavage time of the donor site and of the acceptor site in the first paragraph of the section “Kinetic modeling”. We have also changed the manuscript title, which may have been too short and thus misleading. In summary, the kinetic insights are relevant to understanding splicing and its co-transcriptional nature in vivo.

In addition, this elongation time to the branchpoint is not exponentially-distributed because RNAP is a processive enzyme. This time simply cannot be modeled as a single-step reaction but is rather a series of many exponential reactions.

More complex models could be devised but would need to be fitted to the data. We have now investigated alternative models (Appendix subsections “Kinetic models” and “Comparison of the models”). One is a delay model for modeling the time of intron transcription. Another one is a coupled model for the junction reads that couple two exponential processes, namely donor-site / acceptor-site cleavage and mature RNA degradation. When we compared our model to the other more complex ones (Appendix subsection “Parameter estimation”), we found that the more complex models typically offer no advantage over our simple model because our experimental data is too limited to substantially differentiate between the models. Based on simulated data we show that the simple constant rate model deviates only slightly from the results if we applied more complex models and the differences are typically below our estimated experimental error. Furthermore, the more complex models are less precise to estimate and therefore the overall precision of our analysis would decrease. Systematic errors between the different models only occur if we compare kinetic rates between different models as done in our splicing yield calculation. Therefore, this is the only case where we apply the more complex models. Taken together, although the process of co-transcriptional splicing is complex and should ideally be described with more detailed models in the future, experimental data are limiting what can be modeled and our model is robust and does reflect the process in a meaningful way that enables us to draw simple conclusions.

Moreover, even if one were to somehow include this information, meaning the local context of this donor junction, and then compute an elongation time to the branchpoint, it is impossible to know from this assay which branchpoint was chosen. This omission is a critical, and in my mind, fatal flaw of this analysis. I cannot see how one can possibly determine a rate for the first step of splicing without knowing which branchpoint was used. Going further, it would even be inappropriate in this analysis to use the annotated branchpoints because the authors are purporting to finds all sorts of unannotated introns and splicing events. Indeed, I think it is for this combination of reasons why so many labs are exerting great efforts into branchpoint sequencing and analysis.

We have now clarified that we are not estimating directly the rates of the individual transesterification step, which would indeed lead to these complications. Also, we have used branchpoint predictions later in the manuscript for predicting the rates from sequence (Figure 4, 5), which is a practical way to move forward as maps of branchpoints are still lacking. These turned out to have very reasonable predictive power. Also, we would like to underscore that many introns contain multiple branchpoints but their distance from the acceptor site is narrowly distributed, typically from 18 to 45 nt from acceptor site (Mercer et al., 2015). Taken together, our conclusions are not compromised by the fact that multiple branchpoints may contribute.

) Therefore, statements about "donor and acceptor cleavage times" are meaningless in the splicing context. It could be that the authors intend to report a 'donor cleavage time' which is indeed a combined measure of elongation, branchpoint selection, and the first chemical step of splicing, but I can't see what biochemical insight is gained from that time.

Please compare our answers above. Indeed, we intentionally report donor cleavage time as we think that this is a quantity that can be defined and directly measured from such genome-wide data. Delineating it into its biochemical components (elongation, branchpoint selection, and the first chemical step of splicing) would require strong modeling assumptions and lead to overfitting of the data. We show with multiple analyses that these times do contain information and are thus meaningful. We agree however that in the future other sequencing protocols may provide information that allows for the fitting of more complex mathematical models. This is however clearly beyond the scope of this work.

There is an inclination that they are measuring something related to splicing based on the results in Figure 4B: donor cleavage time varies with departures from the consensus sequence. However, these changes don't seem to be consistent with the known biochemistry of 5' ss.

We discussed our data with senior colleagues who are in the splicing field and they spotted many observations that explain prior biochemical data. Note we have also performed extensive analyses supporting our findings beyond Figure 4B, including: 2C (lncRNA splice slower than mRNAs), 4A,C,D Figure 5, Figure 6 (including recovery of motifs of know splicing factors) and Figure 4—figure supplement 2 (nucleotide in physical contact with the spliceosome complexes). We are happy to discuss any deviations from known biochemistry in the manuscript but we would need this reviewer to specify such inconsistencies. The statement that changes do not seem to be consistent with known biochemistry must be substantiated and we need to obtain references to check into this before we can address this point.

3) The theoretical treatments are in general lacking in rigor.

We have substantially extended the description of the method, modeling assumptions, model interpretation, and performed simulations (Appendix). We think that this improved very much the manuscript. Again, we would need more specific comments if the reviewer wishes us to improve on a certain aspect. Without specification, we cannot further address this concern.

- Several papers have treated the kinetics of splicing and modeled kinetic data in single cells or even genome-wide (PMIDs: 25271374, 22022255, 25541195), yet the authors seem to be unaware of this work. Although one paper they do cite (Schmidt et al., 2011) indicates the importance of the correct theoretical treatment of elongation.

We have analyzed this published work in detail and made the following changes:

1) PMID 25271374 (Coulon et al., 2014)

We added the citation in the text. The authors used a dual color single-molecule RNA imaging to estimate the kinetic of β-globin reporter gene in human U2OS cells. They concluded that introns are spliced both before and after transcript release, although the majority of them is spliced after transcript release. Note that in the submitted manuscript, we have cited similar papers in which splicing rates have been measured in endogenous β-globin human gene (Martin et al., 2013) and with MS2-GFP tagged MINX intron gene (Schmidt et al., 2011) with single-molecule live imaging.

2) PMID 22022255 (Aikten et al., 2011)

The authors modeled splicing kinetics of a single-intron containing reporter Ribo1 gene in yeast, using RT-qPCR data. They concluded that splicing is predominantly co-transcriptional and point mutations at 3´SS reduce the probability that step I is happening co-transcriptionally (a sort of feedback). We included the citation in our text.

3) PMID 25541195 (Davis-Turak et al., 2015)

This is the only genome-wide analysis. The authors used a mathematical analysis to study cotranscriptional splicing. However, they do not analyze RNA labeling data. Their treatment of elongation relates to post-transcriptional splicing (i.e. to the time for polymerase to reach the polyA site) and not to the transcription of the intron. Moreover, Davis-Turak et al. did also used first order kinetics for modeling splicing of single introns for the very same reason than us, namely that more complex models did not improve the fits. We are now referring to this work when stating why we had to adopt first order kinetics models.

- There is no measure for the goodness of fits or examination of other models.

We are now providing in the Appendix subsection “Quality of fits”, scatterplots of read counts versus expected read counts according to the fits (Appendix—figures 16-18). These show very good agreements (Spearman rho >0.89 for all time points except for the RNA-seq samples with rho >0.8 for donor and acceptor sites reads). The lower correlation of the RNA-seq samples for donor and acceptor sites can be explained by low abundance of those unsplit reads at steady-state. Moreover, the aforementioned subsection of the Appendix provides plots demonstrating how accurate the model parameters are estimated when the ground truth is known (simulations, Appendix—figures 12-13, rho >0.99 for every parameter and model). We have also now investigated the accuracy of fitting a non-first order kinetic model to the donor site (model with delay). This showed that the delay is difficult to estimate accurately (Appendix—figure 2). Moreover, we are also now plotting the estimated synthesis rate of donor against the acceptor site. Assuming no polymerase drop- offs during transcription of the intron, these should agree. These parameters are estimated from distinct data (donor sit and acceptor site unsplit reads). While they agree reasonably well with the first order kinetic model (Appendix—figure 14, rho =0.42), the agreement is much poorer when using a delay model for the donor site (rho = 0.26, Appendix—figure 19).Altogether, these analyses demonstrate that the models fit well to the data and failure of a simple more complex model to do so.

- They use a worm-like chain model based on RNA persistence length to compute the energy of looping. With the energy, they apply the Arrhenius equation to get a rate and compare this value to the donor cleavage time (Figure 3E). However, whether there is any trend in this data and whether this model explains that trend is difficult to determine.

We agree with the reviewer that the trend is hard to read and we removed the figure and its discussion from the manuscript. In any case, this was very peripheral to our analysis.

4) I found the splicing yield part of the paper (Figure 7) to be the most interpretable of the results presented. Again, however, none of this was validated with any more quantitative measure such as digital droplet PCR.

We thank the reviewer for the positive comment about splicing yield. However,splicing yield is not directly accessible without kinetic modeling. Using PCR we cannot validate the model.

In summary, I see this analysis (with the underlying parameterization) as a reasonably good way to explain 'donor cleavage time' and 'acceptor cleavage time' as measured in this assay. And there may be some utility in such metrics. My overall difficulty is that I don't think this model parameterized in such a way is telling us much about splicing and transcription in a way that could be validated or interpreted in other assays. So I see this paper as a bioinformatic exercise lacking in biochemical restraint or biophysical insight.

We strongly disagree that our paper is a “bioinformatic exercise”. As demonstrated at various points throughout the manuscript, we obtain biological insights and can explain structural data. We also hope the revised version will convince this reviewer of the added value and practical necessity of modeling the kinetics of individual phosphodiester bonds, which are biochemical entities in their own right and which are directly measured by the assay. We furthermore hope that we could convince this reviewer of the difficulties of using more sophisticated models because they suffer from poor identifiability of the parameters. A paper should not be rejected by *eLife* based on notion that it is an “exercise”. Instead, the community should judge over the years to come how useful our data and conclusions are for moving the field forward. We kindly ask the reviewer to reconsider his/her opinion and to hope he/she can acknowledge the high technical quality of our work, the importance of the problem addressed, the advances made beyond the state-of-the-art, and the implications for future work in this interesting field.

Reviewer #3:[…] Given that the optimization methods used for estimating synthesis/formation and cleavage/degradation of the splice sites/junction is a local algorithm, is 10 random initialization enough to be confident that the global minimum has been reach?

We have used10 random initializations as previously (Eser et al., 2016). To further assess the quality of our data, we now compare simulated counts based on distribution of synthesis, splicing, degradation rate versus ground truth in the Appendix (Appendix—figures 1, 12-13). Based on these results, our model estimates ground truth of kinetic rates with high accuracy compared to the dynamic range of these parameters (Spearman rho >0.99 for all parameters). Moreover, these results empirically show that optimization of the cost function of the first order kinetic model does not suffer from local minima. This is in contrast to the delay model (Appendix—figure 2). We trust this clarifies the reviewers concern.

A key assumption for the model fitting is that 100% of the label is incorporated. The authors should explicitly explore how lower levels of labelling (e.g. 90, 50, 10%) affect their results and conclusions.

A lower level of label incorporation would lead to lower synthesis rate estimates (e.g. a labeling efficiency of 90% would result in a 10% underestimation of the rate) but would not affect estimation of donor and acceptor bond half-lives and of splicing yield. We added a paragraph to the appendix (Appendix subsection “4sU incorporation”) to explain this further.

Some comparison and correlation of synthesis rate estimated has been done separately for different donor sites and different acceptor sites (Figure 2—figure supplement 2B), but as mentioned, these should also be consistent for donor and acceptor of the same gene; however, donor synthesis rate vs acceptor synthesis rate is not shown.

We thank the reviewer to point this out. In the Appendix subsection “Agreement of kinetic parameters donor, acceptor and junction model”, we now show the correlation between donor and acceptor synthesis rate of the same gene. As expected under the assumption that there is no Pol II drop off or transcription termination before the end of the intron, we find that donor and acceptor synthesis rates correlate.

Model fits should be shown for individual genes, highlighting different classes: e.g. single vs multi-intron, highly expressed vs low expressed.

The individual fits are often noisy and hard to read because of the large deviations due to count data. This is in line with our reported accuracies of individual rates, which are relatively large (about 2-fold). Our findings rely on aggregated analyses. Moreover, we did not see a trend between these different classes. We therefore decided to not clutter the manuscript with more individual examples than the one in Figure 2A.

Kinetic language is not precise and causes confusion: For example "Cleavage times" should be referred to as "half cleavage times" or possibly "cleavage half-times". They do not refer to completion times, or characteristic times."Cleavage Rates in the range of minutes" is also imprecise. They seem to refer to cleavage half-life times being in in the minute range. More precise would be to report the actual cleavage rate constants, in addition to the cleavage half-life times. ("Rate" = rate constant x abundance).

We have revised the terminology, also in response to reviewer 3. We are now using the terms ‘donor bond half-life’, referring to the half-life of the exon-intron nucleotide bound and ‘acceptor bond half-life’, referring to the half-life of the intron-exon phosphodiester bond. We agree that ‘rate constant’ is more accurate than ‘rate’. We note however that the literature (including the physics literature, e.g. see the usage of decay rate) also often uses the term rate in lieu of rate constant. We thank the reviewer for pointing this out, these changes have improved the language.

The part on the single nucleotide model methods would benefit of a more detailed description. It seems to be a statistical model based on the previously derived cleavage time and consensus splice site sequence but the methods is not very clear to me.

We have re-written this section of the Materials and methods to give more explanations also for readers not versed in statistical modeling (Lasso regression).

[Editors' note: the author responses to the re-review follow.]

Essential revisions:1) The reviewers remain unconvinced that the measured decay and formation rates of phosphodiester linkages can be directly attributed to individual recognition steps and enzymatic rates. By the time a chemical step causes the loss or gain of a junction read, dozens of events have occurred that could contribute to the rate of that step. For a 5' splice site these include initial recognition by U1, transcriptional elongation/pausing in the downstream intron, branchpoint recognition (which cannot be measured here), myriad spliceosome assembly steps, and finally catalysis. While the effects of splice site mutations agree with the known interaction of the 5' splice site by the U1 snRNP, other moieties will be interacting with these residues, and there is no evidence that the U1 interaction is determinative of the rate of cleavage. It is very simplistic to claim this is so. The rate of exon-intron junction loss will be the aggregate function of many different rates that likely vary from site to site and are not being measured. It should be noted that U1 is not involved in catalytic steps, only initial recognition. To say that "we developed a computational approach that models rates of RNA metabolism at the level of single phosphodiester bonds" is misleading. The authors need to examine their interpretations of their data and be careful not to claim that the rates they are observing can be broken down into steps that are not being measured.

The reviewers are correct. We agree that the rates of single phosphodiester bonds cannot be unambiguously related to individual chemical reactions. We did not mean to suggest so, but obviously there were misunderstandings along these lines. We have therefore carefully revised every sentence related to individual chemical reaction rates so that our claims only relate to the phosphodiester bonds overall kinetics. Also, we changed the sentence “we developed a computational approach that models rates of RNA metabolism at the level of single phosphodiester bonds” to “we developed a computational approach that estimates the metabolic rates of single phosphodiester bonds”.

2) Subsection “New and alternative splice sites”, last paragraph: The authors reported numbers (177,322 and 164,533) of non-split reads mapping to exon-intron boundaries (5' splice sites) sum to the total number of putative introns reported in the paragraph above (341,855, identified by split reads across an exon-exon boundary). This seems wrong, or did they throw out all examples of alternative splicing? They don't report numbers for intron-exon boundaries (3' splice sites), but farther down in the same paragraph they treat the read numbers as if these sites are included. In that later discussion, the authors state that of the split reads, "18% contained a non-annotated donor site, 22% a non-annotated acceptor site and another 38% represented new putative splice sites". What is the difference between a non-annotated donor or acceptor site and a new putative splice site? Furthermore, if aiming to differentiate between annotated and non-annotated sites, wouldn't it make more sense to start with the percentage of split reads mapping to non-annotated regions, where all the sites are likely unannotated, and then discuss new previously unannotated sites in GENCODE?

We were using “split reads” instead of “putative introns” in this part of the text. We have fixed the text and thank the reviewer for spotting this. Also, the 38% correspond to putative introns that map to non-annotated donor sites and acceptor sites. We have made this clear now. Finally, we feel that starting or ending with filtering for unannotated regions does not make a fundamental difference in the way these numbers are presented.

3) The authors indicate that they are fitting the model independently to every donor-acceptor combination that they observe. If so, what is causing the differences in the numbers of donor (162,134), acceptor (177,543), and junction (156,825) bonds that they are able to model? Don't they need sufficient read coverage across all three to confidently estimate the synthesis/cleavage rates on a per-junction level? Or does this include some cases of alternative splice site pairings?

We apologize if we did not explain this clear enough in the text. In order to measure the metabolism of the single bond, we individually model the three bonds (donor, acceptor and junction) for each detected intron. We consider each bond with at least 100 supporting reads across the datasets. Therefore, we do not need to combine their coverage cutoffs. We now stress further that these models are uncoupled with the sentence “we modeled the steady-state rates of synthesis and degradation (or equivalently cleavage) of each of three different phosphodiester bonds individually” in the Results section.

4) The use of simulations to estimate the robustness of the model fitting, across both the first order kinetic model and alternative models is a helpful addition to the paper. It should be highlighted more in the main text how this supports the validity of the method. An assessment should also be provided for how the method performs and where it doesn't work well. For example, the authors indicate that "the robustness of our approach… consistent with high accuracy of the fitting procedure on simulated data." However, all models or experimental systems are underpowered in certain cases. The authors should include a short discussion of where their model falls short (ultra-fast rates, length considerations, etc.) and thus more fully inform the reader about the biological interpretations that can be drawn from the data. Simulations are often the best way to push the model and test these extreme cases.

We thanks the reviewers for the suggestions. We have included a new paragraph that explains the results of these simulations and the new simulations we have conducted in response to this point to explore the extreme cases. We added a new section in the Appendix (subsection “Limits of the model”) in which we investigate the limits of our model. We simulated counts based on the constant rate model given a large range of log-uniform distributed bond synthesis rates (10^-5^ – 100 1 / cell / min) and bond half-lives (10^-3 –^ 7x10^-6^ min). Based on this data, we investigated the typical lower bound of reads necessary in all samples to proper model the kinetics of a bond (Appendix—figure 23). We found that a lower cut-off of 100 reads (after the 6^th^ ventile of simulated data) allows us to estimate the kinetics with a typical error below 100%. The cut-off of 100 reads is the one we have applied to the real data. To test the shortest and longest bond half-lives that our experimental approach is able to capture and model, we stratified the results by bond half-live. We then proceeded similarly with the bond synthesis rates. As a result, we found that most of our modelled data lies within a range in which the median relative error of synthesis rates and half-lives is below 100% or 30%, respectively (Appendix—figures 24, 25).

5) In the section entitled "Intron length constrains co-transcriptional splicing times." It is not clear what the authors mean by "co-transcriptional".

As we defined in the Introduction, splicing is “co-transcriptional” when it occurs on newly synthesised RNA still attached to the RNA Pol II machinery. This does not necessarily imply that introns are excised as soon as the acceptor site gets transcribed. Nonetheless, the results described in this section do not depend on the co-transcriptional nature of splicing. We therefore changed the title of the subsection to avoid misunderstandings.

The authors describe how donor bond half-life increases with intron length, presumably indicating that donor bond half-life is dependent on the time to reach the branchpoint and acceptor site. This would imply that the first step of splicing occurs soon after the acceptor site is transcribed and available.

Not necessarily. We interpret these observations provided that the first step of splicing occurs after (but not necessarily soon after) the branch point and the acceptor site are transcribed, which are close to each other – typically less than 40 nt. We do not claim that the first step occurs “soon after the acceptor is transcribed and available”, but rather that the length between donor and branchpoint is a limiting factor for splicing kinetics only in very long introns (>10.000kbs). We have made sure this is properly understood.

If so, then why is the acceptor bond half-life so long? A median acceptor half-life of 4 min indicates that the Polymerase is on average 16kb away by the time the second step of splicing occurs.

This could be true since 16 kb is not excessively long for a human gene (median size 26kb, average size 66kb). Moreover, Pol II is known to pause at exons (Mayer, et al., 2015), and such pausing is in the range of minutes, and hence the polymerase may not have progressed far within 4 min.

Do they propose that the first step is close in time to the second step, with a lag between transcription of the branchpoint and the loss of the donor? Or are the two chemical steps separated by 4 minutes? This seems unlikely in that it implies that spliceosome assembly up to C complex formation is very fast compared to the transition from the first to the second catalytic steps.

As mentioned in the response to the above questions, we are estimating overall rates but cannot untangle the rates of the individual biochemical steps contributing to the overall rates. Except for very long introns, the half-life of the acceptor bond is similar to the half-life of the donor bond (Figure 3B). Hence our data actually suggests (in agreement with the reviewer’s intuition) that spliceosome assembly up to C complex formation is slow compared to the transition from the first to the second catalytic steps. We now provide this interpretation of Figure 3B in the manuscript.

If the authors think that donor loss occurs closer in time to the second step – well after the appearance of the branchpoint – then the definition of "co-transcriptional" that involves immediate splicing upon transcription/availability of the acceptor site doesn't seem to hold. All of this needs better explanation.

We do not wish to provide an extensive discussion of the co-transcriptional nature of splicing because this would require a different type of data that can be related to nascent RNA and transcribing Pol II. We however checked that we do not make statements on the co-transcriptional nature that are definitive and unsupported. At this time, with no additional data in hand, we do not wish to go beyond this, and trust the reviewer understands.

6) It is not clear how the splicing yield calculation can produce a value of > 1, with the median splicing yield reported for mRNAs as 1.2. Does this mean that the measured levels of precursor mRNAs are consistently underestimated? If it's not bounded as described (0-1), the splicing yields are difficult to assess as it seems one or more of the measured values is incorrect.

The yield is computed by independently estimating the synthesis rate of the precursor (using unsplit reads) and of the mature RNA (using split reads) and taking the ratios. Due to estimation errors, these ratios may turn out to be greater than 1. We find that this is acceptable as long as it does not affect our conclusions. We found by simulations that the cumulative read coverage across all samples as well half-life can lead to bias estimates of synthesis rates. However, the higher yield of mRNA versus non-coding RNA was recapitulated when stratifying by these possible confounders. We have now included these analyses (Appendix—figures 26, 27).